# Evaluation of regional climate models ALARO-0 and REMO2015 at 0.22° resolution over the CORDEX Central Asia domain

Sara Top[1,2], Lola Kotova[3], Lesley De Cruz[4], Svetlana Aniskevich[5], Leonid Bobylev[6], Rozemien De Troch[4], Natalia Gnatiuk[6], Anne Gobin[7,8], Rafiq Hamdi[4], Arne Kriegsmann[3], Armelle Reca Remedio[3], Abdulla Sakalli[9], Hans Van De Vyver[4], Bert Van Schaeybroeck[4], Viesturs Zandersons[5], Philippe De Maeyer[1], Piet Termonia[2,4], Steven Caluwaerts[2,4]

[1]Department of Geography, Ghent University (UGent), Ghent, 9000, Belgium
[2]Department of Physics and Astronomy, Ghent University (UGent), Ghent, 9000, Belgium
[3]Climate Service Center Germany (GERICS), Helmholtz Zentrum Geesthacht, Hamburg, 20095, Germany
[4]Royal Meteorological Institute of Belgium (RMIB), Brussels, 1180, Belgium
[5]Latvian Environment, Geology and Meteorology Centre (LEGMC), Riga, LV - 1019, Latvia
[6]Nansen International Environmental and Remote Sensing Centre (NIERSC), St. Petersburg, 199034, Russia
[7]Flemish Institute for Technological Research (VITO), Mol, 2400, Belgium
[8]Department of Earth and Environmental Sciences, Faculty of BioScience Engineering, Heverlee, 3001, Belgium
[9]Iskenderun Technical University, Iskenderun, 31200, Turkey

*Correspondence to*: Sara Top (sara.top@ugent.be)

**Abstract.** To allow for climate impact studies on human and natural systems high-resolution climate information is needed. Over some parts of the world plenty of regional climate simulations have been carried out, while in other regions hardly any high-resolution climate information is available. CORDEX Central Asia is one of these regions and this article describes the evaluation for two regional climate models (RCMs), REMO and ALARO-0, that were run for the first time at a horizontal resolution of 0.22° (25 km) over this region. The output of the ERA-Interim driven RCMs is compared with different observational datasets over the 1980-2017 period. The REMO model scores better for temperature, whereas the ALARO-0 model prevails for precipitation. Studying specific subregions provides a deeper insight into the strengths and weaknesses of both RCMs over the CAS-CORDEX domain. For example, ALARO-0 has difficulties in simulating the temperature over the northern part of the domain, particularly when a snow cover is present, while REMO poorly simulates the annual cycle of precipitation over the Tibetan Plateau. The evaluation of minimum and maximum temperature demonstrates that both models underestimate the daily temperature range. This study aims to evaluate whether REMO and ALARO-0 provide reliable climate information over the CAS-CORDEX domain for impact modelling and environmental assessment applications. Depending on the evaluated season and variable, it is demonstrated that the produced climate data can be used in several subregions e.g. temperature and precipitation over West Central Asia in autumn. At the same time, a bias adjustment is required for those regions where significant biases have been identified.

# 1 Introduction

There is a strong need for climate information at the regional-to-local scale that is useful and usable for impact studies on human and natural systems (Giorgi et al., 2009). In order to accommodate for this, the World Climate Research Program (WCRP) Coordinated Regional Climate Downscaling Experiment (CORDEX) was initiated with the aim to design and conduct several high-resolution experiments over prescribed spatial domains across the globe. CORDEX creates a framework to perform both dynamical and statistical downscaling, to evaluate these regional climate downscaling techniques and to characterize uncertainties of regional climate change projections by producing ensemble projections (Giorgi and Gutowski, 2015). Within CORDEX there are large ensembles of model simulations available at different resolutions for the Africa (Nikulin et al., 2012; Nikulin et al., 2018), Europe (Jacob et al., 2014; Kotlarski et al., 2014), Mediterranean (Ruti et al., 2016) and North America (Diaconescu et al., 2016; Whan and Zwiers, 2017; Gibson, 2019) CORDEX regions (Gutowski et al., 2016). These large ensembles consist of more than ten different global-regional climate models (GCM-RCM) combinations. In order to provide such ensembles over all CORDEX regions, coordinated sets of experiments were recently performed or are still ongoing for CORDEX regions such as South America (Solman et al., 2013), Central America (Fuentes-Franco et al., 2015; Cabos et al., 2019), South Asia (Ghimire et al., 2018), East Asia (Zou et al., 2016), South-East Asia (Tangang et al., 2018; Tangang et al., 2019; Tuyet et al., 2019), Australasia (Di Virgilio et al., 2019), Arctic (Koenigk et al., 2015; Akperov et al., 2018), Antarctic (Souverijns et al., 2019) and Middle East North Africa (Almazroui et al., 2016; Bucchignani et al., 2018). In addition, a new ensemble of climate change simulations covering all major inhabited regions with a spatial resolution of about 25 km has been established within the WCRP CORDEX COmmon Regional Experiment (CORE) Framework to support the growing demands for climate services (Remedio et al., 2019). Furthemore, a number of high-resolution global simulations at climatic timescales, with resolutions of at least 50 km in the atmosphere and 28 km in the ocean, have been performed within the Coupled Model Intercomparison Project 6 (CMIP6) (Haarsma et al., 2016).

While high-resolution ensembles (up to 0.11° or 12.5 km spatial resolution) are available for certain regions, e.g. EURO-CORDEX (Jacob et al., 2014), for other regions such as Australasia (Di Virgilio et al., 2019) and the Antarctic (Souverijns et al., 2019) the first experiments were performed only recently. For the CORDEX Central Asia (CAS-CORDEX) domain only a single climate run was publicly available through the Earth System Grid Federation (ESGF) archive until 2019. This was performed by the Met Office Hadley Centre (MOHC) with the regional climate model (RCM) HadRM3P (Jones et al., 2004) at a resolution of 0.44°, insufficient for most impact modelling and environmental assessment applications. In addition, climate projections with the RegCM model at 0.44° resolution for the 2071-2100 period and different emission scenarios were reported in Ozturk et al. (2012, 2016), however they are not available through the ESGF archive. Thus higher-resolution climate data over the CAS-CORDEX region is needed (Kotova et al., 2018). Recently, Russo et al. (2019, 2020) presented model evaluation results of the COSMO-CLM 5.0 model run at 0.22° or 25 km resolution over the CAS-CORDEX region. In this study we aim to address the scarcity of reliable climate information over the CAS-CORDEX domain by evaluating two different RCMs based on multiple scores for temperature (mean, minimum and maximum) and precipitation over the longer period of 38 years.

In order to fill the knowledge gap over Central Asia two RCMs, ALARO-0 and REMO, were run over this region at 0.22° resolution in line with the CORDEX-CORE protocol (CORDEX Scientific Advisory Team, consulted on 01/03/2019). Here we present the model evaluation through the use of so-called "perfect boundary conditions" taken from reanalysis data and by comparing the downscaled results to observed data for the period 1980-2017. Such a study is necessary to gain confidence in the RCM downscaling procedure before its application in the context of climate projections where the RCM is driven by a

GCM (Giorgi and Mearns, 1999). The methodology for evaluation is partially based on Kotlarski et al. (2014) and Giot et al. (2016), that compared a large ensemble of RCMs over the EURO-CORDEX region with the high-resolution E-OBS observational dataset (Hofstra et al., 2009). However, in this study a slightly different approach is necessary due to the absence of an ensemble of RCM runs over Central Asia. Additionally, in some regions the quality of gridded observational datasets, constructed through interpolation or area-averaging of station observations, is poor due to over-smoothing of extreme values

(Hofstra et al., 2010) and/or because of station observations that are nonrepresentative for their large-scale environments. This is particularly the case for orographically complex regions such as the Himalayas. The current study compares the model simulations with different gridded observational datasets and reanalysis data. When the different datasets show large deviations and a large spread then their uncertainty is high and no robust conclusions can be drawn (Collins et al., 2013; Russo et al., 2019).

This study contains two assets: for the first time an in-depth evaluation of the RCMs ALARO-0 and REMO is performed at 0.22° spatial resolution over the CAS-CORDEX domain and we reflect on the impact of the observational datasets on the model evaluation. Such an analysis is a prerequisite in order to be able to use the climate data in a sound way for later impact studies, e.g. for investigating climate change impacts on crop yields and biomass production in forest ecosystems, which will be done in the framework of the AFTER project (Kotova et al., 2018).

In the following section we describe the applied methodology for this study (Sect. 2). This section contains details about the study area, the model description, datasets used for the evaluation and the methodology of the analysis. In Sect. 3, we describe the annual cycle, seasonal and annual means, biases and variability of mean, minimum and maximum surface air temperature and precipitation. Further, we evaluate and provide a discussion of some remarkable anomalies in Sect. 4 and in the final Sect. 5 we summarize the conclusions.

## 2 Methods

### 2.1 CORDEX Central Asia domain and subregions

The CAS-CORDEX domain as shown in Fig. 1 contains Eastern Europe, a large part of the Middle East (including: Saudi-Arabia, Jordania, Syria, Iraq and Iran) and Central Asia (including: Kazakhstan, Uzbekistan, Turkmenistan, Afghanistan, Pakistan, Tajikistan, Kyrgyzstan and Mongolia). The majority of Russia and China (excluding the most eastern provinces) and

the northern part of India are included as well. This domain is an exceptional CORDEX domain in the sense that it barely covers any open ocean. It contains several important mountain ranges e.g. Ural, Caucasus, Altay and Himalaya, and deserts

e.g. Arabian, Karakum, Thar, Taklamakan and Gobi desert. Mountainous environments are of special interest for regional climate modelling since global climate models do poorly resolve the mountain ranges with a spatial resolution less than 0.50° and hence RCMs may have an added value here (Torma et al., 2015). In addition, the CAS-CORDEX domain contains a wide

range of climatic and bioclimatic zones, such as permafrost in the north and the hot regions and monsoon-driven climates with abundant precipitation linked to the Inter-Tropical Convergence Zone (ITCZ) passing in the south.

In order to obtain simulations that allow for coordinated intercomparisons, the CORDEX initiative prescribes the minimum inner domain of each CORDEX region that the RCM has to cover. While REMO uses the exact rotated lat-lon CAS-CORDEX grid (Jacob et al., 2007) described by the CORDEX community, ALARO-0 has adopted a conformal Lambert projection (Giot

et al., 2016), which implies that the non-rotated boundary box should be applied in order to define the domain. The grids were set up in such a way that the CAS-CORDEX domain is completely covered by the model domain excluding the relaxation zone. The CAS-CORDEX 0.22° ALARO-0 inner domain encompasses 333 by 223 grid boxes, while REMO circumscribes 309 and 201 grid boxes in the east-west direction and north-south direction, respectively. The outer domain for both RCMs consists of the inner domain plus a relaxation zone of eight grid points at every boundary.


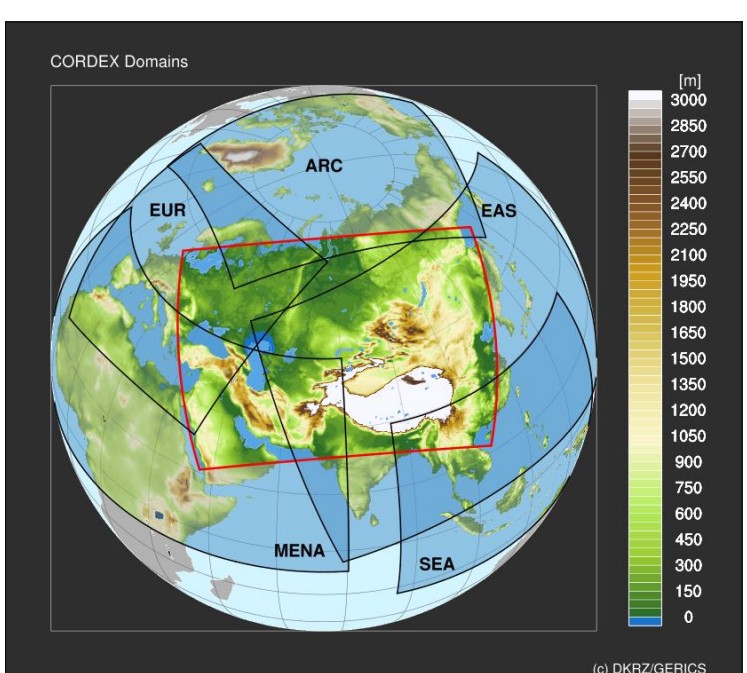

**Figure 1: The CAS-CORDEX domain delineated by a red contour and the main overlapping CORDEX domains (black contour lines): Europe (EUR), Arctic (ARC), South East Asia (SEA), East Asia (EAS) and MENA projected upon the topography of Eurasia (geopotential height [m] of the GTOPO30 global digital elevation model (DEM) 3). All points with orography higher than 3000 m**
**are colored white.**

The CAS-CORDEX domain overlaps with eight other CORDEX domains, including the ones covering Europe, the Arctic, East Asia, South East Asia, South Asia, Africa/MENA and the Mediterranean. Both RCMs used in this study, ALARO-0 and

REMO, were already run and evaluated over the EURO-CORDEX region (Kotlarski et al., 2014; Giot et al., 2016) and additionally, REMO has been validated over five other overlapping CORDEX regions (Remedio et al., 2019).

In the present paper, the CAS-CORDEX domain was further subdivided into five subregions according to the IPCC reference regions (Iturbide et al., 2020) named as: East Europe, West Siberia, East Siberia, West Central Asia and Tibetan Plateau. These subregions, visualized in Fig. 2, were applied to evaluate the spatial differences in the study area and to investigate whether there were differences in the simulation of subcontinental processes.

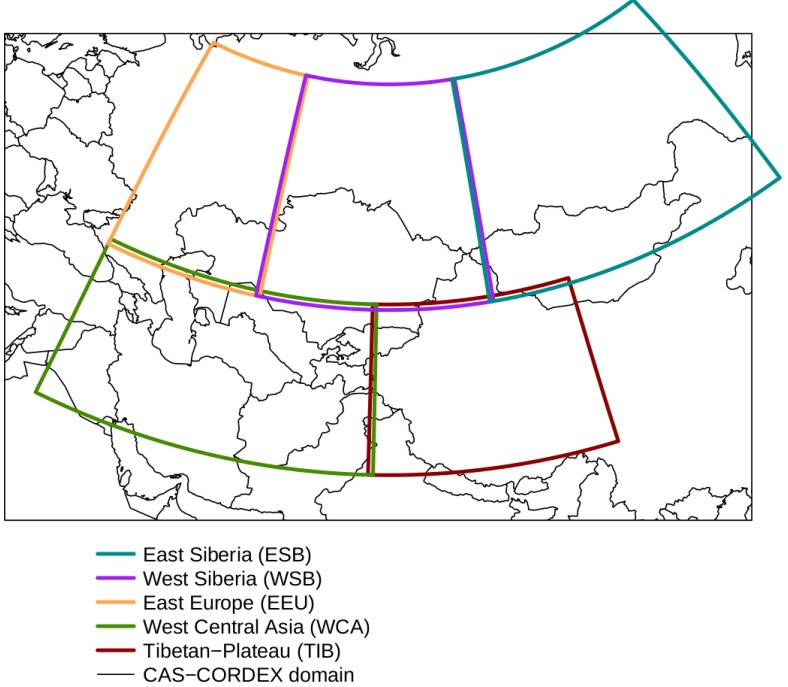

**Figure 2: IPCC6 subregions projected on the CAS-CORDEX region.**

**2.2 Model description and experimental design**

REMO and ALARO-0 are hydrostatic atmospheric circulation models aimed to run over limited areas. The ALARO-0 model is a configuration of the ALADIN model (ALADIN international team, 1997; Termonia et al. 2018a) which is developed,
maintained and used operationally by the 16 countries of the ALADIN consortium. The dynamical core of the ALADIN model is based on a spectral spatial discretization and a semi-implicit semi-Lagrangian time stepping algorithm. The ALARO-0 configuration is based on the physics parameterization scheme 3MT (Modular Multiscale Microphysics and Transport (Gerard et al. 2009)), which handles convection, turbulence and microphysics. ALARO-0 has been used and validated for regional climate studies (Hamdi et al., 2012; De Troch et al., 2013; Giot et al., 2016; Termonia et al. 2018b).
The REMO model is based on the Europa Model, the former NWP model of the German Weather Service (Jacob, 2001). The model development was initiated by the Max-Planck-Institute for Meteorology and is further maintained and extended by the

Climate Service Center Germany (HZG-GERICS). The physical parameterization originates from the global circulation model ECHAM4 (Roeckner et al., 1996), but there have been many further developments (Hagemann, 2002; Semmler et al., 2004; Pfeifer, 2006; Pietikäinen et al., 2012; Wilhelm et al., 2014). REMO is used in its most recent hydrostatic version, REMO
2015, and the dynamical core uses a leap-frog time stepping with semi-implicit correction and Asselin-filter. For both RCMs, the vertical levels are based on hybrid normalized pressure coordinates which follow the orography at the lowest levels. For the ALARO-0 experiment 46 levels were used whereas the REMO run employs 27 levels. More details on the general setup of ALARO-0 can be found in Giot et al. (2016) and for REMO we refer to Jacob et al. (2001) and Jacob et al. (2012). An overview of the model specifications is given in Table S1 of the supplementary material.

In order to evaluate both RCMs, a run driven by a large-scale forcing taken from the ERA-Interim global reanalysis (Dee et al., 2011) was undertaken for the period 1980-2017. A one-way nesting strategy was applied to dynamically downscale the ERA-Interim data, having a horizontal resolution of about 0.70° (approximately 79 km), to a higher resolution over the CAS-CORDEX domain (Denis et al., 2002). The ERA-Interim forcing data has been prescribed at the lateral boundaries using the Davies (1976) relaxation scheme and the downscaling has been performed to a horizontal resolution of 0.22° (approximately
25 km). Both model experiments are continuous runs initialized on the 1st of January 1979 and then forced every 6 hours at the boundaries up to December 31st 2017. Furthermore, constant climatological fields for some parameters were used and updated monthly following the methodology of Giot et al. (2016). These include sea surface temperatures (SSTs), surface roughness length, surface albedo, surface emissivity and vegetation parameters. A spin-up period was needed to allow the models and their surface fields to adjust to the forcing and internal model physics (Giot et al., 2016). While for ALARO-0 the
year 1979 was designated as spin-up year, REMO was spun-up for 10 years to allow the model to reach an equilibrium state for the soil temperature and soil moisture. These soil fields were then used as initial soil conditions when restarting the model from 1979. The data produced by both models have been uploaded to the ESGF data nodes (website: http://esgf.llnl.gov/).

## 2.3 Reference datasets

In order to validate the model results, monthly, seasonally and annually averaged values for temperature and precipitation were
compared with different reference datasets. Gridded datasets are based on interpolated station data and are used instead of station observations to overcome the scale difference between the model and observation field (Tustison et al., 2001). A multitude of datasets were considered to assess the reliability of the gridded observational temperature and precipitation (Gómez-Navarro et al., 2012). The reference datasets are briefly presented in Table 1 and in the next sections we give a more detailed overview of the different datasets used in this study.

### 2.3.1 Climatic Research Unit TS dataset

The gridded Climatic Research Unit (CRU) TS dataset (version 4.02) contains ten climate related variables for the period 1901-2018 at a grid resolution of 0.50° covering the global land mass (excluding Antarctica) (Harris et al., 2020). Monthly values of minimum, maximum and mean near surface air temperature and precipitation were used in the current study. This

dataset is widely used all over the world and in a wide range of disciplines, although some issues have been reported (Harris
et al., 2020), with the main concerns including sparse coverage of measurement stations over certain regions, e.g. Northern
Russia and the dissimilarities in measurement methods that are used by different countries (Harris et al., 2020). In the present
paper, this dataset is used as the reference while the spread of the data in all of the datasets is used to assess the reliability over
the different areas.

### 2.3.2 Matsuura and Willmott gridded dataset

The Matsuura and Willmott (MW) (version 5.01) gridded dataset of the University of Delaware contains monthly values at a
0.5° resolution based on temperature and precipitation station observations. The main differences with the CRU dataset are
the use of different measurement station networks and spatial interpolation methods (Willmott and Matsuura, 1995; Harris et
al., 2020). Additionally, this dataset only contains monthly values of mean near surface air temperature and precipitation,
which are used in this study. It is known that the MW dataset generally underestimates the precipitation in the central part of
the CAS-CORDEX domain, especially during spring (Hu et al., 2018). The MW dataset contains globally up to 0.4 °C warmer
temperatures for the latest decades compared to CRU (Harris et al., 2020).

### 2.3.3 Global Precipitation Climatology Centre dataset

The Global Precipitation Climatology Centre (GPCC) (version 2018) of the German Weather Service is a monthly land surface
precipitation dataset at 0.25° resolution based on rain gauge measurements. The GPCC full data monthly product (version
2018) contains globally regular gridded monthly precipitation totals. This updated version is using "climatological infilling"
to avoid interpolation artefacts for regions where an entire 5° grid is not covered by any station data (Schneider et al., 2018).
Hu et al. (2018) concluded for the central part of our domain that GPCC is more in line with the observed station data in
Central Asia compared to CRU and MW. For this region, they also found that precipitation is underestimated in mountainous
areas and precipitation is slightly underestimated overall by GPCC, especially during spring. In addition, the GPCC has no
similar dataset for other variables and thus, only precipitation can be validated with this dataset.

### 2.3.4 ERA-Interim

Reanalysis products like ERA-Interim are more continuous in space and time than station data, but they do also contain biases.
The ERA-Interim reanalysis of the European Centre for Medium-Range Weather Forecasts (ECMWF) is available from 1979
onwards. The spatial resolution of the dataset is approximately 0.70° (T255 spectral) with 60 levels in the vertical direction
from the surface up to 0.1 hPa (Dee et al., 2011). The ERA-interim data have been further interpolated to be used as forcing
for both RCMs at a spatial resolution of 0.25°. Moreover, the ERA-Interim data is used to study the spread between
observational gridded datasets and reanalysis data. To evaluate precipitation, total monthly precipitation was obtained from
the Monthly Means of Daily Forecast Accumulations dataset. The Monthly Means of Daily Means data at the 2 m temperature
level are used for the mean temperature, while the minimum and maximum temperatures are retrieved by extracting the

minimum and the maximum respectively from the 3-hourly ERA-Interim forecasts. Several studies have shown that ERA-Interim tends to have a warm bias in the northern part of the CAS-CORDEX region, especially during winter (Ozturk et al., 2012 and 2016). Ozturk et al. (2012) relates this to the insufficient ability of ERA-Interim to produce a snow cover in winter. Additionally, ERA-Interim globally overestimates precipitation, particularly over mountainous regions (Sun et al., 2018).

### 2.4 Analysis methods

The grids of the observational and reanalysis datasets generally differ from the model grid. Therefore, an interpolation to one common grid is needed in order to compare them (Kotlarski et al., 2014). The output of the RCMs was upscaled and bilinearly interpolated to the 0.50° resolution grid of the observational gridded datasets.

For ALARO-0 and REMO, hourly values for temperature at 2 m and convective and stratiform rain and snow are available. The precipitation variables were added up in order to obtain the hourly total precipitation which in turn was used to calculate monthly totals and seasonal and annual means. Seasons are defined as meteorological seasons, where winter includes: December, January and February (DJF), spring: March, April and May (MAM), summer: June, July and August (JJA), and autumn: September, October and November (SON).

The diurnal temperature range was obtained by subtracting the minimum temperature from the maximum temperature and a height correction was performed for mean, minimum and maximum temperature assuming a uniform temperature lapse rate of 0.0064 K m$^{-1}$.

The model evaluation was done by calculating different evaluation metrics over the CAS-CORDEX domain and the defined subregions for the 1980-2017 period. We computed the monthly, seasonal and annual climatological means of the evaluated variables to obtain graphs of the annual cycle and maps that visualize the spatial patterns of the bias between the RCMs and reference datasets. The relative bias for precipitation is computed by subtracting the CRU value from the RCM and dividing it by the CRU value.

The climatological means, biases and mean absolute errors (MAE) were spatially averaged to obtain one mean value over the complete domain and each of the subregions, respectively. Moreover, Taylor diagrams were produced in order to study the model performance for the different seasons and for annual means. These diagrams supplement the bias analysis by visualizing in a concise way information about the spatial correlation, the centered root mean square error (RMSE) and the ratio of spatial variability (RSV) between the model and the observational dataset (Taylor, 2001). These metrics are computed over all grid points of the CAS-CORDEX domain. The RSV is defined as the ratio of the model standard deviation and the standard deviation of the reference dataset (CRU in this case) averaged over the domain. For the used formulas we refer to appendix A of Kotlarski et al. (2014).

Limitations of the observational datasets should be kept in mind when interpreting the evaluation results (Kotlarski et al., 2014). These limitations are investigated by comparing the different observational datasets and their implications for the evaluation will be described in Sect. 4. The spread between the different reference datasets (observational datasets and ERA-

Interim reanalysis dataset) is calculated for each grid point by computing the difference between the maximum and the minimum value of the different datasets, and this for every 3-month period (season) averaged over the 1980-2017 period.

## 3 Results

In this section, the results of the model evaluation are presented with a focus on evaluation metrics of seasonal means of mean, minimum and maximum near surface air temperature (henceforth denoted as temperature) and seasonal mean precipitation (henceforth precipitation). This is done for the complete CAS-CORDEX domain and for the five subregions.

### 3.1 Mean temperature

Figure 3 shows the mean seasonal and annual temperature observations of CRU, the model biases with respect to CRU and the spread between the reference datasets (ERA-Interim, MW and CRU)for the 1980-2017 period.  Table 2 shows the spatially averaged mean seasonal and annual CRU temperature for the 1980-2017 period over the CAS-CORDEX domain and subregions, the biases and MAE of the RCMs (REMO and ALARO-0) and the other reference datasets (ERA-Interim and MW) against CRU.

Both RCMs are producing similar mean annual temperature patterns in the western part of the domain since they have similar biases with respect to CRU (Fig. 3). Contrasting error patterns can be seen in the temperature bias of ALARO-0 between north and south and for REMO between east and west, with a peak in positively biased temperatures over north-western Mongolia. Annual biases generally vary between -3°C and 3°C for both RCMs, with the exception of orographically complex regions and some areas in North and East Siberia for ALARO-0. The biases and MAE of the annual mean temperature are very comparable between ALARO-0 and REMO (table 2), with small biases and MAEs that are only slightly larger than the spread of the observational datasets.

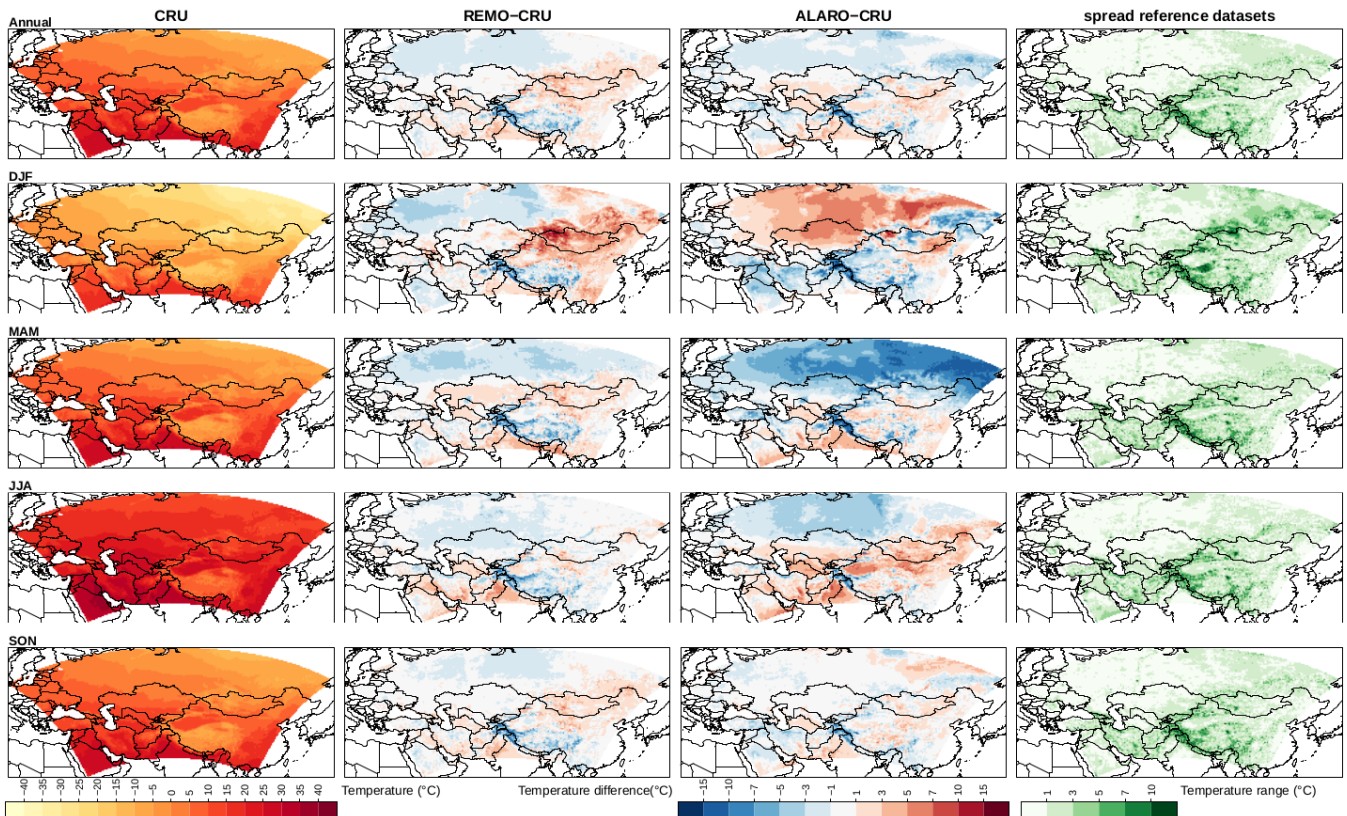

**Figure 3: Left column: mean air temperature (°C) at 2 m height over the CAS-CORDEX domain based on the observational CRU dataset for the 1980-2017 period on annual level and for winter (DJF), spring (MAM), summer (JJA) and autumn (SON). In the middle columns: difference in mean temperature between models and CRU. Right column: the range in mean temperature (°C) between the different reference datasets (CRU, MW and ERA-Interim).**

On the seasonal timescale, biases over larger areas are mainly pronounced in winter (DJF) and spring (MAM). In particular, both models locally show strong biases in the north-eastern part of the domain for winter with values ranging up to 15 °C. Additionally, ALARO-0 shows strong negative biases up to -15 °C during spring in this area. These large biases are reflected by the values in Table 2 for the northern subregions EEU, WSB and ESB for ALARO-0 and the ESB subregion for REMO,. Additionally, the REMO model has a cold bias in the western part of Russia during winter, while ALARO-0 shows a warm bias. During spring, cold biases are found for both models in the northern part of the domain, but the biases of ALARO-0 are more pronounced than those of REMO (Fig. 3 and Table 2). For the summer (JJA) season, warm biases occur over the southern part of the domain for both RCMs, with exception of some regions such as the Himalayas, south-eastern China and the northern border of Iran, which exhibit cold biases. On the contrary, cold biases in summer are overall more dominant in the north. These biases in summer are more pronounced for ALARO-0. The small mean bias during summer (JJA) for ALARO-0 over the complete domain (Table 2) is the result of averaging the warm biases in the south and the cold biases in the north (Fig. 3). Both models have the smallest biases and MAE over the ESB region in this season (Table 2). Both models show modest bias patterns in autumn (SON), with notably modest warm biases over the eastern part of the domain (Fig. 3). In agreement with

Fig. 3 the spatial averaged biases and MAE in Table 2 are small for both RCMs during autumn, especially for East Europe (EEU), the west and central Russian region and Kazachstan (WSB).

Biases in the high-altitude regions are largely persistent throughout the seasons. More specifically, both RCMs have large negative biases over the Pamir Mountains (Tadjikistan) and the Himalayas, while they also feature negative biases over the Tibetan Plateau, although this is to a lesser extent the case for ALARO-0 where this is only clearly visible for the winter season.

Figure 4 shows the normalized Taylor diagram expressing the spatial performance of mean temperature for seasonal and annual means for both RCMs (ALARO-0 and REMO), the ERA-Interim reanalysis and MW observational data with respect to CRU for the five subregions and the complete CAS-CORDEX domain.

Both models have in general a good performance for annual and seasonal temperature over the CAS-CORDEX domain since the spatial correlation between the model output and the CRU data is high ($> 90$ %), while the centred RMSE is small ($< 0.5$) and the normalized RSV is mostly close to 1. Moreover, the spatial correlation is high ($> 90$ %) for ALARO-0 over all subregions at the annual level. Annual mean temperatures of REMO have slightly lower spatial correlations with CRU when compared to those of ALARO-0, but they are still high ($> 90$ %), except for the ESB subregion.

On the other hand, the Taylor diagrams for the subregions illustrate how scores calculated over the complete CAS-CORDEX domain can hide underlying regional patterns. The spatial pattern correlation is lowest during winter for both RCMs, except for the ESB subregion where ALARO-0 shows a lower spatial correlation during summer. When considering the spatial correlation and the RMSE of the different subregions, both RCMs are closest to the CRU data over the WCA subregion. Based on the centred RMSE, the RCMs perform generally best during autumn, except for the REMO simulations in the subregions WSB and TIB. During the other seasons both RCMs simulate the temperature clearly worse in the northern part of the CAS-CORDEX domain (EEU, WSB, ESB). Both RCMs overestimate the normalized RSV, but ALARO-0 underestimates it in winter over the EEU subregion and in autumn over the WCA subregion. In general, both RCMs simulate the normalized standard deviation of the temperature well (RSV deviates less than 0.25 from 1) during autumn and winter. Additionally, REMO simulates the normalized standard deviation well during summer for the northern subregions. During spring the cold bias in the north is limited to -5 °C for the REMO model but not for ALARO-0. which is reflected in a higher RSV for the northern regions. High RSVs are also observed for ALARO-0 in summer over the complete domain (Fig. 4) and this is due to the underestimation of the cold temperatures in cold regions, while warm temperatures are overestimated in regions that are characterized by warmer temperatures (Fig. 3). is reflected in a normalized standard deviation that is higher than the one of REMO (Fig. 4). Comparing the metrics of the RCMs (Fig. 3, Fig. 4 and Table 2) shows that REMO is better in simulating the seasonal variability in temperature compared to ALARO-0, except for the autumn in all subregions and winter in the WSB and TIB subregion. On the other hand ALARO-0 often better captures spatial temperature patterns since the spatial pattern correlation is slightly higher than for REMO, except during winter and summer over the ESB and WCA subregions and spring and summer over the TIB subregion.

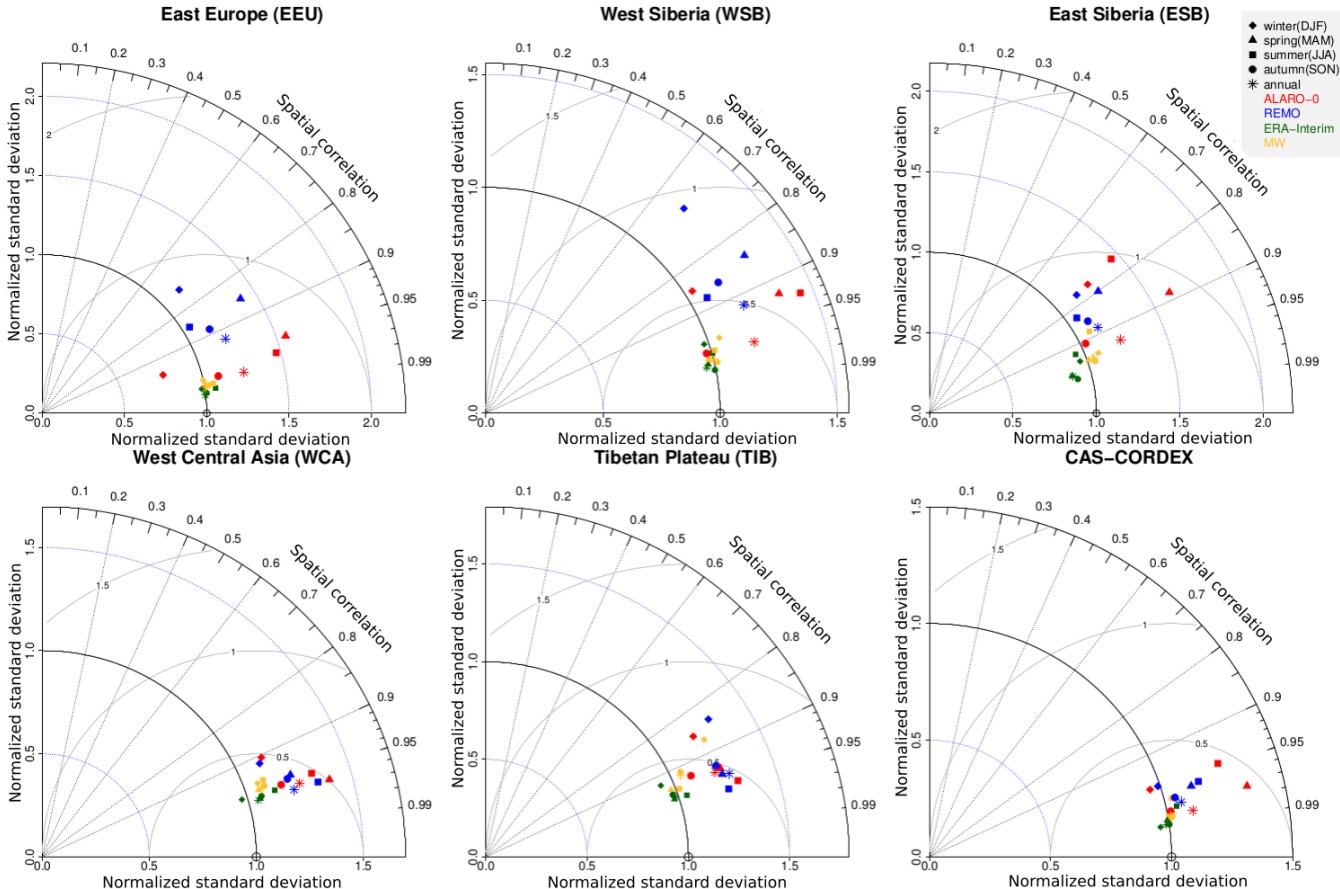

**Figure 4: Normalized Taylor diagram expressing the spatial performance of mean temperature for seasonal and annual means for both RCMs (ALARO-0 and REMO), the ERA-Interim reanalysis and MW observational data with respect to CRU for the five subregions and the complete CAS-CORDEX domain.**

Figure 5 shows the annual cycles of the mean, minimum and maximum temperature for both RCMs (ALARO-0 and REMO) compared to the ERA-Interim reanalysis, MW and CRU observational data over five subregions. From this figure, it can be seen that in the northern subregions EEU and WSB, there is on average a strong warm bias in December and January for ALARO-0, reaching a maximum of 4.1 °C and 5.8 °C respectively during December. REMO simulates winter temperatures (months 12, 1 and 2) within the uncertainty range of the observational datasets for WSB and underestimates the temperatures on average by 1.4 °C in January for EU. REMO simulates warm biases around 2 °C in December and January over ESB. On average there is no strong warm bias observed for ALARO-0 during the winter months in ESB (Table 2) due to the compensation effect of cold biases, both in time (Fig. 5) and space (Fig. 3). Furthermore, there is a remarkable cold bias observed for ALARO-0 during spring (months 3, 4 and 5) and June in the northern subregions EEU, WSB and ESB, reaching up to -7.3 °C over ESB during April. REMO is performing well during spring months over the northern subregions. From figure 5, it can be seen that the RCMs simulate the spatially averaged temperatures well during the autumn months (months 9, 10 and 11), since they are within the observational spread or deviate slightly from the observational spread (<1 °C). The

exceptions are the spatially averaged temperatures for ALARO-0 over WSB and WCA in November where the spatially
averaged temperature deviates 2 °C from CRU.

Compared to the northern subregions, ALARO-0 simulates the annual cycle better for the southern subregions WCA and TIB but slightly overestimates the amplitude of the annual temperature cycle. REMO simulates the mean temperature well over the WCA subregion with only a slight overestimation of the temperatures in July and August. In the mountainous area of TIB REMO underestimates the temperatures, except for January and December. The better results in spring, summer and autumn for ALARO-0 over the subregion TIB are due to spatial averaging of cold biases in the northern Himalayas and warm biases over the Taklamakan Desert and the opposite is true for REMO during winter (Fig. 3). This effect is reflected by the large MAE over this subregion during the mentioned seasons (Table 2).

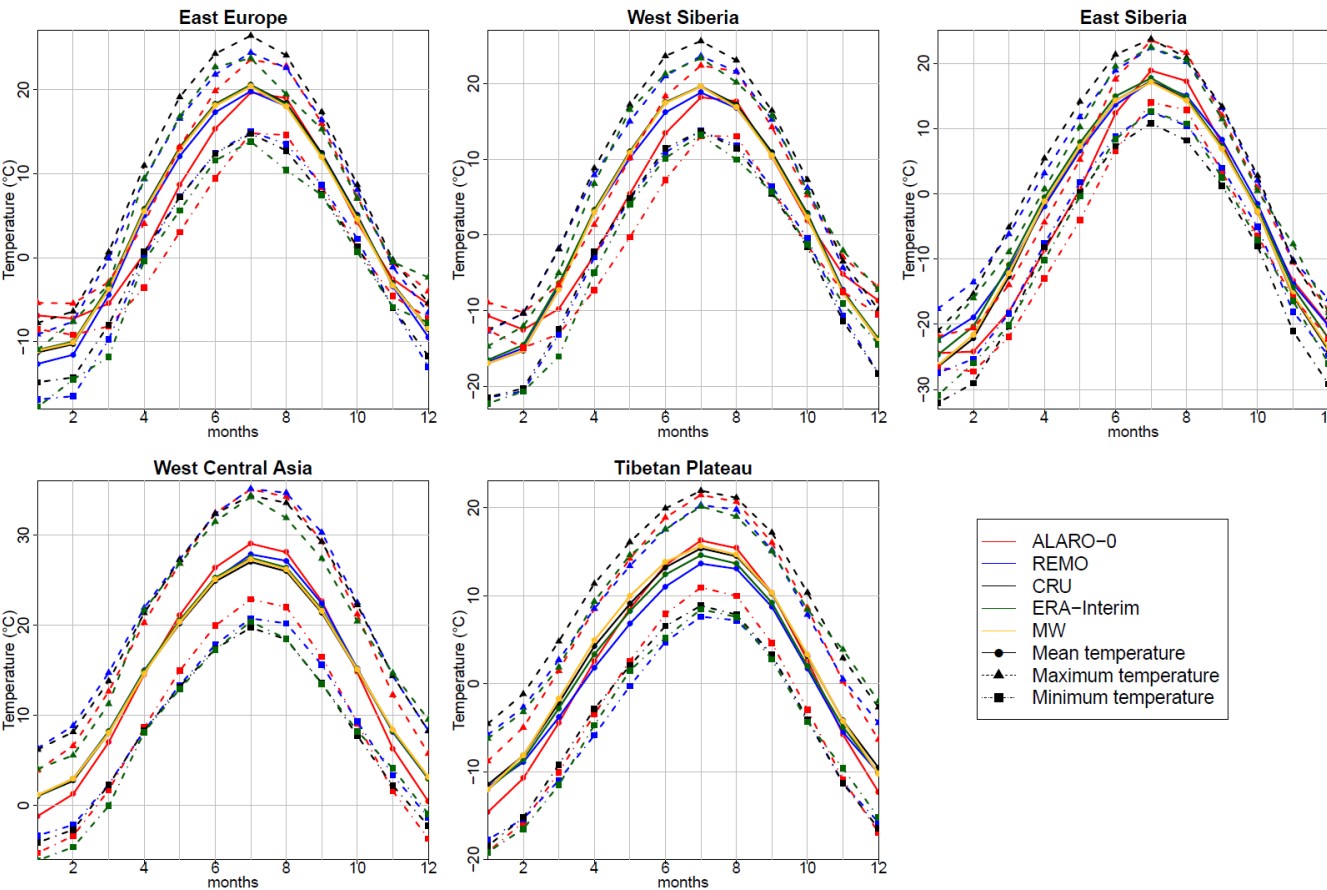

**Figure 5: Annual cycles of the mean, minimum and maximum temperature for both RCMs (ALARO-0 and REMO) compared to the ERA-Interim reanalysis, MW and CRU observational data over five subregions.**

**3.2 Diurnal temperature range**

Here, we first discuss the model performance of both RCMs for the minimum and maximum temperature, and then the diurnal
range taken as the difference between the two.

Similar to the mean temperature in Fig. 3, the modelled daily minimum temperature averaged over the different seasons and
years during 1980-2017 is compared with the observational CRU data in Fig. 6. Annual biases of the minimum temperature
over Russia in general vary between -3 °C and 3 °C for REMO and between -1 °C and 5 °C for ALARO-0, with a few
exceptions in the orographically complex regions, e.g. in the Stanovoy Range and Central Siberian Plateau where higher biases
are found.

Compared to ALARO-0, the REMO model shows larger warm biases over Mongolia during all seasons, except for summer.
The warm biases for REMO in the eastern part of the domain are most pronounced during winter reaching up to 15 °C.
ALARO-0 also shows equally large biases, but they cover the northern part of the domain. Moreover, strong cold biases are
present in the north during spring for both models, but they are more pronounced for the ALARO-0 model with biases up to -
10 °C in the north-eastern part of the domain. During the summer season the biases for the REMO model are limited between
-5 °C and 7 °C except for the Himalayan mountain range, while the ALARO-0 model output has, except for the Himalayas, a
cold bias up to -7 °C in the north-western part of Russia and warm bias up to 10 °C in the southern and eastern part of the
domain (Fig. 6). In autumn, both models have a warm bias over almost the entire domain, except for the cold biases in the
mountainous areas, the Arabian Peninsula, northern Iran, western Russia and for REMO also in the central northern part of the
domain. The increased minimum temperatures obtained with the RCMs indicate that they do not capture the coldest diurnal
temperatures.

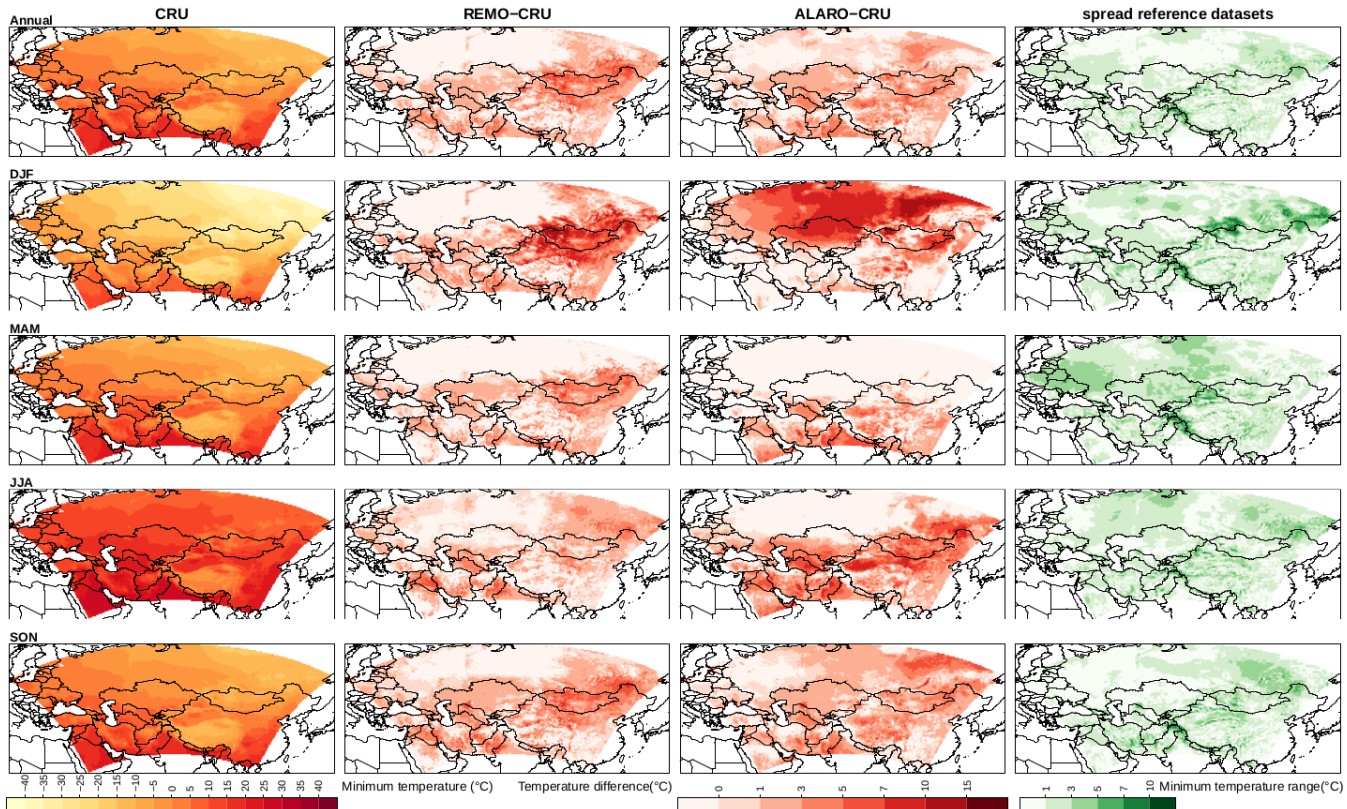

Figure 6: Left column: minimum air temperature (°C) at 2 m height over the CAS-CORDEX domain based on the observational CRU dataset for the 1980-2017 period on annual level and for winter (DJF), spring (MAM), summer (JJA) and autumn (SON). In the middle columns: difference in minimum temperature between models and CRU. Right column: the range in minimum temperature (°C) between the different reference datasets (CRU and ERA-Interim).

Table 3 shows the spatially averaged biases and MAE for minimum temperature during the 1980-2017 period of both RCMs and ERA-Interim compared to the minimum temperatures of CRU for the different seasons over the CAS-CORDEX domain and subregions. These scores confirm that the RCMs ALARO-0 and REMO are not able to reproduce the minimum temperature over the northern and eastern part of the domain during winter. During winter and spring, both models simulate minimum temperature best over the subregion WCA, while during summer and autumn they both perform best over the EEU region. REMO is able to simulate the minimum temperature accurately over the EEU and WSB subregions during summer since the errors are small (MAE < 1 °C). In general ALARO-0 has difficulties in simulating the minimum temperature correctly in any season and is only able to simulate the minimum temperature well over the EEU region during autumn.

The normalized Taylor diagrams in Fig. 7 confirm that, in general, the RCMs struggle to simulate the spatial pattern of minimum temperature well over the north-eastern part of the domain (ESB), while on an annual level ALARO-0 is able to simulate the spatial pattern well. The RCMs simulate the spatial pattern of minimum temperature well over the WCA region. Additionally, ALARO-0 produces minimum temperatures with a high spatial correlation to CRU over the EEU subregion, compared to REMO. At annual and seasonal scale, except for summer in WSB, ESB and TIB, ALARO-0 has a slightly better

spatial pattern correlation with the minimum temperatures of the CRU dataset than REMO. On the other hand, REMO has a better centered RMSE and spatial variability during summer, except for the WCA region.

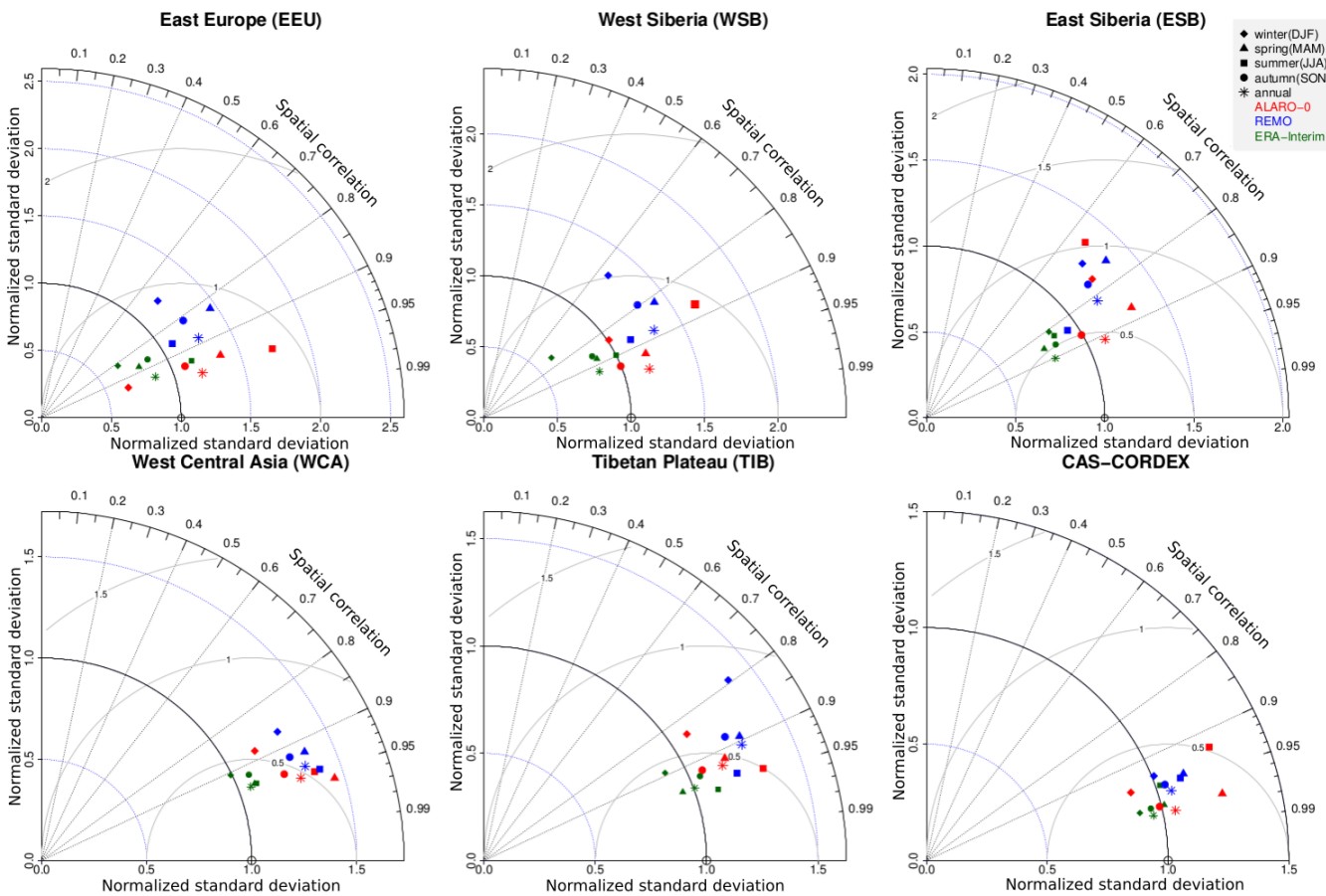

**Figure 7: Normalized Taylor diagram expressing the model spatial performance of the minimum temperature for seasonal and annual means for both RCMs (ALARO-0 and REMO) and ERA-Interim reanalysis with respect to CRU for the five subregions and the complete CAS-CORDEX domain.**

Biases in Fig. 8 and Table 4 show that for both RCMs a pronounced cold bias is present for maximum temperatures over the northern part of the domain at the annual scale and for all seasons, except for ALARO-0 in winter. During winter, ALARO-0 produces warm biases up to 5°C in the north and cold biases in the south-west and north-east up to -15 °C, while REMO has cold biases up to – 5 °C in the north-west and up to -15 °C on the Tibetan Plateau. ALARO-0 has the best performance over the EEU region during winter, while REMO has the best performance over the WCA subregion (Table 4 and Fig. 8). Both RCMs have a cold bias over a large area in the north during spring, which is very pronounced for the ALARO-0 model in the north-east (< -15 °C), while the biases remain limited to -7 °C for REMO (Fig. 8). The numbers in Table 4 confirm that during spring, the maximum temperature over the northern part of the domain deviates strongly (MAE > 2.50 °C) from CRU for both RCMs. During summer, these cold biases are reduced with biases up to -5 °C for REMO and -10 °C for ALARO-0. Both

models have warm and cold biases in the southern part of the domain during spring and summer. In autumn, the cold bias in the north is limited to -3 °C, but some stronger biases up to -7 °C appear in the north-east for the ALARO-0 model. The warm biases during autumn are limited to 5 °C and, excluding the Himalayas, the smallest range in biases is obtained for both RCMs during this season. Based on the MAE in Table 4, both RCMs show the best performance for maximum temperature during autumn, except for REMO over the TIB subregion and ALARO over the EEU region.

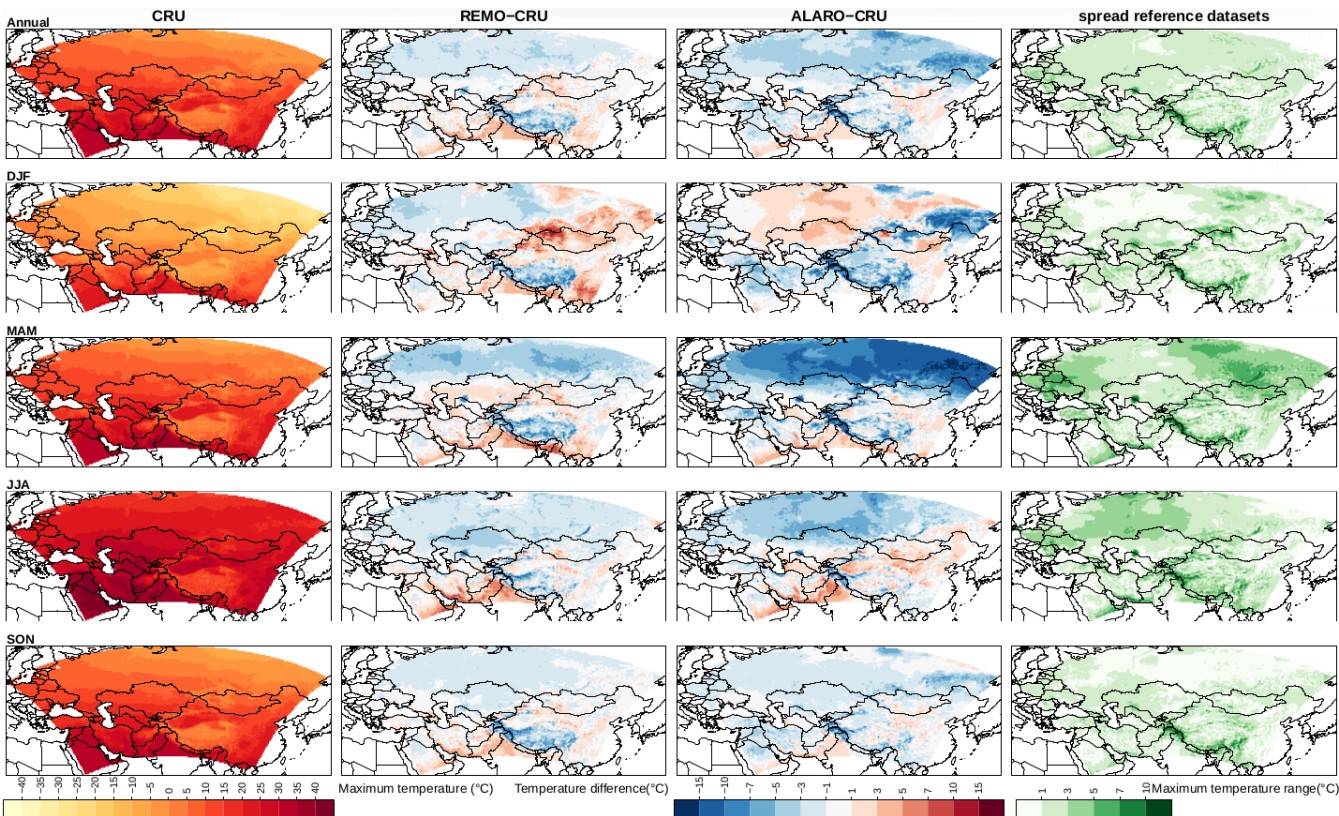

Figure 8: : Left column: maximum air temperature (°C) at 2 m height over the CAS-CORDEX domain based on the observational CRU dataset for the 1980-2017 period on annual level and for winter (DJF), spring (MAM), summer (JJA) and autumn (SON). In the middle columns: difference in maximum temperature between models and CRU. Right column: the range in maximum temperature (°C) between the different reference datasets (CRU and ERA-Interim).

Figure 9 shows that for all seasons, both RCMs have a high spatial correlation (> 90%) and a normalized RSV close to 1 for maximum temperature over the WCA subregion. This is the case as well for the TIB subregion, excluding the winter season. ALARO-0 has a high spatial correlation over the EEU subregion during all seasons and over the WSB subregion except for winter. Both RCMs struggle the most with reproducing the spatial patterns over the ESB subregion. ALARO-0 has higher spatial pattern correlations with CRU compared to REMO, except for autumn over the TIB subregion and winter over the ESB and WCA subregions.

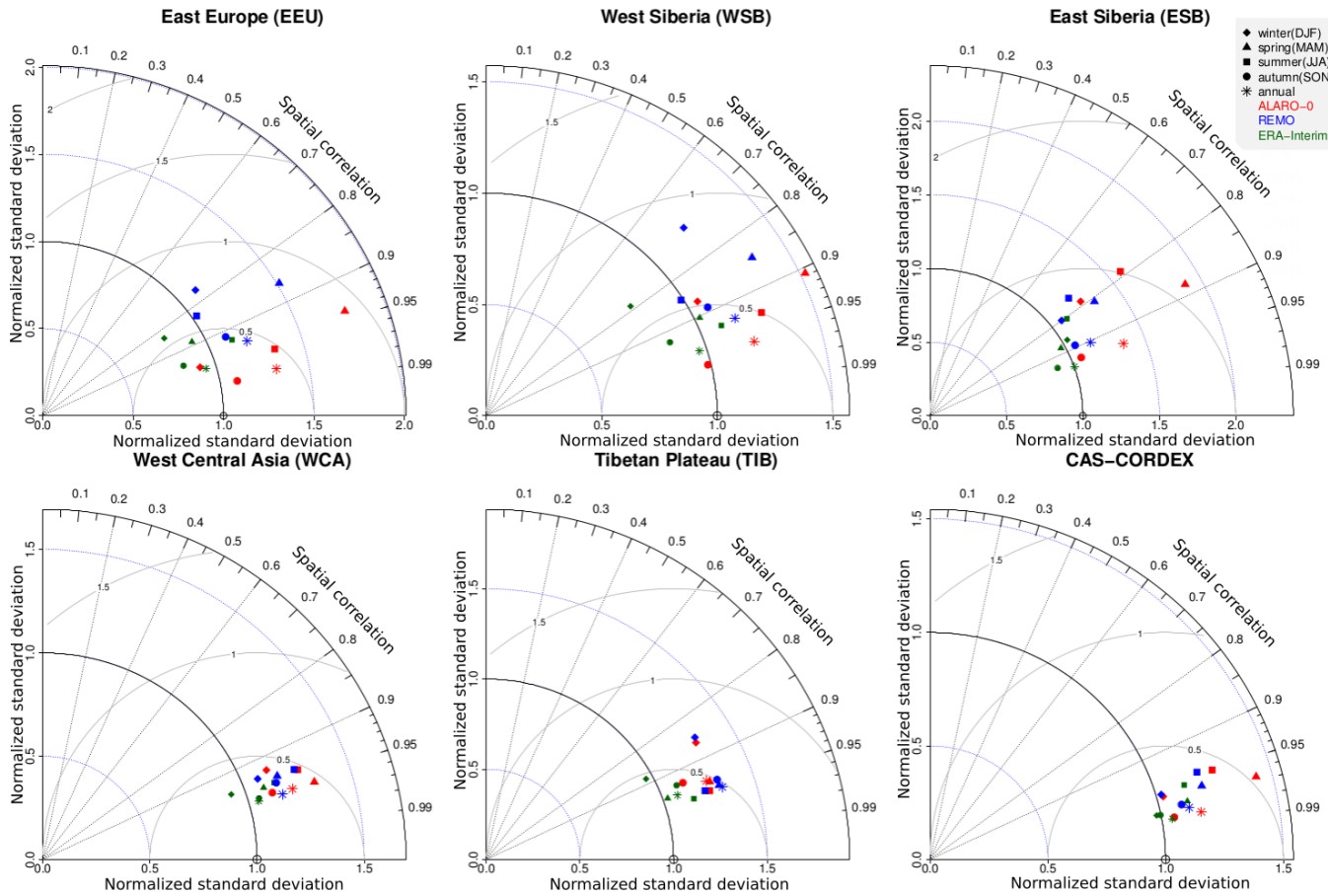

**Figure 9: Normalized Taylor diagram expressing the model spatial performance of the maximum temperature for seasonal and annual means for both RCMs (ALARO-0 and REMO) and ERA-Interim reanalysis with respect to CRU for the five subregions and the complete CAS-CORDEX domain.**

REMO has more often a normalized RSV value closer to 1 than ALARO-0, for the different subregions and seasons. Additionally, it is seen that both RCMs overestimate the normalized RSV of the maximum temperature for each subregion and season, except for winter in EEU and summer and autumn in WSB (Fig. 9). Based on Fig. 8 and 8, both RCMs simulate the maximum temperature best during autumn.

Finally comparing the minimum to the maximum temperature, it can be seen that minimum temperature (Table 3 and Fig. 5) shows warmer biases than the mean temperature (Table 2 and Fig. 3) over the different seasons, except for winter in EEU and WSB and spring in WSB and TIB. On the other hand, the maximum temperature (Table 4 and Fig. 7) shows colder biases compared with the mean temperature , except for winter and spring in WCA and summer in TIB. The increased minimum

temperatures obtained with the RCMs indicate that they do not capture the coldest diurnal temperatures. Nor do they capture the warmest diurnal temperatures because of the decreased maximum temperatures. From this it can be concluded that the daily temperature range is generally underestimated by both RCMs.

Moreover, the annual cycles in Fig. 5 show that both minimum and maximum temperatures are overestimated by ALARO-0 during winter in the northern part of the domain, while they are underestimated during spring. In summer the model is able to evolve to a more accurate balanced state and to simulate spatial averaged minimum temperatures as they are observed, resulting in better model results during autumn. REMO overestimates the minimum temperatures during the complete annual cycle for East Siberia, while the maximum temperatures in East Siberia are only overestimated during winter and underestimated during spring and summer. Both RCMs underestimate the maximum temperatures of CRU for the entire annual cycle over the Tibetan Plateau subregion. ALARO-0 overestimates minimum temperatures during the summer months, while REMO slightly overestimates winter and underestimates summer minimum temperatures.

### 3.3 Precipitation

Figure 10 and Table 5 present respectively the spatial pattern of precipitation and the spatially averaged precipitation over the 1980-2017 period for CRU over the full domain and subregions, as well as the relative biases and MAE of the RCMs with respect to CRU during the different seasons and on an annual level are presented as well.

At the annual level, the REMO model mainly shows a wet bias in the northern and the eastern part of the domain and a dry bias in the south-western part of the domain, while ALARO-0 has a wet bias in the north-west and south-east (Fig. 10). Furthermore, a strong wet bias is persistent over the annual cycle for both RCMs over the East Asian monsoon region, with a less notable wet bias during summer.

For both RCMs the overall bias for precipitation is wet, except for spring and summer in the WCA subregion and for ALARO-0, during summer in WSB, winter in WCA and spring and summer in the ESB subregion. Next to the wet biases in the monsoon region, both models show dry biases over the Taklamakan desert, except for winter.

During winter both RCMs have a strong wet bias in the eastern part of the domain (Fig. 10 and Table 5). This is partly due to the low observed precipitation quantities in several regions e.g. less than 5 mm per month in the Gobi desert region. Some of the largest relative biases can be found in relatively dry regions and therefore the absolute biases are presented in the supplementary material Fig. S4 and Table S2.

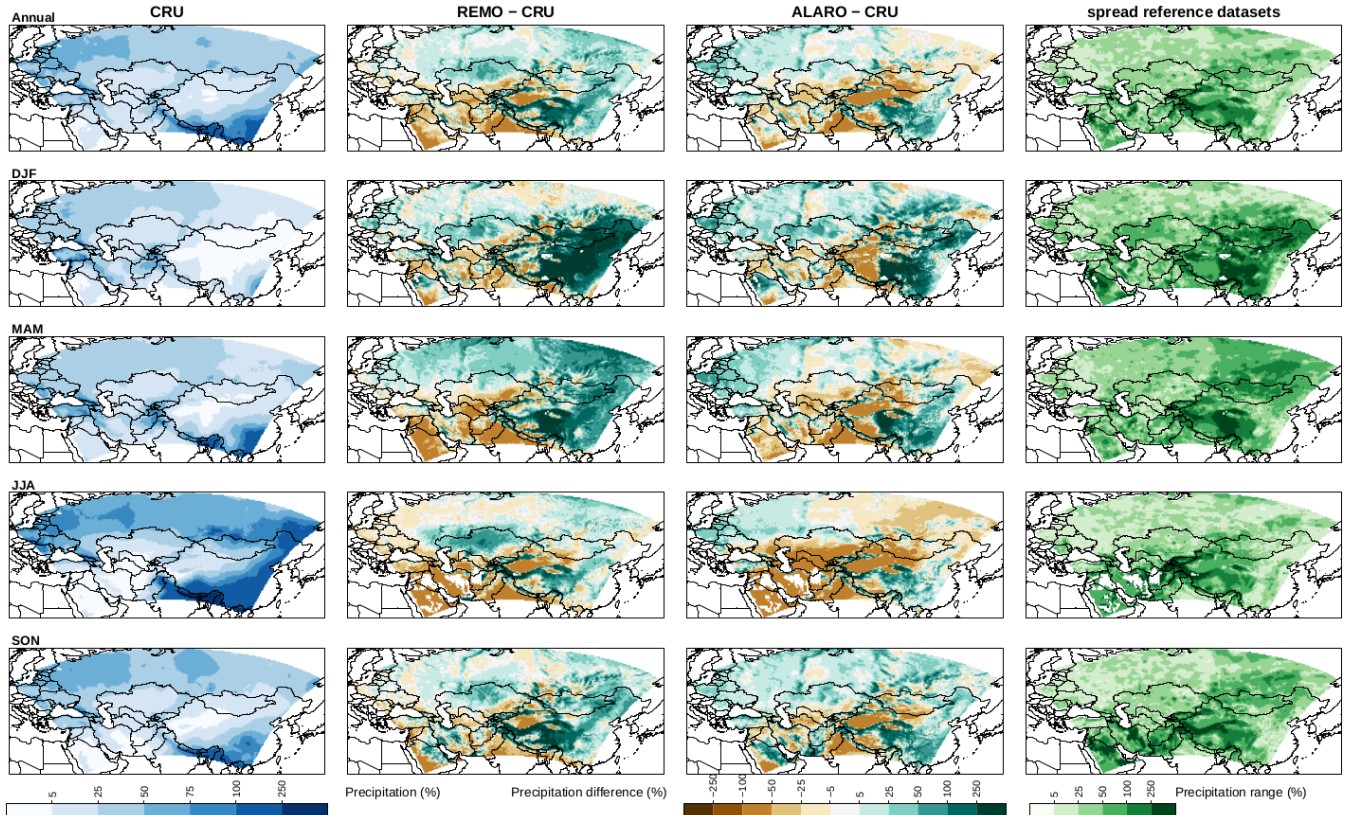

**Figure 10: Left column: mean monthly precipitation amounts (mm month-1) over the CAS-CORDEX domain based on the observational CRU dataset for the 1980-2017 period on annual level and for winter (DJF), spring (MAM), summer (JJA) and autumn (SON). In the middle columns: relative difference between the average annual and seasonal CRU precipitation and the precipitation simulated by the models (%). Right column: the range in precipitation (%) between the different reference datasets (CRU, MW, GPCC and ERA-Interim).**

In spring, a clear wet bias is present for REMO over the complete northern part of the domain and for ALARO-0 over the north-western part, while a strong dry bias is present in the south-western part of the domain for both RCMs (Fig. 10). The wet bias for REMO over ESB during spring is low in absolute values when compared to the subregion TIB (Fig. 12 and S1). In summer, both RCMs have a dry bias over the south-western part of the domain. The Taklamakan and Arabian deserts are located in these areas with a dry bias. In Fig. S4, the absolute dry biases over these regions are less pronounced (> -25 mm per month). The dry biases over the south-western part of the domain result in spatially averaged negative biases for precipitation over the WCA subregion in spring and summer for both RCMs (Table 5). Additionally, a smaller relative wet bias is present over the East Asian monsoon region during summer compared to the other seasons (Fig. 10). This is related to the higher precipitation rates in the south-eastern part of the domain during summer due to the East Asian Monsoon. Moreover, both RCMs have a dry bias in the northern part of the domain during summer (Fig. S4). For REMO this dry bias is situated in the north-western part of the domain and for ALARO-0, a stronger dry bias is situated in the north-eastern part of the domain,

resulting in a significant dry bias over the ESB subregion (Table 5). Furthermore, the dry bias over the Taklamakan desert is
470 more outspoken in summer. In autumn, both RCMs mainly produce a wet bias over the CAS-CORDEX domain, excluding
some areas with low precipitation rates that have dry biases e.g. the Taklamakan desert. In absolute numbers these dry biases
are limited (> -25 mm per month).

From Fig. 11 it can be deduced that REMO is only able to reliably reproduce the precipitation over the TIB subregion during
summer and not during the other seasons. Additionally, ALARO-0 better captures the spatial patterns since the correlations
are larger than those for REMO, except for the summer precipitation over WCA. Despite the substantial ALARO-0 biases
shown in Table 5 over most parts of the domain, the spatial patterns are thus well represented (Fig. 10 and 12). Both RCMs
overestimate the variability in precipitation for all seasons and subregions, except for REMO in summer over WCA (Fig. 11).
This excessive spatial variation is due to an overestimation of the precipitation in the wettest regions combined with an
underestimation in the driest regions (Fig. 10).

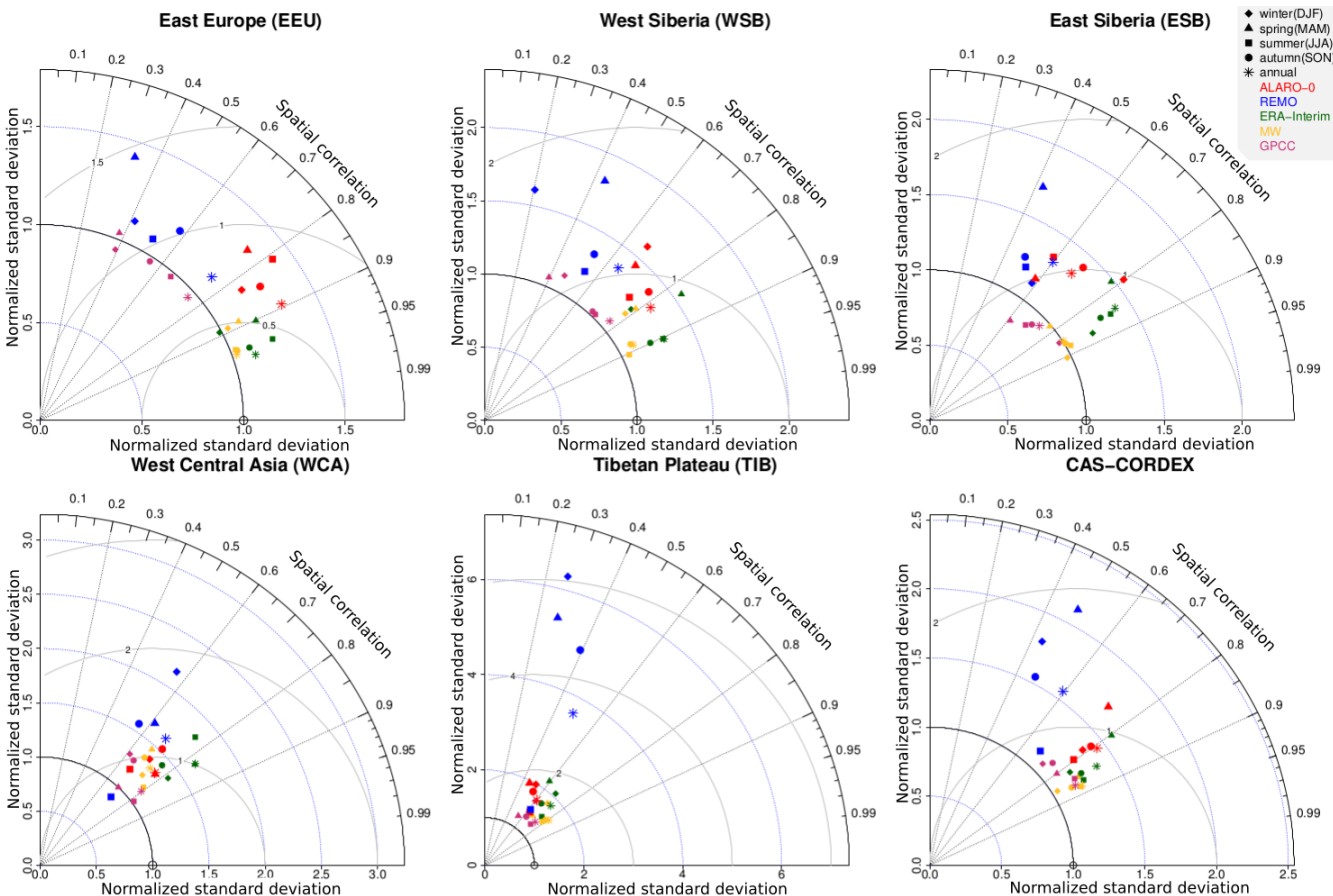

**Figure 11: Normalized Taylor diagram expressing the model performance of precipitation for seasonal and annual means for both RCMs (ALARO-0 and REMO), gridded observational datasets (MW, GPCC) and the ERA-Interim reanalysis data with respect to CRU for the five subregions and CAS-CORDEX domain.**

The annual cycles over the subregions show that ALARO-0 and REMO indeed mostly overestimate the precipitation values of CRU in the different subregions (Fig. 12). However, ALARO-0 does underestimate the precipitation slightly in May and June over West Siberia and in June and July over East Siberia. For the West Central Asian subregion, both RCMs underestimate the precipitation in spring and summer. REMO overestimates the precipitation slightly over the East Siberian subregion in March and June. As mentioned before, it is seen that REMO is unable to simulate the annual cycle of precipitation correctly over the subregion of the Tibetan Plateau. The precipitation rates are too high, except during the summer when the Asian Monsoon takes place. As seen in Fig. 12 and Table 5 the spatially averaged precipitation rate of REMO is slightly closer to the observations than ALARO-0 over the EEU subregion during winter and autumn. In addition, the annual cycle and MAE show that REMO better captures the precipitation over the ESB region than ALARO-0 during summer.

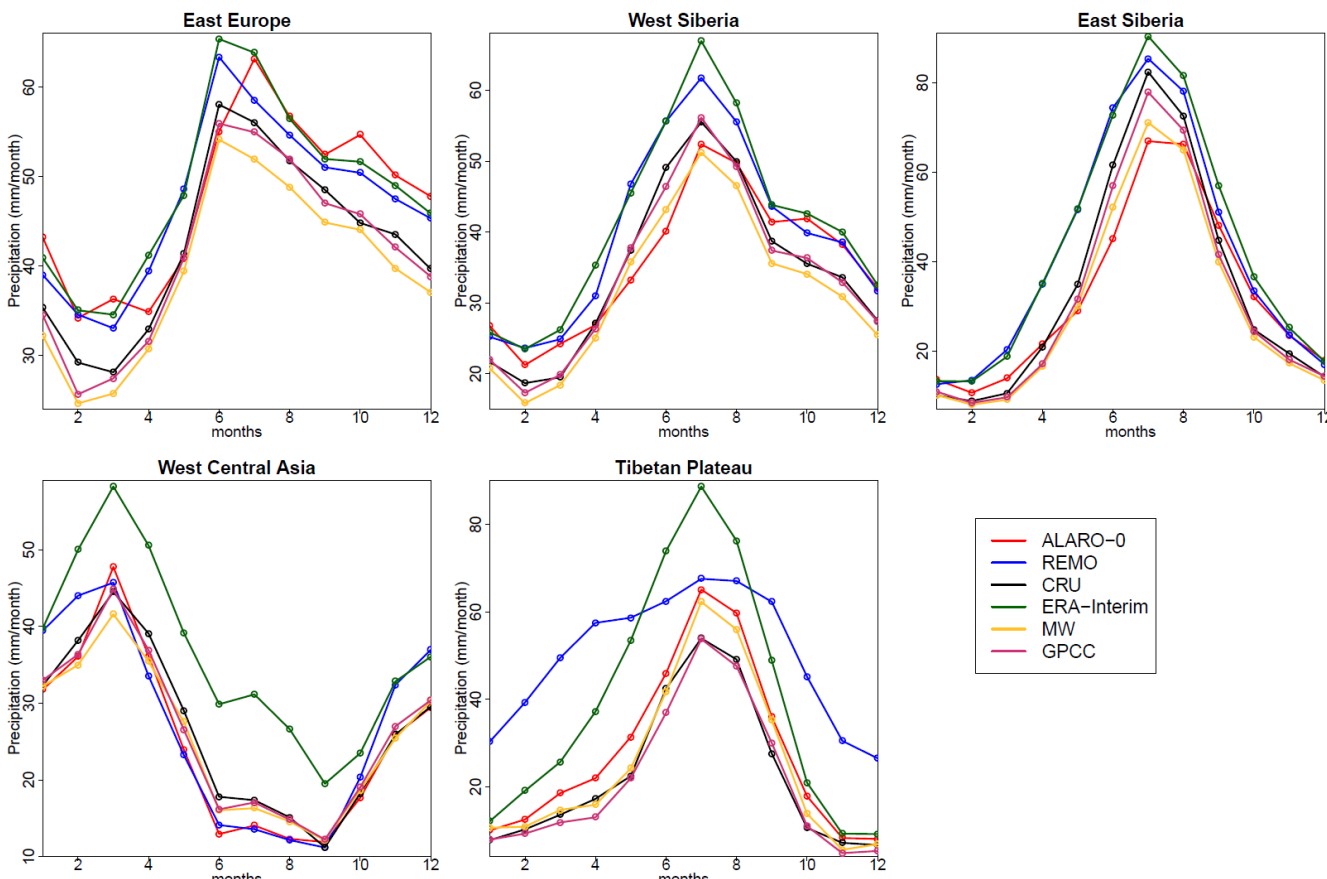

**Figure 12: Annual cycles of precipitation (mm/month) for both RCMs (ALARO-0 and REMO) compared to the ERA-Interim reanalysis, MW, GPCC and CRU observational data over five subregions.**

## 4 Discussion

### 4.1 Temperature

#### 4.1.1 Performance of ALARO-0 and REMO with respect to observational spread and other RCMs

When considering the temperature biases of the RCMs with respect to CRU, larger values are partly located in regions where the range of the different reference datasets is large (> 3 °C) (Fig. 3). Some regions where ALARO-0 and REMO show a bias over 3 °C also exhibit a spread of at least 3 °C between the reference datasets (CRU, MW and ERA-Interim), resulting in an insignificant bias when compared to the spread (Fig. 3 and Fig. S1). This is for example the case over mountainous regions such as the Himalayas and Stanovoy Range, which makes the evaluation of the models less reliable over these mountainous regions. The observational temperature spread is larger for the ESB subregion compared to EEEU and WSB, indicating there is a larger uncertainty for temperature evaluation over ESB. Significant observational uncertainties are typical over complex

orography, but this does not explain why there is a larger uncertainty over the complete ESB subregion. New et al. (1999)
mentioned that CRU contains colder temperatures in winter over Russia, which could explain this larger spread.

However, not all RCM biases are located within the spread of the reference datasets. For instance, the strong biases in the north-eastern part of the domain for ALARO-0 during winter and spring exceed the spread in temperatures between the different reference datasets, indicating that ALARO-0 is not able to simulate the temperatures accurately over this region (Fig. S1). Furthermore, the smaller biases for both RCMs over EEU (< 3 °C) are not situated within the small (< 1°) range of the
reference datasets (Fig. S1). The biases over WSB are not within the range of the reference datasets either, except for ALARO-0 during autumn. Figure S1 shows that for the majority of grid points the mean temperatures of ALARO-0 and REMO lie within the range of spread between the reference datasets during autumn. From this we conclude that both RCMs simulate temperatures fairly well in autumn. During winter and spring none of the RCMs are able to reproduce temperatures that can be completely explained by the observational uncertainty over a large part of the CAS-CORDEX domain, while this is also
the case for ALARO-0 during summer (Fig. 3 and Table 2).

When comparing the mean spatial biases and MAE for the 1980-2017 period (Table 2), it is seen that in most cases the differences between the observational datasets are smaller than the differences between the RCMs and CRU. However, the MAE and spatially averaged bias are smaller for both RCMs than for MW during autumn over the WSB subregion since both RCMs perform well over Kazakhstan with grid points with biases between -1 °C and 1°C. Moreover, REMO has lower MAE
values than MW over the ESB subregion during summer and autumn and over the WCA subregion during winter. ALARO-0 has lower MAE values than MW during autumn over the TIB subregion.

The Taylor diagrams of temperature (Fig. 4) show that the normalized standard deviation of ERA-Interim and MW differs less from CRU than the RCMs, except for REMO over the EEU and ESB subregions during summer and for ALARO-0 over ESB during autumn and WSB and TIB during winter. This smaller difference between the reference datasets implies that the
deviation in spatial variation of temperature between the RCMs and CRU cannot be completely explained by the observational uncertainty, meaning that the data of the RCMs deviates from the observations and can be improved. The spatial correlations between CRU and ERA-Interim or MW are lower than or close to those between CRU and the RCMs for the subregions WCA and TIB, which indicates that the RCMs are able to reproduce the spatial temperature patterns within the range of observational uncertainty, even though they slightly deviate from the spatial temperature patterns in the CRU data. It is seen that the observed
spatial patterns are less reliable during summer over East Siberia since the MW and ERA-Interim both show a lower spatial correlation (< 90 % for ERA-Interim) with CRU during summer compared to the other seasons. However, the lower spatial correlation of the RCMs during summer over East Siberia can only partly be explained by the observational uncertainty in spatial correlation of temperatures.

Similar to our findings, Ozturk et al. (2016) reported a lower spatial correlation during summer over the complete CAS-
CORDEX domain with RegCM4.3.5 at 0.50° horizontal resolution. Additionally, similarly high spatial correlations are obtained during the different seasons for ALARO-0 and REMO at 0.22° horizontal resolution when compared to the results of Ozturk et al. (2016). For summer temperatures, Russo et al. (2019) found that COSMO-CLM 5.0 produces a spatial pattern

with a cold temperature bias in the north and warm biases in the southern part of the domain except for some locations on the Tibetan Plateau, which are similar to ALARO-0.

In general both ALARO-0 and REMO produce biases within a similar order of magnitude as those obtained with other RCMs over the CAS-CORDEX region (Russo et al., 2019) and Central Asian subregions (Wang et al., 2020; Zhu et al., 2020). Zhu et al. (2020) conducted model runs with different land cover schemes in the WRF model over a smaller domain than CAS-CORDEX containing Kazakhstan, Uzbekistan, Kyrgyzstan, Turkmenistan and Tajikistan. None of their experiments produced biases over Kazakhstan as small as those of REMO in winter and at the annual level, while they obtained biases with different signs and similar magnitude in summer. However, it should be mentioned that they used the observational dataset from the Climate Prediction Center (CPC) which makes comparison difficult. ALARO-0 has biases with the same magnitude at the annual level as the WRF runs, but the absolute value of the biases is larger during winter and summer.

Similar to our findings, larger differences between temperatures of the reference datasets in the region of the Tibetan Plateau (Fig. 3) were also observed by Ozturk et al. (2012 and 2016) and Russo et al. (2019) and this is partly due to the fact that observational gridded data, such as MW and CRU, are based on measurements of meteorological stations in the valleys (New et al., 1999). The gridded observations are thus less reliable over the Himalayas and Tibetan Plateau, creating a larger observational uncertainty, and resulting in large biases of the RCMs that lie within the range of observational uncertainty in most of the grid points(Fig S1). Further, the amplification of the biases over the mountainous regions for the RCMs can be attributed to the used assumption of the lapse rate of 0.0064 K m-1 for the elevation correction (Kotlarski et al., 2014).

### 4.1.2 Spring and winter biases in northern subregions

In this section the temperature biases over snow covered areas during winter and spring will be explained. As mentioned in the previous sections, both RCMs have large temperature biases in the northern part of the domain that are not within the range of the reference datasets during winter and spring (Fig. S1). During winter, ALARO-0 simulates warm biases over the northern part of the domain and REMO simulates cold biases over the north-western part of the domain, while in spring they both show a cold bias over the north (Fig. 3 and 5).

Compared to the northern part in the CAS-CORDEX region, a similar warm bias during winter was found over Scandinavia in the EURO-CORDEX runs with ALARO-0 (Giot et al., 2016). Both regions have a similar climate which suggests that similar physical processes might be at the basis of these biases (Jacob et al., 2012; Remedio et al., 2019). The warm bias during winter and cold bias during spring in the north-eastern part of the domain for ALARO-0 are not due to a temporal shift in the annual cycle in the northern part of the domain, although there is a delay in warming temperatures during spring. A limited warm bias arises in the north during autumn, when the first snow cover appears over this region (Fig. 5). This bias increases when the snow covered region expands (not shown). ALARO-0 seems to underestimate cooling above snow cover during stable conditions. Mašek (2017) linked exceedingly warm temperatures above snow to the used single layer snow scheme (Douville et al., 1995). REMO is using a multi-layer snow scheme and does not encounter this problem.

A similar strong warm bias in the north, as found for ALARO-0 in winter, was also found by Ozturk et al. (2012) and Russo et al. (2019) for the RegCM and COSMO-CLM 5.0 models, respectively. Ozturk et al. (2012) related this warm bias to shortcomings in the simulation of snow, whereas Russo et al. (2019) found that changes in the snow scheme did not affect the simulation results significantly and did not reduce the warm bias in the north-east during winter. This shows that a more complex multi-layer snow scheme might not be enough to solve the warm bias for ALARO-0 during winter. Therefore, further

investigation should be done to see whether the warm bias in winter over the northern part of the domain is due to the inability of the current snow scheme to reproduce the heat conductivity of snow.

In spring, the warm temperature bias of the ALARO-0 simulation over the northern subregion evolves into a significant cold bias. This remarkable evolution is probably related to another issue connected to the snow scheme as we find a delay in the springtime melting of the snowpack (not shown). Additionally, ALARO-0 simulates exceedingly high pressure values over

585 the northern area (not shown). Further research is needed to clarify whether this overestimation of the Siberian High in the ALARO-0 simulations is related to the difficulties with the snow cover.

The cold bias for REMO during winter over the EEU subregion is likely due to the surface treatment of the model when there is snow (Pietikäinen et al., 2018). Pietikäinen et al. (2018) already reported that the thermodynamics of the snow layer plays an important role in the cold bias that appears over East Europe during the months when snow cover is present.

**4.2 Diurnal temperature range**

Similar to the mean temperature the observational spread for minimum and maximum temperature is larger in the orographically complex regions (Fig. 6 and Fig. 8). ALARO-0 and REMO are not able to reproduce the minimum and maximum temperature since they produce biases that are outside this significant observational range (e.g. the range for maximum temperature is 5 °C to 7 °C in the north-eastern part of the domain in spring) (Fig. S2 and S3). However, during

summer REMO simulates minimum and maximum temperatures within the observational range over western Russia. The MAE of REMO for minimum and maximum temperatures is acceptable during summer over East Europe and the West Siberia subregions since the MAE between ERA-Interim and CRU is larger than the MAE between REMO and CRU (Table 3 and 4). Moreover, the MAE of REMO for maximum temperature is lower than the MAE of ERA-Interim over the WCA domain, indicating that REMO is able to produce maximum temperatures over this subregion within the range of the reference datasets.

Both RCMs generally produce a smaller daily temperature range, resulting in biases that are generally warmer for the minimum temperature and colder for the maximum temperature, when compared to those of the mean temperature (Fig. 3, 6, 8 and Tables 2, 3 and 4). The smaller daily temperature range causes a stronger warm bias in winter for the minimum temperature and a stronger cold bias for maximum temperature in spring, which is notably visible in the northern part of the domain for the ALARO-0 model (Fig. 3, 6, 8 and Table 2, 3 and 4). Additionally, it is seen that the cold bias in the north during spring for

the ALARO-0 model is weaker for the minimum temperature than for the mean temperature, while the REMO model shows warmer biases over Mongolia during winter and spring for minimum temperature and colder biases in maximum temperature in the north during spring when compared to the mean temperature. Moreover, the smaller daily temperature range causes

larger MAE scores for minimum temperature during winter and for maximum temperature during spring, except for ALARO-0 over the WCA and TIB subregion (Table 3 and 4). This indicates that minimum temperatures are less accurately simulated by both RCMs compared to temperature during winter, while maximum temperatures are simulated less accurately during spring.

The underestimation of the diurnal range is similar to the findings over other regions (Laprise et al., 2003; Kyselý and Plavcová 2012) and was also observed over the CAS-CORDEX domain by Russo et al. (2019). Their RCM produced smaller diurnal ranges compared to different observational datasets. In particular ALARO-0 shows a smaller range in the diurnal cycle of temperatures due to very high minimum temperatures (Fig. 5) and this could be due to the inability of the model to simulate temperatures correctly over snow cover during stable conditions (Mašek, 2017).

Although the magnitude of the biases is different for mean, minimum and maximum temperature, similar spatial patterns are found in the biases of both RCMs over the different seasons and for the annual mean (Fig. 3, 6 and 8). This means that these variables are spatially highly correlated with each other in both models and observations. Additionally, both minimum and maximum temperatures have a similar temporal pattern as the mean temperature (Fig. 5).

The metrics in Fig. 7 and 9 show that spatial pattern correlations of ERA-Interim deviate more from CRU for minimum and maximum temperature compared to mean temperature (Fig. 4). This larger uncertainty makes it harder to draw sound conclusions from the lower spatial pattern correlations of ALARO-0 and REMO.

The evaluation of temperature and its diurnal cycle shows that a bias adjustment is essential before the climate data is applied in impact modelling. However, REMO simulates mean and maximum temperatures well over the West Central Asia subregion when the observational range is taken into account.

### 4.3 Precipitation

Compared to the RegCM4.3.5 model (Ozturk et al., 2016) ALARO-0 has lower RMSEs over all seasons and REMO has higher RMSEs, excluding summer (Fig. 11). The spatial correlations between CRU and REMO are similar to the values obtained with RegCM4.3.5, except for winter where REMO has a higher spatial correlation (Fig. 11). ALARO-0 obtains higher values for the spatial correlations and they are close to those of the other observational datasets.

For the majority of the grid points, the precipitation of ALARO-0 and REMO is situated within the spread of the different gridded datasets for all seasons (Fig. S5). However, there are some subregions where the precipitation of ALARO-0 and/or REMO exceeds the observational spread during one or more seasons. For example, both RCMs show slightly lower precipitation amounts in summer over West Central Asia compared to the different reference datasets (Fig. 12 and S5). Additionally, the overestimation in precipitation by both RCMs in the East Asian monsoon region exceeds the observational spread, especially in winter and spring for REMO and in spring and autumn for ALARO-0, indicating that the models do not completely capture the East Asian monsoon system. Moreover, the ALARO-0 model overestimates the precipitation significantly over the East European subregion during all seasons when compared to the spread of the reference datasets (Fig S5 and Fig. 12).

Ozturk et al. (2012) and Russo et al. (2019) obtained similar seasonal patterns in precipitation, with their model simulations at a horizontal resolution of 0.50° and 0.22°, respectively. For example, an extreme excess of precipitation was simulated over the East Asian monsoon region, with a smaller relative wet bias in summer. Additionally, they obtained a dry bias in summer over the western part of the domain which is similar for REMO, while ALARO-0 shows only a dry bias in the south-western part of the domain. Moreover, ALARO-0 produces a dry bias over the north-eastern part of the domain during summer, while this is not the case for the other RCMs (REMO, COSMO-CLM 5.0 and RegCM4.0) (Ozturk et al., 2012; Russo et al., 2019). The underestimation in precipitation by ALARO-0 during spring and summer in the north-eastern part of the domain might be related to the Siberian High that remains too strong (not shown).

Table 5 and Fig. 12 show that on average, CRU contains higher precipitation amounts than the two other observational datasets, MW and GPCC. As mentioned before, it is known that the MW and GPCC datasets generally underestimate the seasonal precipitation over Central Asia, especially during spring for the central part of the CAS-CORDEX domain (Hu et al., 2018). The overestimation of the annual precipitation by the RCMs over the Himalaya, Altay, Tian Shan and Kunlun Mountains is partly due to the fact that gridded observational datasets CRU, MW and GPCC underestimate the precipitation over these mountainous regions. It is known that the accuracy of gridded precipitation datasets decreases with elevation, especially over an altitude of 1500 m (Zhu et al., 2015). By contrast, ERA-Interim generally overestimates the precipitation, particularly over mountainous regions (Sun et al., 2018). Moreover, a similar pattern of an underestimation by gridded observational datasets and overestimation by reanalysis data is present over the Tibetan Plateau (Sun et al., 2018), causing larger biases (Fig. 10 and Fig. 12). The discrepancy between the observational gridded datasets and the ERA-Interim reanalysis data (Fig. 10 and Fig. 12) explains why the strong wet biases of the RCMs compared to CRU over the mountainous areas and Tibetan Plateau are not significant (Fig. S5). The pronounced difference between the observational and reanalysis datasets makes it difficult to draw sound conclusions over these regions.

Even when taking into account the large spread between the reference datasets, REMO is not able to reproduce the annual cycle of precipitation over the Asian monsoon region. Remedio et al. (2019) also found a wet bias for REMO at the annual level over the subtropical region where the Asian monsoon takes place.

In the north, the precipitation amounts of REMO bear more resemblance to those of ERA-Interim and COSMO-CLM 5.0 described by Russo et al. (2019) (Fig. S8). This similarity is probably due to the fact that they all use a convection scheme that is based on Tiedtke (1989) (Table S1; www.ecmwf.int, consulted on 07/07/2020), while ALARO-0 uses the 3MT cloud microphysics scheme.

It can be concluded that for the different subregions and seasons, REMO and ALARO-0 simulated precipitation mostly within the range of the observational spread, although it should be mentioned that the observational uncertainty is large. MW, GPCC and ERA-Interim deviate more from CRU than was the case for temperature, resulting in a larger observational uncertainty for precipitation. Russo et al. (2019) showed additionally that the influence of observational datasets on the RSV is larger for precipitation than for temperature. Moreover, both models are worse in simulating the spatial correlation of precipitation (Fig. 11) compared to the mean, minimum and maximum temperature (Fig. 4, 7 and 9). This lower correlation is due to the fact that

precipitation is less systematically affected by land cover and topography compared to temperature (Kotlarski et al., 2014). Furthermore, the uncertainty range and error in the observational products should be reduced in the future to improve the evaluation of precipitation (Russo et al., 2019).

## 5 Conclusion

The evaluation over the CAS-CORDEX domain of ALARO-0 and REMO, run at 0.22° resolution, showed that in general both
RCMs reproduced realistic spatial patterns for temperature since there is a high spatial correlation with observational data. Additionally, the values of spatial variation for mean temperature of both RCMs correspond closely to the values obtained with other reference datasets. When evaluating the modelled precipitation, poorer scores were obtained for these metrics but the spread between the different observational datasets is also larger for precipitation as compared to temperature.

Both RCMs performed best during autumn for temperature and precipitation, showing biases within the range of the
observational uncertainty for the majority of the CAS-CORDEX domain. Nevertheless, there are significant biases in several regions during several seasons e.g. a warm bias in the north during winter and a wet bias over the Asian monsoon region. For ALARO-0 the northern part of the CAS-CORDEX domain is subject to significant positive temperature biases in winter, followed by large negative temperature biases in spring. This behavior is probably linked to limitations of the used snow scheme. REMO produced excessive precipitation amounts over the Tibitian Plateau subregion during all seasons and
incorrectly simulated the annual cycle of the East Asian Monsoon system. In general, REMO was better than ALARO-0 in reproducing the seasonal mean temperatures, except during autumn, whereas ALARO-0 estimated the precipitation well.

Additionally, the evaluation of minimum and maximum temperatures showed that the RCMs underestimate the daily temperature range. This illustrates the added value of taking more evaluation variables into account than only the commonly evaluated variables mean temperature and precipitation.

We conclude that REMO and ALARO-0 can be used for climate modelling over Central Asia e.g. for precipitation and temperature over West Central Asia. However, the deficiencies of both models over Central-Asia described in this evaluation study should be kept in mind. Climate data produced by both RCMs can only be used for impact studies if a suitable bias adjustment is applied for those subregions where the RCMs perform less well e.g. temperature over East Siberia and precipitation over the Tibetan Plateau.

**Code availability**

The R code used for the analysis is available through: http://doi.org/10.5281/zenodo.3659716 (Top et al., 2020).
For the code of the ALARO-0 model we refer to the Code availability section in Termonia et al. (2018). More information about the REMO model is available on request by contacting the Climate Service Center Germany (contact@remo-rcm.de).

**Data availability**

The climate data produced by ALARO-0 and REMO2015 have been uploaded to the ESGF data nodes (website: http://esgf.llnl.gov/). In order to obtain the data, one of the nodes must be chosen. Thereafter, click on 'CORDEX' or search for 'CORDEX' and then select the domain 'CAS-22' and the RCM model in the left column. The exact identifiers can be found in Table S3 of the supplementary material.

The CRU data is available through (http://www.cru.uea.ac.uk). The MW data is freely available at:
http://climate.geog.udel.edu/~climate/html_pages/download.html and NetCDF files can be found here: https://www.esrl.noaa.gov/psd/data/gridded/data.UDel_AirT_Precip.html: air.mon.mean.v501.nc and precip.mon.total.v501.nc. The GPCC data can be accessed through: doi: 10.5676/DWD_GPCC/FD_M_V2018_025.

**Author contribution**

Modelling and performing simulations: C.S., D.C.L., D.T.R., K.L., K.A, R.R.A. ;Post-processing: D.C.L., D.T.R., K.L., K.A
R.R.A. ;Visualization: K.L., T.S. ;Writing - original draft: T.S. ;Writing - review & editing: A.S., B.L., C.S., D.C.L., D.M.P., D.T.R., G.N., G.A., H.R., K.A, K.L., R.R.A., S.A., T.P., T.S., V.D.V.H., V.S.B., Z.V. ;Supervision: C.S., D.M.P., T.P. ;Funding acquisition: A.S., B.L., D.M.P., G.A., K.L., T.P.

**Acknowledgement**

The AFTER project is granted by the ERA.Net RUS Plus Initiative, ID 166. HZG-GERICS received funding from the Federal
Ministry of Education and Research (BMBF). Ghent University and VITO received funding from the Research Foundation - Flanders (FWO), grant G0H6117N. NIERSC received funding from the Russian Foundation for Basic Research (RFBR) - grant № 18-55-76004. LEGMC received funding from the State Education Development Agency (SEDA) and ISTE received funding from the scientific and technological research council of Turkey (TUBITAK agreement nr: 2017O394).

The computational resources and services for the ALARO-0 regional climate simulations were provided by the Flemish
Supercomputer Center (VSC), funded by the Research Foundation - Flanders (FWO) and the Flemish Government department EWI. The CORDEX-CORE REMO simulations were performed under the GERICS/HZG share at the German Climate Computing Centre (DKRZ).

We would like to thank Ján Mašek for his insights on the warm bias above snow cover.

We are grateful for the remarks and suggestions of the anonymous reviewers and topical editor, which improved the manuscript
substantially.

**Competing interests**

The authors declare that they have no conflict of interest.

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

**Table 1: Overview of the used reference datasets.**

| Dataset | Short name | Type | Resolution | Used variables | Frequency | Temporal coverage | Domain |
|---------|-----------|------|-----------|----------------|-----------|-------------------|--------|
| gridded Climatic Research Unit TS dataset (version 4.02) | CRU | gridded station data | 0.50° | 2 m mean air temperature, 2 m maximum air temperature, 2 m minimum air temperature, precipitation | monthly | 1901 - 2018 | global land mass (excluding Antarctica) |
| Matsuura and Willmot, University of Delaware (version 5.01) | MW | gridded station data | 0.50° | 2 m mean air temperature, precipitation | monthly | 1900 - 2017 | global land mass |
| Global Precipitation Climatology Centre gridded dataset (version 2018) | GPCC | gridded station data | 0.50° or 0.25° | precipitation | monthly | 1891 - 2016 | global land mass (excluding Antarctica) |
| ERA-Interim | ERA-Interim | reanalysis data | 0.70° | 2 m mean air temperature, precipitation | monthly | 1979 - 2017 | global |


**Table 2: Climatological mean CRU temperature (°C) for the 1980-2017 period over the CAS-CORDEX domain and subdomains, biases (°C) and MAE (°C) of the RCMs (REMO and ALARO-0) and the other reference datasets (ERA-Interim and MW) against CRU.**

| | EEU | | | | | WSB | | | | | ESB | | | | |
|---|---|---|---|---|---|---|---|---|---|---|---|---|---|---|---|
| | DJF | MAM | JJA | SON | Annual | DJF | MAM | JJA | SON | Annual | DJF | MAM | JJA | SON | Annual |
| CRU | -10.01 | 5.09 | 19.08 | 4.77 | 4.8 | -15.44 | 2.39 | 18.16 | 2.13 | 1.89 | -24.29 | -2.34 | 15.35 | -3.66 | -3.64 |
| REMO - CRU | -1.53 | -1.42 | -1.06 | -0.46 | -1.11 | -0.40 | -0.94 | -1.22 | -0.52 | -0.77 | 3.11 | -0.42 | -0.13 | 0.90 | 0.86 |
| MAE REMO CRU | 1.85 | 2.06 | 1.11 | 0.72 | 1.31 | 1.94 | 1.95 | 1.33 | 0.86 | 1.28 | 3.40 | 1.78 | 0.71 | 1.25 | 1.40 |
| ALARO - CRU | 3.27 | -4.35 | -1.56 | -0.44 | -0.79 | 4.57 | -5.26 | -2.16 | -0.14 | -0.77 | 1.26 | -6.90 | 0.63 | 0.57 | -1.12 |
| MAE ALARO CRU | 3.28 | 4.36 | 2.32 | 0.66 | 1.22 | 4.87 | 5.31 | 2.79 | 0.51 | 1.18 | 3.97 | 6.99 | 2.09 | 1.45 | 1.65 |
| ERA-Interim - CRU | 0.24 | -0.10 | -0.15 | -0.23 | -0.06 | 0.41 | 0.06 | -0.19 | -0.29 | -0.01 | 1.68 | 1.04 | 0.49 | 0.41 | 0.91 |
| MAE ERA-Interim CRU | 0.41 | 0.3 | 0.43 | 0.31 | 0.25 | 0.85 | 0.53 | 0.62 | 0.49 | 0.43 | 1.94 | 1.25 | 0.83 | 0.80 | 1.10 |
| MW - CRU | 0.01 | -0.42 | -0.39 | -0.49 | -0.32 | -0.20 | -0.46 | -0.36 | -0.65 | -0.42 | 0.08 | 0.12 | -0.14 | -0.26 | -0.05 |
| MAE MW CRU | 0.46 | 0.52 | 0.56 | 0.46 | 0.46 | 0.88 | 0.78 | 0.73 | 0.88 | 0.88 | 1.55 | 0.96 | 0.94 | 1.55 | 1.55 |

| | WCA | | | | | TIB | | | | | CAS-CORDEX | | | | |
|---|---|---|---|---|---|---|---|---|---|---|---|---|---|---|---|
| | DJF | MAM | JJA | SON | Annual | DJF | MAM | JJA | SON | Annual | DJF | MAM | JJA | SON | Annual |
| CRU | 2.25 | 14.34 | 25.98 | 14.89 | 14.42 | -9.79 | 3.69 | 14.36 | 3.05 | 2.88 | -9.35 | 5.87 | 19.23 | 5.72 | 5.44 |
| REMO - CRU | -0.11 | -0.05 | 0.57 | 0.22 | 0.16 | -0.07 | -1.49 | -1.16 | -0.90 | -0.90 | 0.48 | -0.56 | -0.33 | 0.01 | -0.11 |
| MAE REMO CRU | 1.48 | 1.64 | 2.03 | 1.46 | 1.47 | 3.31 | 2.76 | 2.50 | 2.37 | 2.59 | 2.33 | 1.82 | 1.34 | 1.20 | 1.43 |
| ALARO - CRU | -2.13 | -0.38 | 1.70 | -0.41 | -0.29 | -2.57 | -1.04 | 1.29 | -0.28 | -0.63 | 0.83 | -3.19 | 0.02 | -0.03 | -0.60 |
| MAE ALARO CRU | 2.77 | 2.38 | 2.79 | 1.59 | 1.81 | 3.24 | 2.92 | 3.25 | 1.94 | 2.32 | 3.16 | 4.20 | 2.42 | 1.24 | 1.56 |
| ERA-Interim - CRU | -0.03 | 0.11 | 0.32 | 0.07 | 0.12 | -0.46 | -0.62 | -0.60 | -0.82 | -0.62 | 0.42 | 0.21 | 0.16 | -0.02 | 0.19 |
| MAE ERA-Interim CRU | 1.26 | 1.27 | 1.58 | 1.21 | 1.17 | 1.77 | 1.95 | 2.02 | 1.80 | 1.77 | 1.16 | 1.02 | 0.98 | 0.85 | 0.87 |
| MW - CRU | -0.09 | -0.23 | 0.08 | -0.09 | -0.08 | -0.46 | 0.75 | 0.56 | 0.14 | 0.26 | -0.41 | -0.19 | -0.09 | -0.43 | -0.28 |
| MAE MW CRU | 1.53 | 1.38 | 1.48 | 1.53 | 1.53 | 2.78 | 2.22 | 2.12 | 2.78 | 2.78 | 1.32 | 1.10 | 1.07 | 1.32 | 1.32 |


**Table 3: Spatial average over the CAS-CORDEX domain and subdomains of climatological mean CRU minimum temperature (°C) for the 1980-2017 period, and biases (°C) and MAE (°C) against CRU for REMO, ALARO-0 and ERA-Interim.**

| | EEU | | | | | WSB | | | | | ESB | | | | |
|---|---|---|---|---|---|---|---|---|---|---|---|---|---|---|---|
| | DJF | MAM | JJA | SON | Annual | DJF | MAM | JJA | SON | Annual | DJF | MAM | JJA | SON | Annual |
| CRU | -13.56 | -0.03 | 13.3 | 0.99 | 0.24 | -20 | -3.26 | 12.24 | -2.48 | -3.3 | -30.12 | -9.47 | 8.78 | -9.27 | -9.93 |
| REMO - CRU | -2.21 | -1.29 | 0.05 | 0.35 | -0.77 | -0.67 | -1.16 | -0.32 | 0.47 | -0.42 | 3.64 | 0.87 | 1.77 | 2.48 | 2.18 |
| MAE REMO CRU | 2.73 | 2.17 | 0.56 | 0.90 | 1.42 | 2.38 | 2.24 | 0.82 | 1.37 | 1.49 | 4.13 | 2.40 | 1.86 | 2.66 | 2.49 |
| ALARO - CRU | 5.10 | -3.21 | -0.79 | 0.45 | 0.37 | 7.15 | -4.02 | -1.51 | 1.26 | 0.69 | 4.74 | -3.92 | 2.18 | 2.79 | 1.43 |
| MAE ALARO CRU | 5.11 | 3.26 | 2.45 | 0.67 | 0.88 | 7.24 | 4.07 | 2.78 | 1.36 | 0.97 | 5.35 | 4.10 | 3.00 | 2.86 | 1.73 |
| ERA-Interim - CRU | 0.24 | -2.21 | -1.38 | -0.23 | -0.90 | 0.81 | -2.53 | -1.19 | 0.86 | -0.52 | 2.32 | -0.83 | 1.85 | 2.18 | 1.38 |
| MAE ERA-Interim CRU | 1.35 | 2.24 | 1.50 | 0.56 | 1.00 | 1.60 | 2.60 | 1.42 | 0.96 | 0.88 | 2.73 | 1.38 | 2.02 | 2.25 | 1.62 |

| | WCA | | | | | TIB | | | | | CAS-CORDEX | | | | |
|---|---|---|---|---|---|---|---|---|---|---|---|---|---|---|---|
| | DJF | MAM | JJA | SON | Annual | DJF | MAM | JJA | SON | Annual | DJF | MAM | JJA | SON | Annual |
| CRU | -3.02 | 7.93 | 18.54 | 7.84 | 7.87 | -16.76 | -3.35 | 7.76 | -4.03 | -4.04 | -14.43 | -0.22 | 13.18 | 0.40 | -0.20 |
| REMO - CRU | 0.68 | 0.00 | 1.07 | 1.57 | 0.83 | 1.00 | -1.70 | -0.61 | 0.55 | -0.19 | 0.77 | -0.25 | 0.60 | 1.09 | 0.55 |
| MAE REMO CRU | 2.4 | 2.10 | 2.56 | 2.60 | 2.29 | 4.31 | 3.44 | 2.29 | 2.90 | 2.98 | 3.02 | 2.22 | 1.52 | 1.96 | 1.97 |
| ALARO - CRU | -1.00 | 0.34 | 3.05 | 1.27 | 0.92 | -0.26 | 0.09 | 2.44 | 1.32 | 0.91 | 2.85 | -1.71 | 1.10 | 1.42 | 0.90 |
| MAE ALARO CRU | 2.43 | 2.60 | 3.82 | 2.30 | 2.31 | 2.80 | 3.06 | 3.86 | 2.55 | 2.71 | 4.07 | 3.21 | 2.93 | 1.88 | 1.59 |
| ERA-Interim - CRU | -0.84 | -0.98 | 0.22 | 0.80 | -0.19 | -0.13 | -1.44 | -0.46 | 0.47 | -0.39 | 0.39 | -1.46 | 0.00 | 0.79 | -0.08 |
| MAE ERA-Interim CRU | 1.95 | 1.70 | 1.68 | 1.89 | 1.46 | 2.11 | 2.30 | 2.18 | 2.14 | 1.90 | 1.90 | 1.96 | 1.63 | 1.46 | 1.33 |

**Table 4: Spatial average over the CAS-CORDEX domain and subdomains of climatological mean CRU maximum temperature (°C) for the 1980-2017 period, and biases (°C) and MAE (°C) against CRU for REMO, ALARO-0 and ERA-Interim.**

| | EEU | | | | | WSB | | | | | ESB | | | | |
| --- | --- | --- | --- | --- | --- | --- | --- | --- | --- | --- | --- | --- | --- | --- | --- |
| | DJF | MAM | JJA | SON | Annual | DJF | MAM | JJA | SON | Annual | DJF | MAM | JJA | SON | Annual |
| CRU | -6.50 | 10.23 | 24.91 | 8.57 | 9.38 | -10.94 | 8.04 | 24.13 | 6.74 | 7.08 | -18.52 | 4.78 | 21.97 | 1.93 | 2.64 |
| REMO - CRU | -1.58 | -2.27 | -2.42 | -1.25 | -1.89 | -0.87 | -1.77 | -2.44 | -1.59 | -1.67 | 2.03 | -2.42 | -1.62 | -0.74 | -0.70 |
| MAE REMO CRU | 1.67 | 2.77 | 2.43 | 1.27 | 1.90 | 2.03 | 2.61 | 2.50 | 1.69 | 1.91 | 2.50 | 2.81 | 1.77 | 1.01 | 1.33 |
| ALARO - CRU | 1.34 | -6.06 | -3.36 | -1.47 | -2.41 | 1.97 | -7.10 | -3.83 | -1.68 | -2.69 | -1.85 | -9.87 | -1.28 | -1.51 | -3.64 |
| MAE ALARO CRU | 1.40 | 6.06 | 3.47 | 1.49 | 2.46 | 2.54 | 7.14 | 3.97 | 1.71 | 2.74 | 3.90 | 9.94 | 2.22 | 1.78 | 3.80 |
| ERA-Interim - CRU | -0.48 | -2.65 | -3.02 | -1.33 | -1.88 | -0.47 | -2.13 | -2.63 | -0.39 | -1.41 | -0.65 | -4.17 | -1.14 | -0.64 | -1.66 |
| MAE ERA-Interim CRU | 0.92 | 2.65 | 3.04 | 1.36 | 1.88 | 1.21 | 2.20 | 2.75 | 0.90 | 1.55 | 1.78 | 4.20 | 1.40 | 0.99 | 1.77 |
| | WCA | | | | | TIB | | | | | CAS-CORDEX | | | | |
| | DJF | MAM | JJA | SON | Annual | DJF | MAM | JJA | SON | Annual | DJF | MAM | JJA | SON | Annual |
| CRU | 7.53 | 20.8 | 33.47 | 21.98 | 21.01 | -2.86 | 10.73 | 21.00 | 10.13 | 9.81 | -4.29 | 11.97 | 25.34 | 11.06 | 11.09 |
| REMO - CRU | -0.04 | 0.18 | 0.26 | 0.07 | 0.11 | -1.13 | -1.90 | -1.15 | -1.77 | -1.49 | 0.08 | -1.24 | -1.07 | -0.71 | -0.74 |
| MAE REMO CRU | 1.49 | 1.66 | 2.00 | 1.43 | 1.44 | 3.22 | 3.23 | 2.56 | 2.88 | 2.84 | 2.15 | 2.49 | 2.08 | 1.48 | 1.75 |
| ALARO - CRU | -2.31 | -1.24 | 0.15 | -1.32 | -1.18 | -3.68 | -2.20 | -0.07 | -1.47 | -1.85 | -0.77 | -4.84 | -1.46 | -1.24 | -2.08 |
| MAE ALARO CRU | 2.73 | 2.28 | 2.16 | 1.87 | 1.89 | 3.96 | 3.12 | 2.74 | 2.24 | 2.59 | 2.63 | 5.54 | 2.79 | 1.70 | 2.61 |
| ERA-Interim - CRU | -1.25 | -1.10 | -1.14 | -1.33 | -1.21 | -0.93 | -2.03 | -1.90 | -0.90 | -1.45 | -0.84 | -2.43 | -1.77 | -0.80 | -1.46 |
| MAE ERA-Interim CRU | 2.02 | 1.76 | 1.86 | 1.66 | 1.61 | 2.01 | 2.61 | 2.93 | 2.28 | 2.23 | 1.53 | 2.86 | 2.34 | 1.36 | 1.82 |

**Table 5: Climatological mean CRU precipitation (mm month$^{-1}$) for the 1980-2017 period over the CAS-CORDEX domain and subdomain, and relative biases (%) and MAE (%) against CRU for the RCMs (REMO and ALARO-0), and the other reference datasets (ERA-Interim, MW and GPCC).**


| | EEU | | | | | WSB | | | | | ESB | | | | |
|---|---|---|---|---|---|---|---|---|---|---|---|---|---|---|---|
| | DJF | MAM | JJA | SON | Annual | DJF | MAM | JJA | SON | Annual | DJF | MAM | JJA | SON | Annual |
| CRU | 34.91 | 34.16 | 55.26 | 45.62 | 42.51 | 22.74 | 27.99 | 51.53 | 35.94 | 34.6 | 11.13 | 22.10 | 72.28 | 29.62 | 33.90 |
| REMO - CRU | 12 | 20 | 7 | 9 | 11 | 16 | 25 | 13 | 14 | 16 | 30 | 63 | 8 | 21 | 22 |
| MAE REMO CRU | 18 | 22 | 21 | 13 | 14 | 33 | 34 | 28 | 26 | 25 | 133 | 74 | 17 | 37 | 28 |
| ALARO - CRU | 21 | 12 | 10 | 18 | 15 | 20 | 3 | -4 | 17 | 7 | 35 | -1 | -19 | 21 | -3 |
| MAE ALARO CRU | 25 | 17 | 22 | 19 | 16 | 28 | 17 | 22 | 22 | 15 | 65 | 24 | 28 | 30 | 19 |
| ERA-Interim - CRU | 13 | 19 | 10 | 9 | 12 | 18 | 27 | 16 | 15 | 18 | 29 | 57 | 11 | 31 | 24 |
| MAE ERA-Interim CRU | 18 | 20 | 11 | 10 | 13 | 25 | 29 | 19 | 19 | 21 | 79 | 66 | 16 | 36 | 26 |
| MW - CRU | -11 | -7 | -7 | -6 | -7 | -8 | -5 | -8 | -6 | -7 | -4 | -15 | -13 | -9 | -12 |
| MAE MW CRU | 14 | 10 | 10 | 14 | 14 | 17 | 14 | 15 | 17 | 17 | 33 | 23 | 16 | 33 | 33 |
| GPCC - CRU | -24 | -15 | -7 | -11 | -13 | -12 | -11 | -4 | -8 | -8 | -7 | -21 | -9 | -13 | -12 |
| MAE GPCC CRU | 24 | 17 | 11 | 24 | 24 | 23 | 18 | 10 | 23 | 23 | 30 | 26 | 12 | 30 | 30 |
| | WCA | | | | | TIB | | | | | CAS-CORDEX | | | | |
| | DJF | MAM | JJA | SON | Annual | DJF | MAM | JJA | SON | Annual | DJF | MAM | JJA | SON | Annual |
| CRU | 33.18 | 37.52 | 16.74 | 18.45 | 26.46 | 8.12 | 17.73 | 48.56 | 15.02 | 22.45 | 22.60 | 32.34 | 64.75 | 35.50 | 38.88 |
| REMO - CRU | 17 | -10 | -19 | 18 | 2 | 259 | 194 | 31 | 187 | 110 | 29 | 39 | 4 | 20 | 18 |
| MAE REMO CRU | 45 | 46 | 66 | 43 | 39 | 1169 | 638 | 243 | 240 | 137 | 205 | 107 | 52 | 53 | 39 |
| ALARO - CRU | -2 | -5 | -18 | 9 | -4 | 26 | 36 | 14 | 38 | 23 | 22 | 19 | 1 | 22 | 13 |
| MAE ALARO CRU | 32 | 33 | 78 | 44 | 33 | 260 | 279 | 185 | 107 | 84 | 73 | 54 | 49 | 42 | 30 |
| ERA-Interim - CRU | 21 | 29 | 77 | 38 | 36 | 59 | 117 | 63 | 73 | 75 | 22 | 38 | 19 | 21 | 24 |
| MAE ERA-Interim CRU | 32 | 33 | 123 | 51 | 34 | 267 | 384 | 340 | 131 | 104 | 80 | 72 | 63 | 40 | 32 |
| MW - CRU | -4 | -8 | -2 | 7 | -3 | 14 | 3 | 9 | 20 | 10 | -6 | -4 | -3 | -2 | -3 |
| MAE MW CRU | 32 | 28 | 81 | 32 | 32 | 104 | 100 | 64 | 104 | 104 | 39 | 27 | 31 | 39 | 39 |
| GPCC - CRU | 0 | -7 | -7 | -2 | -4 | -9 | -17 | -4 | -2 | -7 | -7 | -8 | -1 | -5 | -4 |
| MAE GPCC CRU | 31 | 24 | 55 | 31 | 31 | 88 | 90 | 61 | 88 | 88 | 39 | 27 | 28 | 39 | 39 |