# Peer review of "Evaluation of regional climate models ALARO-0 and REMO2015 at"

_Geoscientific Model Development, 2019_

## Referee Comment (RC1) · Anonymous Referee #1 · 20 Mar 2020

Central Asia is one of the least investigated CORDEX domains and any paper dealing with this area is more than welcome. I do not have an objection to this paper but have a general comment on the approach. Central Asia region has a rough topography and is sparsely populated. Therefore the station data in this region is not as reliable and dense as some other regions. Keeping this in mind, does it really make sense to use CRU data as the basis for model evaluation. In these regions, it may make much more sense to use Era-Interim as the basis and not take CRU (that much) into account.

---

## Referee Comment (RC2) · Anonymous Referee #2 · 21 Apr 2020

**Review of "Evaluation of regional climate models ALARO-0 and REMO2015 at 0.22 resolution over the CORDEX Central Asia domain" by Top et al. 2020**

**General Comments**

In this paper the authors present the results of an evaluation conducted over the CORDEX Central Asia domain for two different RCMs: REMO and ALARO-0. Comparing climatological seasonal and annual means obtained from two simulations covering the period 1980-2017 against gridded observational data-sets, they aim to assess the reliability of the models for the region of study, setting the basis for their use for future climate projections. The paper complements the results of other studies on RCMs for the same region, and is believed to be interesting for the regional climate modeling community.

Nevertheless, in its current form the paper suffers from a series of major issues that need to be carefully addressed before it may be considered for publication for Geoscientific Model Development. In general, the quality of the paper is not very satisfactory. The text and the structure of the manuscript need a thorough revision, since information is many times not very clearly expressed or confusing. The presented analyses are too generic and not at all exhaustive. Explanations for evinced models behavior are often hypothesized without any appropriate investigation. Further, I think that different sources of uncertainty such as the error related to the use of different observational data-sets are not properly considered. I discuss the mentioned issues, together with additional ones, in more details below:

**Specific comments**

- The presented analyses are neither exhaustive nor accurate enough for a proper evaluation study. In particular, the analyses of the spatial correlation and of the spatial mean calculated over the entire domain are not very useful. First of all, the evinced conclusions for the mean of the spatial biases calculated over the entire domain might simply be the results of some compensating effect and could vary significantly from one area to another. At the same time, given the heterogeneity of the domain of study, spatial means and correlations calculated over the entire domain can hide model limitations specific to single regions characterized by different physical phenomena. Determining and understanding possible model limitations is one of the final goals of models evaluation and serves as the basis for models development. For these reasons, a quantitative analysis of model performances per sub-regions is therefore required.

- In the text there is lot of confusion between the different sections and their contents, with discussion performed in the results section and some

of the results commented in the discussion part. Also, the authors discuss several variables not in the appropriate subsections. One example is the subsection with the discussion on precipitation results, where the results of temperatures are also partly discussed.

- The authors somehow considered the effect of different observations on the comparison of the maps of climatological biases, as well as for the spatial mean calculated over the entire domain. Nevertheless, also the analysis of the spatial correlation, the ratio of standard deviation and the RMSE should take into account the effect of different sources of uncertainties, among which one of the most important is certainly the effect of different observations. In this context, sub-regions analyses assume even more importance. Additionally, other uncertainties could play a big role for the different regions, such as for example the effect of different boundaries. What happens when these sources of uncertainties are considered? The authors should acknowledge the possible effect of different uncertainty sources and all their analyses must at least take into account the effect of the observational uncertainty on the considered metrics.

- The authors conducted their evaluation considering a single observational data-set for each variable. Then, they basically discussed in each case whether and when the model bias was related to the poor quality of the reference observations, by comparing these with two or three additional data-sets. I am quite critical with the approach they used. In fact, a simple comparison of three or four gridded observational data-sets does not allow to determine the best data for the different regions of the considered domain. For doing this a more robust analysis is needed, considering the initial observational stations of each data-set, their number, their precision and the uncertainty related to the employed interpolation methods. On the other hand, what the authors can do, given the considered data-sets, is to evaluate and take into account the reliability of the given observational data-sets for each point of the domain, by calculating for example the spread of the different observations. Instead of determining whether evinced model biases are due to the reference observational data-set (that in my opinion is not possible to conclude for all the points of the domain, given the available data), the authors should compare models results with the available data-sets, and then discuss whether those biases are within or outside of the range of the observations. In this way they could be able to affirm whether any conclusion on model performances can be drawn for a considered area.

- The authors only investigate climatological values, focusing on the mean bias and on spatial variability. I think that they should be more specific on the choices they made, discussing at least why they focused only on seasonal and annual means and why they did not decide to tackle temporal variability and the seasonal cycle. In particular, I would suggest to add some analysis on the mean seasonal cycle, since the authors claim in their

manuscript that some of evinced model biases might likely be related to a wrong simulation of it.

- The authors focus their analyses on four variables, but then they discuss the results only for temperature and precipitation. Why the discussion is not conducted consistently among the different variables? Additionally, why are the analyses of tmin and tmax not carefully conducted in the same way as for precipitation and temperature, considering different observational data-sets?

- The effect of the boundaries on the different variables can only be estimated by performing different simulations changing the boundary conditions. The authors should take into account this point whenever they claim that errors in the boundaries are the cause of evinced biases. They can eventually be able to (only partially) support these claims only by performing additional simulations with different boundary conditions.

- In many cases the presented analyses are very superficial and most of the raised conclusions are mainly hypothesized without a proper demonstration. Additionally, sometimes the authors simply use the maps of the bias to interpret the results of the spatial means. Why should that be interesting and how such an analysis might help in evaluating models results? More in depth analyses are required, including the already mentioned investigation of the seasonal cycle and a quantitative comparison of model results and observations per sub-regions. Every hypothesis on the possible reason of evinced biases should be effectively supported by specific analyses or by bibliographic references.

- In the manuscript there is a tendency of justifying the bias of the model with the poor quality of CRU. For precipitation, for example, the authors state that CRU underestimates precipitation values: in this case, why do not you perform the comparison against the GPCC as reference then?

- I would be more careful stating that evinced results are in the range of the ones obtained for other studies and that indeed the models can be employed for climate projections. You affirm this only considering the reference of Kotlarski for Europe. Additional studies for more regions should be considered. Still, the authors must acknowledge the fact that extremely large biases are present over extensive parts of the domain. For these, either any conclusion on model reliability can be drawn due to high observational uncertainties or model results can not be considered very trustworthy.

Beside these major concerns there are a series of minor, but still relevant issues that need to be addressed by the authors.

**Minor Comments**

- lines 35-37: How large are these ensembles? what about ensembles for the other CORDEX regions, such as for example North America?

- line 51: the term "validating" is normally considered not very appropriate when comparing climate models and observations, in particular in cases like this one, where large uncertainties in observations are present. "evaluating" would be a more appropriate term.

- line 56: Same as above. Replace validation with evaluation everywhere in the text.

- lines 62-63: delete "...that are sparsely populated" since it is repetitive (you already said that in the first line of the period).

- line 65: "more extreme values": more extreme than what? just use "extreme values"

- lines 67-68: The comparison against different observational datasets is useful only to address the reliability of observational datasets and does not help solving the problem of the lack of an ensemble. Please reformulate.

- lines 70-71: Similarly as expressed in my major concerns, you cannot directly prove that similar biases in the two models are due to observational errors. In principle, uncertainty in the observational data-sets allow to say that over certain areas the observations are more or less reliable and whether robust conclusions can be drawn in this case. Please reformulate this part.

- line 80: complemented by

- lines 83-88: I think that this part would be more appropriate for the introduction rather than for the methods.

- line 106-107: "The outer domain consists of the inner domain plus a coupling zone of eight grid points in each direction.": This holds true for both domains, right? eventually specify.

- Fig.1: Where did the authors take the information on the topography from? the upper limit of the colorbar of 3000m seems not reasonable for the area.

- line 131: what is the vertical extension in meters of the domain of study for each of the two models?

- lines 139-140: Correct into: "...at the boundaries, up to the 31st of December 2017."

- lines 143-147: Is there any reason why in the case of ALARO-0 one year can be considered enough for spin-up with respect to the 31 years considered for REMO? Please specify.

- lines 149-150: This sentence, in the way it is expressed, is not properly correct. In fact, you do a comparison of model results only against the CRU, while then you compare the different observations among them. Please better reformulate this sentence according to the comparison you will decide to perform.

- lines 151-152: Again, given your analyses you can not tell whether the bias of the models is due to the observational uncertainty. What you can eventually say is that high uncertainties do not allow to draw robust conclusions.

- lines 160-165: I do not manage to find the reference of New et al. 2002 in your paper. Are not there any more up-to-date publications discussing problems of the latest CRU releases? Also, do not you think that the New et al. 1999 publication is more general and it might also apply to other observational datasets rather than simply the CRU? Please consider that all your considered data-sets are somehow characterized by uncertainties (Flaounas et al., 2012; Gómez-Navarro et al., 2012).

- lines 168-171: what about quality of UDEL for other variables than precipitation?

- lines 176-179: Please, make clear that Hu et al. 2018 only investigated the most central part of your domain of study. Also, the same study states that GPCC underestimates all seasonal means, not only but especially in spring.

- lines 181-186: The original resolution of ERAInterim is not 25 Km but approximately 80 km. If you used the data provided by the ECMWF at 25 Km, be aware that these are interpolated data. Please specify this in the text.

- lines 181-186: an additional question concerns your choice of using ERAInterim data interpolated at 25 km: why you do not directly download ERAInterim data already onto a 50 Km grid?

- lines 186-188: First of all avoid saying initial errors in the boundary conditions, since it generates confusion. Then, how should the comparison of temperature derived from ERAInterim with the one of the models help you determining what is the effect of errors in the boundaries? The only way to assess the effect of the boundaries on the RCM results is to drive the same simulations with different boundaries.

- lines 189-190: The outputs of an RCM are dependent (but not univocally

determined) on the values of several variables with which the model is forced at its boundaries. These variables will have an effect on several model variables. The temperature of the model is not only dependent on the values of temperatures provided at the boundaries, but other variables play a role. The same holds true for precipitation. If the model is forced with wrong temperatures it is very likely, at least from a theoretical perspective, that both model temperature and precipitation will be both badly reproduced.

- lines 198-199: first of all UDEL and CRU have the same 0.5 degree resolution. Also GPCC is available at such resolution. Reformulate this period in a more accurate way, considering the fact that the "upscale" is only necessary for the models outputs.

- lines 207-209: reformulate this period.

- line 219: seasonal means of

-line 227: what do you mean by limited bias? better specify.

- lines 228-229: First of all, you start discussing annual means but you put the relative figures at the bottom row of your image: move them up. Then, in my opinion, according to the scale you use in your plots, it seems that in both cases the absolute bias exceeds 3C over a very extensive part of the domain and not only over mountainous regions. Maybe the scale you are using does not help to clearly distinguish which areas are above or below a certain threshold. Try to change your scale.

- lines 229-230: Also the REMO exceeds the 3C range, in particular in winter. Please reconsider your sentence.

- lines 230-231: not totally correct. In fact, the biases ,when considering the entire domain, are particularly pronounced for ALARO-0 mainly in spring, over the northern part of the domain. In winter the most pronounced bias seems to be the one of REMO for north-western Mongolia. For summer and autumn the biases for the two models present a very similar range. The same holds true when considering only the eastern half of the domain. Reformulate this part.

- lines 231-233: Actually you should really emphasize that the two models seem to have a completely different pattern of the bias of temperature in winter: one shows a bipolar behavior between North and South, while the other between East and West, with a peak in warmer simulated temperatures over north-western Mongolia. I think that it would be really important for the authors (and a very nice opportunity) to better investigate the causes of the two different behaviours. This could give us some clue on model limitations in the simulations of temperatures over the region, that seems to be a general issue for climate models.

- lines 233-234: What do you want to evince from this? why Scandinavia and not another region? Also, how is the bias similar in the two cases?

- lines 238-239: Important biases are present in MAM also for REMO, for some regions such as the Western fringes of the Tibetan Plateau. Also, for both models biases exceed 3C over a large part of the domain in MAM. Reformulate.

- lines 239-241: What do you mean by limited? you mean that biases are not very pronounced in summer? reformulate.

- lines 239-241: also for REMO there are warm biases, even though they are inherent to a smaller portion of the domain, in particular with respect to ALARO. Be more precise.

- Fig 2: Beside my previous comment on the figure colorbar, the quality of the image could be further improved by reducing white spaces in between rows and moving the names of the seasons on the left side of the figures. Additionally, units should be added to the colorbars, that should also be moved: the colorbar of the bias should be positioned in between the two columns for the bias of REMO and ALARO.

- lines 248-249: The mentioned gradient is not very clear, in particular in summer.

- line 249: "The outcomes of both RCMs for the mean temperature agree well with the CRU data in autumn (SON)": That is not totally true. In fact, performances of REMO in terms of simulated seasonal climatologies are very similar for autumn, but also for spring and summer.

- lines 254-255: what do you mean by "should be placed in perspective"? in which perspective? please reformulate this period.

- lines 258-259: "it is clear from Table 2 that the strong cold bias during spring in the north for the ALARO-0 model has a larger negative impact on the spatially averaged bias than the warm bias during winter": I would avoid talking about "negative impact" of the bias over some region on the calculation of the spatial mean bias. Instead, you could say that the spatial bias is largely influenced by the pronounced negative/positive bias over specific regions.

- lines 264-267: "However, the biases during summer are ... due to the smaller spatial variability in temperature during summer". I think that this period is not very clear and needs to be reformulated, eventually considering additional analyses supporting your conclusions. First of all in summer, in the observations, you have less spatial variability (more accurate than smaller spatial range) than in the other seasons. This is evident from the figure, even though it would

be nice if you could support such conclusion with a more quantitative analysis of the CRU spatial variability. Additionally (and most importantly), in your analyses you do not effectively demonstrate that a lower correlation is due to a lower spatial variability in summer. Why can it not be simply due to a worse agreement in the spatial variations between the models and the observational dataset?

- lines 276-277: that is exactly one of the reasons why it would be better to consider the analyses per sub-regions.

- lines 290-291: I think that your explanation on the reasons of a more negative bias for TMIN than for T2 is not exhaustive. Additionally, this needs to be moved to the discussion part.

- lines 297-298: "Following the main trend..": confusing, reformulate.

- lines 299-301: "The warm minimum temperatures of the RCMs indicate that they underestimate the coldest diurnal temperatures or that the observational CRU dataset overestimates them." There are several issues in this period. First of all you need to reformulate your sentence because it is not the minimum temperature of the model that underestimates observation values but rather the model itself. Also, if the minimum temperatures are warmer than observations, it means that the model overestimates (and not underestimates) the coldest diurnal temperatures. Finally, from the comparison of model results against CRU you can only affirm that the models underestimate minimum diurnal temperatures. You can not prove that the observations overestimate them. The fact that CRU might overestimate them is a possibility, but still is not inherent to the behaviour of the model (nor it is evident from the figure you are commenting).

- lines 312-313: "except for the summer": why except if your are talking about annual values?

- line 315: less good than what?

- line 323: you do not need to specify that temperature is a variable here

- lines 323-324: You need to reformulate this sentence. In this case you have to specify that the negative TMAX bias is particularly remarkable in spring for the northern part of the domain, and also, to a less degree, in summer. In winter some other less extended parts of the domain, such as the north-eastern part, show a colder bias than REMO. In Autumn results are more similar between the two models.

- end of line 326: the cold bias

- lines 326-328: Fig. 4 shows minimum temperatures. Then, how can we deduce from this figure that the bias in TMIN is due to maximum temperatures? please better explain and eventually reformulate this period.

- line 342: "This means that ALARO-0 fails to reproduce the low nocturnal temperatures": This belongs to the discussion on minimum temperatures. Additionally, the model still fails in simulating warmer temperatures, despite the smaller bias when compared to TMIN.

-lines 344-346: You should discuss about minimum temperature in the appropriate section.

- When you comment the maps of the bias, try to discuss the different seasons from up to down, consistently with the figures.

- lines 364-366: This part should be moved to the discussion section.

- lines 365-366: why however? also, you did not discuss until this point the uncertainty of CRU: how can you claim that the reason for the wet bias is due to the observations?.

- lines 370-372: By whom is the bias turned into something else in summer? and how?

- lines 372-376: It would be nice if you could perform the analyses of the seasonal cycle to support your conclusions. This would make your evaluation more complete and exhaustive, while at the same time allowing to effectively confirm or deny your conclusions.

- lines 390-391: "The dry biases for ALARO-0 in Table 5 are thus caused by the simulation of systematically less precipitation than the precipitation amounts in the CRU data.": Reformulate. It is obvious that if the model underestimates precipitation, it simulates less precipitation than observations.

-
- line 393: systematically

- lines 392-394: "The lower accuracy of simulated precipitation is due to the fact that precipitation is less systematic affected by land cover and topography compared to temperature": First of all that is quite a strong assumption given the extent and heterogeneity of the domain you are considering. Additionally, you did not perform (at least it is not reported in the paper) any analysis that supports your conclusion.

- lines 400-404: This is incorrect. In fact Russo et al. 2019 showed that uncertainty in observations is high over the north-eastern part of the domain,

not that CRU overestimates the diurnal temperature range over the region.

- line 404: Why hence?

- lines 404-406: Again, how can you surely state that the model underestimates values of the diurnal temperature range due to higher observation values?

- lines 407-408: why Czech Republic? what happens in other regions?

- lines 420-422: This is just an assumption that needs to be proven. Models develop their answer that is, to a certain degree, independent from the boundaries. To test your hypothesis, one easy experiment that could be conducted is to use different boundaries and compare the results.

- line 425: "They related this warm bias already to shortcomings in the simulation of snow": this means that they explained the bias differently than with the boundary effect as you explained in the lines from 423 to 425.

- lines 430-431: "Hence, we conclude that the warm forcing is the main reason for the warm bias over Eastern Russia during winter.": I further have to highlight that you can not make such conclusion, until you do not test different boundaries.

- lines 435-436: As before, it would be nice if you could do the analyses of the seasonal cycle since you mention it for the interpretation of your results.

- lines 440-442: How the fact that for Belgium there is some correlation between warm bias and cloud cover representation could explain the same for northeastern CAS. You could do some analyses on cloud cover to support your conclusion.

- lines 443-445: "Both could be due to too much cloud cover": according to whom? In theory it could be due to any reason.

- lines 448-451: These considerations are important: it would be nice to put them in a more objective context. Additionally, you say that New et al. show that CRU underestimates temperatures for Russia. Then you talk about Western Russia. If you state that temperatures from CRU are not good for Russia, then they can not be good for a part of it and bad for the rest. Reformulate.

- Fig. 10,11: To make the discussion easier I would suggest to plot the maps of the differences between different observational data-sets together, using the spread of the observations among the different data-sets. In this way you can easily know which areas are more reliable and which are not.

- lines 465-467: the less reliable observations do not explain the bias, rather they do not allow to draw any conclusion.

- line 471: "...winter and overestimate it during." During what?

- lines 482-484: what happens when you compare ALARO-0 with the other data-sets over the entire domain?

- lines 484-486: you mention two gridded data-sets: to which data-sets are you referring here? please better specify.

- lines 486-489: again, you can claim that the bias is relative to the employed boundaries only performing a new simulation with different boundaries. Also, how can you be sure that ERA-Interim overestimates specific humidity?

- lines 489-490: You are claiming this from the field means I guess. I think that plotting the maps of the bias of the models against all the different observational data-sets might help the discussion of your results.

- lines 489-490: How do ERA and REMO parameterize precipitation since you mention that they do it differently? Specify.

- lines 492-493: "This difference between ALARO-0 and REMO is related to the 3MT cloud microphysics scheme of ALARO-0": where did you demonstrate this?

- lines 496-497: Again, this is hard to affirm simply using three observational data-sets. The authors have to acknowledge the low number of observational data-sets. As I mentioned in many previous comments I personally think that it would be better to approach the differences between the observational datasets in terms of reliability rather than determining who is more correct.

- Fig. 11: In the colorbar of the bias, are units percentage? with respect to what? Please specify.

- lines 501-502: Not completely true. Specify that, as evinced from your maps, the wet bias in ERA (with respect to the other 3 data-sets) is only relative to the eastern part of the domain.

- line 520: "that that". Correct.

- lines 536-537: "REMO simulates the precipitation fairly well and ALARO-0 performs very well." How can you state that their performances are good?

- lines 539-540: "The warm temperatures obtained with REMO ... can be linked with the dry and wet bias in winter and spring respectively." Why and

how can they be linked?

- lines 540-541: In which way the link between temperatures and precipitation should strengthen your hypothesis of a delay by REMO in the simulation of snow cover? Can you be more specific?

- lines 545-546: "The persistent warm bias over Pakistan and Northern India of both RCMs can be explained by the persistent underestimation in simulated precipitation over this region by both RCMs.": how can you state that given your analyses?

- lines 547-550: You refer to the fact that your results are within the ranges of models for other domains, but then you only mention the results of Kotlarski et al. for Europe. You need more references.

- lines 562-563: That is arguable, given your analyses. How do you define an acceptable range?

- lines 565-567: You cannot state this, until you force the model with different boundaries and you conduct an analysis of snow cover (what you can eventually do for snow cover is to reference to the evidences from other studies).
- Table 1: This table is not easily readable. Could you find a way to make the distinction between the different data-sets a bit clearer?

---

## Referee Comment (RC3) · Anonymous Referee #3 · 26 Apr 2020

This paper describes the results of two models (REMO and ALARO-0) simulations over CORDEX Central Asia domain. Authors compared simulated temperature and precipitation climatology and concluded that both the models are capable to reproduce CAS climate.

Reading the paper I had an impression it is a kind of technical report but not a scientific manuscript suitable for GMD. I do not see any science by describing how large biases in models are without any reasonable explanation where they come from. Authors took models which were tuned for Europe, implemented them for CAS, obtained huge biases and concluded: "That's it." Therefore I would recommend the manuscript for

publication only in case it will be substantially revised.

Major points

1.Analysis (but not referring to other models results) of model biases is required. Where they come from? Is it large scale atmospheric circulation or local processes, e.g. atmosphere – land heat/moisture exchange? In this sense it would be interesting to look in mean sea level pressure (MSLP) biases. For example, the warm temperature DJF bias as well as huge overestimation of DJF precipitation in REMO could be because of underestimation of Siberian High.

2.The models show quite a substantial differences in biases. Considering the eastern part of CAS it is clearly seen that in cold seasons REMO simulates 2m temperature much better then ALRO. Furthermore ALRO results with almost 10K bias over quarter of the domain are inacceptable. The opposite is seen for precipitation which is simulated by ALRO better. Based on these results authors can take heat (moisture) fluxes as well as heat (moisture) transports from both the models (assuming that "better" model reproduces better fluxes (transports)) and try to analyze which of them leads to produce mentioned above biases.

3.For better understanding I would also recommend to analyze the climatological annual cycle of some quantities, like temperature, precipitation and heat fluxes at least for the eastern part of the domain (from Mongolia to the east), where the biases are really large. For such a big domain with a plenty of climatological zones Taylor diagrams are more a kind of speculation. E.g. in case the climatological temperature varies from +30C in the South to -30C in the North spatial correlation will be high with any kind of model.

4.Authors should have a more deeper look into previous studies done with the same models. In particular ones were done with REMO. Since REMO existence (more then 20 years) there are many papers with REMO simulation results over regions partially included in CAS, e.g. whole the northern part: Niederdrenk, 2013 (PhD), Niederdrenk

et al., 2016 (Clim. Dyn.), Sein et al., 2014 (Tellus); south-eastern part: Xu et al, 2018 (Clim. Dyn.).

5.Authors claim that some of the biases come from the ERA-Interim forcing. That is quite an ambitious conclusion, in particular for Siberian continental climate. This conclusion has to be proven with some additional simulations. It is not a big deal to take a lateral boundary conditions from some of the global climate model, to simulate ca. 10 years and to look if the large scale biases are similar or not. I think with available computer recourses it should be just 3-4 working days.

Minor points

L. 23: I do not think that with large scale 8-10K 2m temperature biases and more then 100% precipitation biases over quarter of the model area both models reproduce climate "reasonably well".

L.24-25: It has to be done in this work, but not postponed to the unclear future

L.35: Even being a not an expert in CORDEX and even for CORDEX domains mentioned by authors, I know much more works based on multi-model regional simulations. E.g. Africa: Paxian et al. (JGR-Atmos, 2016); Mediterranean: Damaraki et al. (Clim.Dyn, 2019), Gaertner et al. (Clim. Dyn, 2017), Soto-Navarra et al. (2020, Clim.Dyn).

L.61: "Absence of reliable observational data sets". Over China and Russia? Maybe 20 years ago "yes" (describing CRU data authors site work from 1999), but at the present time it sounds at least strange.

2. Methods. See above (L.35) Central America: Cabos et al. (2019, Clim. Dyn.), Southeast Asia: Zhu et al. (2020, TAC), Arctic: Akperov et al. (2019, Global and Planetary Change; 2018, JGR)

L.94: I would remove word "sea". In a middle school I have learned that Black, Caspian Red and Baltic seas are seas, but it is hard to say that they are barely covered with

CAS domain.

L.96: Before claiming it, authors should "google" a word "HighResMIP". In the framework of this project there are many global climate model simulating climate on 25 km resolution, i.e. the same resolution as authors use for their regional simulations.

L.106 and in other places: I would suggest to use not "coupled zone", but "sponge zone". Forcing a regional model with reanalysis has nothing to do with coupling.

L.129: But what about dynamical core itself? Please explain at least in the way it is done for ALRO above, i.e. special discretization, advection (e.g. in ALRO it is based on semi-Lagrangian algorithm and what about REMO?)

L.137-138: What about upper boundary? Which height does it have? 10hPa? 50hPa?

L.202: As far as I know almost all the atmospheric models (including REMO and ALADIN) provide direct output of Tmax and Tmin which are obtained every model time step. Why not to use them directly?

3. Results: As I mentioned in "major points", not only seasonal means but also climatological annual cycle for the quantities averaged over different areas has to be included.

L.229: Exceeded. How much does it exceeded? On the plot I can only see that it is larger then 10K.

L.234: What has Scandinavia to do with Mongolia? They have completely different climate. In the same way REMO group can write: Paxian et al. (2016) showed a strong precipitation bias over Guinea in Africa. Maybe that is also a reason of REMO prcip. bias over East Siberia?

L238: Actually the strongest cold bias over Europe in REMO is at Spring. It is not visible in most of the papers, because mainly they show DJF and JJA only.

L.360 (Fig.8) Relative difference in mm/month? I think it should be in (%)

To all the figures with biases: For the biases I would avoid linear color bar and extend it for larger values. E.g. for the temperature something like: 0,1,2,3,5,7.5,10,12.5,15 and for precip. (%) 0,10,20,30,50,75,100,125,150,200

L.405: What the Czech Republic has to do with Central Asia? Do they have similar climate? I have here the same claim as at L.234. Authors should provide arguments which has something to do with CAS and not speculations like: we have warm bias in Mongolia, because in French Polynesia is to rainy.

L.414: I would not say that up to 10K large scale temperature bias is something which is VERY well

L.423: "..assigned to this forcing". As it was mentioned above (Major points), before speculating about it, please do some simulations with different forcing.

L.433: "Ozturk et al. . . ., but they did not explain it." And? If Ozturk did not explain it, it is over? Why don't you try to explain it in your manuscript.

L.428, 448, etc. New et al. (1999). You discuss present climate and present observational data set citing a work from 1999? There is a quite a big difference between the number of observations before 1999 and now.

Fig. 11: I think should be MW, but not WM. As well as (%), but not mm/month

Conclusion: In the scientific sense conclusion is very poor simply describing how large model biases are only. The only one "explanation" of their origin is "models are good, but observations are bed", based on results obtained more then 20 years ago, in 1999. I would suggest to authors to bring more "scientific analysis" into the manuscript considering comments written above. Maybe it will bring the paper from "technical report" to "scientific manuscript".

———————————————————

---

## Author Comment (AC1) · 30 May 2020

**Author response to the review of Anonymous Referee #1**

*Referee 1, thank you for reviewing our manuscript. Our answers on all questions, suggestions and remarks can be found on the next pages. Firstly, we summarize the major changes we will make to the revised version of the manuscript based on the comments of the different reviewers:*

- *We will include an analysis of the annual cycle over the subdomains as defined by the IPCC6 report (Iturbide et al., 2020) which are situated within the CAS-CORDEX domain. The results, both for the RCMs and the gridded datasets, for the mean temperature and precipitation are given in Fig. A1 and A2.*
- *We will approach the differences between the gridded datasets in a different way. The spread between the gridded datasets (Fig. A3) will be used as an estimate of the uncertainty.*
- *We will improve the discussion section by describing which model features can explain the significant biases that were obtained over certain regions.*
- *We will include some additional recently published scientific papers in our revised manuscript e.g. Harris et al. 2020; Wang et al. 2020; Zhu et al. 2020.*

[Figure]

*Fig. A1: Annual cycle of the mean temperature (°C) over different subdomains.*

[Figure]

*Fig. A2: Annual cycle of the precipitation (mm month$^{-1}$) over different subdomains.*

[Figure]

*Fig. A3: Spread in mean temperature between the gridded datasets CRU, MW and ERA-Interim.*

Central Asia is one of the least investigated CORDEX domains and any paper dealing with this area is more than welcome. I do not have an objection to this paper but have a general comment on the approach. Central Asia region has a rough topography and is sparsely populated. Therefore the station data in this region is not as reliable and dense as some other regions. Keeping this in mind, does it really make sense to use CRU data as the basis for model evaluation. In these regions, it may make much more sense to use Era-Interim as the basis and not take CRU (that much) into account.

*Thank you for your positive comment. It is indeed true that the gridded datasets are not very reliable in some regions, as we stressed in our paper. We did not take ERA-Interim as reference since this product has some model dependency and might suffer from similar errors that are reproduced by our models which are forced by ERA-Interim for this evaluation study. The station observations also undergo manipulations to obtain a gridded dataset but these steps are not linked with any NWP model. Moreover, the relatively coarse (80km or 0.75 degrees) resolution of ERA-Interim makes it less suitable to serve as a reference for higher-resolution regional climate models due to the larger representativity issues. CRU is also quite coarse (0.5 degrees) but it has still a higher resolution than ERA-Interim.*

---

## Author Comment (AC2) · 30 May 2020

**Author response to the review of Anonymous Referee #2**

*Referee 2, thank you for your very detailed comments. We are delighted to take all of your comments into account to improve the manuscript. Our answers on all questions, suggestions and remarks can be found on the next pages. Firstly, we summarize the major changes we will make to the revised version of the manuscript based on the comments of the different reviewers:*

- *We will include an analysis of the annual cycle over the subdomains as defined by the IPCC6 report (Iturbide et al., 2020) which are situated within the CAS-CORDEX domain. The results, both for the RCMs and the gridded datasets, for the mean temperature and precipitation are given in Fig. A1 and A2.*
- *We will approach the differences between the gridded datasets in a different way. The spread between the gridded datasets (Fig. A3) will be used as an estimate of the uncertainty.*
- *We will improve the discussion section by describing which model features can explain the significant biases that were obtained over certain regions.*
- *We will include some additional recently published scientific papers in our revised manuscript e.g. Harris et al. 2020; Wang et al. 2020; Zhu et al. 2020.*

[Figure]

*Fig. A1: Annual cycle of the mean temperature (°C) over different subdomains.*

[Figure]

*Fig. A2: Annual cycle of the precipitation (mm month$^{-1}$) over different subdomains.*

[Figure]

*Fig. A3: Spread in mean temperature between the gridded datasets CRU, MW and ERA-Interim.*

General Comments

In this paper the authors present the results of an evaluation conducted over the CORDEX Central Asia domain for two different RCMs: REMO and ALARO-0. Comparing climatological seasonal and annual means obtained from two simulations covering the period 1980-2017 against gridded observational data-sets, they aim to assess the reliability of the models for the region of study, setting the basis for their use for future climate projections. The paper complements the results of other studies on RCMs for the same region, and is believed to be interesting for the regional climate modeling community. Nevertheless, in its current form the paper suffers from a series of major issues that need to be carefully addressed before it may be considered for publication for Geoscientific Model Development. In general, the quality of the paper is not very satisfactory. The text and the structure of the manuscript need a thorough revision, since information is many times not very clearly expressed or confusing. The presented analyses are too generic and not at all exhaustive. Explanations for evinced models behavior are often hypothesized without any appropriate investigation. Further, I think that different sources of uncertainty such as the error related to the use of different observational data-sets are not properly considered. I discuss the mentioned issues, together with additional ones, in more details below:

Specific comments

- The presented analyses are neither exhaustive nor accurate enough for a proper evaluation study. In particular, the analyses of the spatial correlation and of the spatial mean calculated over the entire domain are not very useful. First of all, the evinced conclusions for the mean of the spatial biases calculated over the entire domain might simply be the results of some compensating effect and could vary significantly from one area to another. At the same time, given the heterogeneity of the domain of study, spatial means and correlations calculated over the entire domain can hide model limitations specific to single regions characterized by different physical phenomena. Determining and understanding possible model limitations is one of the final goals of models evaluation and serves as the basis for models development. For these reasons, a quantitative analysis of model performances per sub-regions is therefore required.

*We agree that there might be some compensating effects due to the spatial means over the large domain. In order to improve our analysis we will add a section evaluating the RCMs over subdomains that are defined by the IPCC6 report (Iturbide et al., 2020) and that are situated in the CAS-CORDEX domain.*

- In the text there is lot of confusion between the different sections and their contents, with discussion performed in the results section and some of the results commented in the discussion part. Also, the authors discuss several variables not in the appropriate subsections. One example is the subsection with the discussion on precipitation results, where the results of temperatures are also partly discussed.

*To account for this comment we will rearrange the text in the results and discussion section to improve the readability of the text.*

- The authors somehow considered the effect of different observations on the comparison of the maps of climatological biases, as well as for the spatial mean calculated over the entire domain. Nevertheless, also the analysis of the spatial correlation, the ratio of standard deviation and the RMSE should take into account the effect of different sources of uncertainties, among which one of the most important is certainly the effect of different observations. In this context, sub-regions analyses assume even more importance. Additionally, other uncertainties could play a big role for the different regions, such as for example the effect of different boundaries. What happens when these sources of uncertainties are considered? The authors should acknowledge the possible effect of different uncertainty sources and all their analyses must at least take into account the effect of the observational uncertainty on the considered metrics.

*In order to visualize the uncertainty in data of gridded observational datasets we will add graphs to the manuscript with curves that show the differences between the gridded observational datasets for the annual cycle of the different subregions. The spread of the curves of the different gridded datasets can be considered as a measure of uncertainty.*

*It is indeed true that the positioning of the boundaries might have an impact on the climate experiments (Rummukainen, 2009), but it is not the aim of this paper to investigate the effect of the domain choice on the resulting RCM data. This work is undertaken within the CORDEX framework which provides guidelines on domains, resolution,... in order to enable RCM intercomparisons between different modelling groups. Therefore, we used the CAS-CORDEX domain as described by the CORDEX project for our model experiments. Although running the same RCMs over different domains would be interesting, it does not fit the aim of our study that frames into the AFTER project. Moreover, it is impossible to realize such an investigation on a short timescale as it would necessitate writing new proposals to obtain computing time on the Tier-1 HPC infrastructure.*

- The authors conducted their evaluation considering a single observational data-set for each variable. Then, they basically discussed in each case whether and when the model bias was related to the poor quality of the reference observations, by comparing these with two or three additional data-sets. I am quite critical with the approach they used. In fact, a simple comparison of three or four gridded observational data-sets does not allow to determine the best data for the different regions of the considered domain. For doing this a more robust analysis is needed, considering the initial observational stations of each data-set, their number, their precision and the uncertainty related to the employed interpolation methods. On the other hand, what the authors can do, given the considered data-sets, is to evaluate and take into account the reliability of the given observational data-sets for each point of the domain, by calculating for example the spread of the different observations. Instead of determining whether evinced model biases are due to the reference observational data-set (that in my opinion is not possible to conclude for all the points of the domain, given the available data), the authors should compare models results with the available data-sets, and then discuss whether those biases are within or outside of the range of the observations. In this way they could be able to affirm whether any conclusion on model performances can be drawn for a considered area.

*We agree with the reviewer that the spread on the observational datasets is relevant when evaluating the performance of the RCMs. Therefore, we will add maps with the spread between the gridded datasets instead of using Fig. 10 and 11. Additionally, we will add graphs to the manuscript showing the annual cycle of each gridded dataset and each RCM for different subregions. The difference between the curves of the different gridded datasets shows the spread between the different gridded datasets which can be considered as a measure of the observational uncertainty and provides evidence of the performance of the RCMs.*

- The authors only investigate climatological values, focusing on the mean bias and on spatial variability. I think that they should be more specific on the choices they made, discussing at least why they focused only on seasonal and annual means and why they did not decide to tackle temporal variability and the seasonal cycle. In particular, I would suggest to add some analysis on the mean seasonal cycle, since the authors claim in their manuscript that some of evinced model biases might likely be related to a wrong simulation of it.

*We indeed focused too much on the spatial variability. We agree that the temporal aspects can not be fully understood with the current figures in the manuscript, therefore we opted to add graphs with annual cycles based on monthly data for different subregions.*

- The authors focus their analyses on four variables, but then they discuss the results only for temperature and precipitation. Why the discussion is not conducted consistently among the different variables? Additionally, why are the analyses of tmin and tmax not carefully conducted

in the same way as for precipitation and temperature, considering different observational data-sets?

*We needed the minimum and maximum temperature together to bring the story about the limited diurnal cycle of the RCMs. That is the reason why we decided to split the discussion only in temperature and precipitation which indeed is different to the results section that had four subsections. We will merge the sections of minimum and maximum temperature in the results in a general section about the diurnal temperature range so it is clear that the different variables should be interpreted together to understand the processes later on in the discussion. In the discussion we will use the same structure of subtitles.*

*The evaluation of Tmin and Tmax is not conducted in the same way since the observational data was not available for all gridded datasets. The Matsuura and Willmott dataset of UDEL does not contain data about the Tmin and Tmax or the diurnal temperature range.*

- The effect of the boundaries on the different variables can only be estimated by performing different simulations changing the boundary conditions. The authors should take into account this point whenever they claim that errors in the boundaries are the cause of evinced biases. They can eventually be able to (only partially) support these claims only by performing additional simulations with different boundary conditions.

*As mentioned before, it is out of the scope of our research and the manuscript to do an in depth study of the effect of the boundaries due to the aim of the use of the CORDEX domain, the restricted computing time and the goals of the AFTER project which were the driver of these CAS-CORDEX simulations.*

- In many cases the presented analyses are very superficial and most of the raised conclusions are mainly hypothesized without a proper demonstration. Additionally, sometimes the authors simply use the maps of the bias to interpret the results of the spatial means. Why should that be interesting and how such an analysis might help in evaluating models results? More in depth analyses are required, including the already mentioned investigation of the seasonal cycle and a quantitative comparison of model results and observations per sub-regions. Every hypothesis on the possible reason of evinced biases should be effectively supported by specific analyses or by bibliographic references.

*As mentioned above, we will add a section with subregions that are lying within the CAS-CORDEX domain. We agree that an annual cycle based on monthly data improves the evaluation and insights. We will check which statements are not substantiated enough and we will add evidence that is forthcoming out of the added figures or we will refer to other scientific articles where needed.*

- In the manuscript there is a tendency of justifying the bias of the model with the poor quality of CRU. For precipitation, for example, the authors state that CRU underestimates precipitation values: in this case, why do not you perform the comparison against the GPCC as reference then?

*GPCC does not contain temperature data. Since it was important for us to refer for each variable to the same reference dataset in order to compare the performance of the different variables, we took CRU as a reference. By adding the analysis of the annual cycle over the subregions it will be easier to compare the RCM outcomes with the different datasets directly.*

- I would be more careful stating that evinced results are in the range of the ones obtained for other studies and that indeed the models can be employed for climate projections. You affirm this only considering the reference of Kotlarski for Europe. Additional studies for more regions should be considered. Still, the authors must acknowledge the fact that extremely large biases are present over extensive parts of the domain. For these, either any conclusion on model reliability can be drawn due to high observational uncertainties or model results can not be considered very trustworthy.

*We agree, for some parameters significant biases are present over parts of the domain for some seasons. The ALARO-0 RCM has a large positive temperature bias in winter over the northern part of the domain. The REMO model has difficulties in reproducing the observed precipitation patterns over the orography of Central-Asia. We agree that the biases observed in this study should be kept in mind when presenting future projections. We find it therefore important to publish an exhaustive evaluation study. In this evaluation study we saw that the main patterns are modelled correctly and therefore we concluded that we can move on towards climate projections. We will add to our conclusion that these large biases should be kept in mind when looking to the future projections. Additionally, to deal with the biases in impact studies, several bias adjustment methods have been tested within the AFTER project and the most suitable method will be applied before simulations for impact studies are done with these climate data. It is not in the scope of this evaluation study to explain the details about bias adjustments and impact modelling but to avoid misunderstandings we will add that bias adjustment is one of the possibilities when mentioning that the RCMs can be used for future projections.*

*In other scientific publications where models over the CAS-CORDEX domain were run there are as well large biases over certain parts of the domain (Ozturk et al., 2012; Ozturk et al., 2016; Russo et al., 2019) and even for RCMs run over subregions large biases were found (Wang et al., 2020; Zhu et al., 2020). There are not a lot of scientific articles to compare our results with and to refer to, however in the meantime some new studies are published with model evaluations over a subdomain of our domain and we will refer to them in the updated manuscript. We will thus rewrite the discussion and refer to more scientific articles.*

Minor Comments

- lines 35-37: How large are these ensembles? what about ensembles for the other CORDEX regions, such as for example North America?

*These large ensembles consist all out of more than ten GCM-RCM combinations. For example, the ensemble of the EURO-CORDEX domain consists of 14 GCM-RCM combinations; 18 GCM-RCM combinations are available for CORDEX-Africa. North America contains as well a large ensemble of 13 GCM-RCM combinations for the 0.22° resolution but we did not want to list all the different CORDEX regions and the number of GCM-RCM combinations. In our submitted manuscript we mentioned EURO-CORDEX, CORDEX-Africa and MED-CORDEX but NA-CORDEX has indeed more GCM-RCM combinations at the 0.22° resolution. In the revised version we will therefore replace MED-CORDEX by NA-CORDEX. A detailed overview of the available ensembles over the different CORDEX regions can be found at the official CORDEX website: https://cordex.org/ and for each CORDEX domain there is a tab on this website with more information or a link to the website of that particular CORDEX domain.*

- line 51: the term "validating" is normally considered not very appropriate when comparing climate models and observations, in particular in cases like this one, where large uncertainties in observations are present. "evaluating" would be a more appropriate term.

*As suggested, we have changed "validating" to "evaluating".*

- line 56: Same as above. Replace validation with evaluation everywhere in the text.

*As suggested, we replaced "validation" with "evaluation" in the text.*

- lines 62-63: delete "...that are sparsely populated" since it is repetitive (you already said that in the first line of the period).

*As suggested, we removed it.*

- line 65: "more extreme values": more extreme than what? just use "extreme values"

*We agree and we changed it in the text.*

- lines 67-68: The comparison against different observational datasets is useful only to address the reliability of observational datasets and does not help solving the problem of the lack of an ensemble. Please reformulate.

*It is reformulated.*

- lines 70-71: Similarly as expressed in my major concerns, you cannot directly prove that similar biases in the two models are due to observational errors.
In principle, uncertainty in the observational data-sets allow to say that over certain areas the observations are more or less reliable and whether robust conclusions can be drawn in this case. Please reformulate this part.

*It is reformulated.*

- line 80: complemented by

*It has been corrected.*

- lines 83-88: I think that this part would be more appropriate for the introduction rather than for the methods.

*We agree, the text has been changed.*

- line 106-107: "The outer domain consists of the inner domain plus a coupling zone of eight grid points in each direction.": This holds true for both domains, right? eventually specify.

*Indeed, this is true for the domains of both RCMs. We specified this in the text so it is clear that we refer to both RCMs with this sentence.*

- Fig.1: Where did the authors take the information on the topography from? the upper limit of the colorbar of 3000m seems not reasonable for the area.

*The figure shows the values of the topography used in the regional climate model REMO [GTOPO30 global digital elevation model (DEM) 3 https://www.usgs.gov/centers/eros/science/usgs-eros-archive-digital-elevation-global-30-arc-second-elevation-gtopo30?qt-science_center_objects=0#qt-science_center_objects]. The explanation is added to the figure's caption.*

*We have increased the upper limit of the colorbar to the upper limit of the orography within the study area.*

- line 131: what is the vertical extension in meters of the domain of study for each of the two models?

*For REMO, with 27 levels, the top is approximately at 25 km height. The top of the uppermost gridbox is set equal to 0 hPa, but in reality the midpoint of the uppermost gridbox is ~25 km. ALARO-0 uses a vertically staggered grid and the top of the uppermost gridbox is also set equal to 0 hPa. The midpoint of the uppermost gridbox is situated at 67 km for a standard atmosphere.*

- lines 139-140: Correct into: "...at the boundaries, up to the 31st of December 2017."

*We corrected this sentence.*

- lines 143-147: Is there any reason why in the case of ALARO-0 one year can be considered enough for spin-up with respect to the 31 years considered for REMO? Please specify.

*Both RCMs are using a different soil model. The soil model used for REMO is using five layers with a mean rooting depth up to 5.7 m (Kotlarski, 2007), while there are only two layers in the ISBA model for ALARO-0. One year spin-up is enough for ALARO-0 since different variables reach their equilibrium after maximum one year. Most soil properties find their equilibrium after about one month. To reach an equilibrium state for the soil temperature and soil moisture, a warm spin-up period of ten years instead of thirty years was used for REMO. We will correct this in the text.*

- lines 149-150: This sentence, in the way it is expressed, is not properly correct. In fact, you do a comparison of model results only against the CRU, while then you compare the different observations among them. Please better reformulate this sentence according to the comparison you will decide to perform.

*We decided to add annual cycles of the different datasets and RCMs, thus this sentence should not be changed since in those new graphs the results of the RCMs are compared with the different datasets.*

- lines 151-152: Again, given your analyses you can not tell whether the bias of the models is due to the observational uncertainty. What you can eventually say is that high uncertainties do not allow to draw robust conclusions.

*It is reformulated.*

- lines 160-165: I do not manage to find the reference of New et al. 2002 in your paper. Are not there any more up-to-date publications discussing problems of the latest CRU releases? Also, do not you think that the New et al. 1999 publication is more general and it might also apply to other observational datasets rather than simply the CRU? Please consider that all your considered data-sets are somehow characterized by uncertainties (Flaounas et al., 2012;Gómez-Navarro et al., 2012).

*We checked and updated our references in this part of the text. Recently a new paper for the CRU data was published (Harris et al., 2020) and we updated our text taking this paper into account. Indeed, New et al. (1999) is rather describing general features about gridded datasets but they do focus on the first versions of CRU, that is why we mentioned this reference as well in the section about CRU. We agree that it is better to refer to more recent and concrete papers for the CRU dataset. Additionally, we will add a sentence in the general part about the reference datasets taking into account Gómez-Navarro et al. (2012). The study of Flaounes et al. (2012) (about the ECA&D gridded dataset over MED-CORDEX) is not general enough to be relevant for our text.*

- lines 168-171: what about quality of UDEL for other variables than precipitation?

*We added information about the variable temperature.*

- lines 176-179: Please, make clear that Hu et al. 2018 only investigated the most central part of your domain of study. Also, the same study states that GPCC underestimates all seasonal means, not only but especially in spring.

*Adaptations have been made in the text as suggested.*

- lines 181-186: The original resolution of ERAInterim is not 25 Km but approximately 80 km. If you used the data provided by the ECMWF at 25 Km, be aware that these are interpolated data. Please specify this in the text.

*As suggested, the explanation has been added to the text and adapted in Table 1.*

- lines 181-186: an additional question concerns your choice of using ERAInterim data interpolated at 25 km: why you do not directly download ERAInterim data already onto a 50 Km grid?

*We had ERA-Interim available at 25 km on our HPC infrastructure and a projection to the 50 km grid results to the same. The new graphs with annual cycles are not produced at the 0.50° resolution but at the resolution of each dataset and 0.22° for ALARO-0 and REMO.*

- lines 186-188: First of all avoid saying initial errors in the boundary conditions, since it generates confusion. Then, how should the comparison of temperature derived from ERAInterim with the one of the models help you determining what is the effect of errors in the boundaries? The only way to assess the effect of the boundaries on the RCM results is to drive the same simulations with different boundaries.

*We agree that it is confusing. To be certain about the effect of the errors at the boundaries, other boundaries should indeed be applied. We have deleted this part in the text.*

- lines 189-190: The outputs of an RCM are dependent (but not univocally determined) on the values of several variables with which the model is forced at its boundaries. These variables will have an effect on several model variables. The temperature of the model is not only dependent on the values of temperatures provided at the boundaries, but other variables play a role. The same holds true for precipitation. If the model is forced with wrong temperatures it is very likely, at least from a theoretical perspective, that both model temperature and precipitation will be both badly reproduced.

*We agree, the text at this line was deleted.*

- lines 198-199: First of all UDEL and CRU have the same 0.5 degree resolution. Also GPCC is available at such resolution. Reformulate this period in a more accurate way, considering the fact that the "upscale" is only necessary for the models outputs.

*We have changed "upscale" in the text as was suggested. The annual cycle graphs were created using the highest resolution of each dataset (0.50° for CRU and UDEL, 0.25° for GPCC and 0.80° interpolated to 0.25° for ERA-Interim).*

- lines 207-209: reformulate this period.

*We reformulated this part in the text.*

- line 219: seasonal means of

*We corrected this.*

-line 227: what do you mean by limited bias? better specify.

*We reformulated this part in the text.*

- lines 228-229: First of all, you start discussing annual means but you put the relative figures at the bottom row of your image: move them up. Then, in my opinion, according to the scale you use in your plots, it seems that in both cases the absolute bias exceeds 3C over a very extensive part of the domain and not only over mountainous regions. Maybe the scale you are using does not help to clearly distinguish which areas are above or below a certain threshold. Try to change your scale.

*We agree that the maps of annual means have to be placed at the top of the figure. It is indeed difficult to see the difference between each degree on the figure. We will change the scale.*

- lines 229-230: Also the REMO exceeds the 3C range, in particular in winter. Please reconsider your sentence.

*We included this REMO temperature bias a bit further in the text where we discuss the biases in the mountainous regions and say that REMO has a warm bias in winter over the Altai region. We agree that this might have been confusing and as suggested, the warm bias of REMO that exceeds 3 °C in winter over the north-western part of Mongolia has been added at this particular location in the text.*

- lines 230-231: not totally correct. In fact, the biases ,when considering the entire domain, are particularly pronounced for ALARO-0 mainly in spring, over the northern part of the domain. In winter the most pronounced bias seems to be the one of REMO for north-western Mongolia. For summer and autumn the biases for the two models present a very similar range. The same holds true when considering only the eastern half of the domain. Reformulate this part.

*We reformulated this part as suggested.*

- lines 231-233: Actually you should really emphasize that the two models seem to have a completely different pattern of the bias of temperature in winter: one shows a bipolar behavior between North and South, while the other between East and West, with a peak in warmer simulated temperatures over north-western Mongolia. I think that it would be really important for the authors (and a very nice opportunity) to better investigate the causes of the two different behaviours. This could give us some clue on model limitations in the simulations of temperatures over the region, that seems to be a general issue for climate models.

*As suggested, we will emphasize this different behavior of the models in the text. By including an additional subsection showing the yearly cycle of both temperature and precipitation of the observational datasets and model output over subdomains the reader gets more insight into these bias patterns.*

- lines 233-234: What do you want to evince from this? why Scandinavia and not another region? Also, how is the bias similar in the two cases?

*We moved this information to the discussion section. The climate in Scandinavia is similar to the climate in the northern part of the CAS-CORDEX domain. The reason why in both regions a warm bias is obtained for ALARO-0, is probably linked with a process that occurs in regions with a subarctic climate and not somewhere else. Deviations in snow related processes might explain the warm winter and cold spring temperature biases in the northern part of the domain and therefore we will add some additional information in the discussion part about this feature. We are currently investigating this.*

- lines 238-239: Important biases are present in MAM also for REMO, for some regions such as the Western fringes of the Tibetan Plateau. Also, for both models biases exceed 3C over a large part of the domain in MAM. Reformulate.

*As suggested, these sentences have been reformulated.*

- lines 239-241: What do you mean by limited? you mean that biases are not very pronounced in summer? reformulate.

*We reformulated this sentence as suggested.*

- lines 239-241: also for REMO there are warm biases, even though they are inherent to a smaller portion of the domain, in particular with respect to ALARO. Be more precise.

*We reformulated this sentence as suggested.*

- Fig 2: Beside my previous comment on the figure colorbar, the quality of the image could be further improved by reducing white spaces in between rows and moving the names of the seasons on the left side of the figures. Additionally, units should be added to the colorbars, that should also be moved: the colorbar of the bias should be positioned in between the two columns for the bias of REMO and ALARO.

*We decided not to change the location of the names of the seasons in the figure. By placing the names to the left side of the maps the maps would become smaller in order to fit the page. We want to present our figures as large as possible and that is why we structured it in this way. We will add the units to the colorbars and place the colorbar of the bias at the right side of the figure.*

- lines 248-249: The mentioned gradient is not very clear, in particular in summer.

*We removed this statement.*

- line 249: "The outcomes of both RCMs for the mean temperature agree well with the CRU data in autumn (SON)": That is not totally true. In fact, performances of REMO in terms of simulated seasonal climatologies are very similar for autumn, but also for spring and summer.

*We reformulated this sentence.*

- lines 254-255: what do you mean by "should be placed in perspective"? in which perspective? please reformulate this period.

*We reformulated this sentence to make clear that the uncertainty in observational gridded datasets is known to be larger at locations in mountainous areas.*

- lines 258-259: "it is clear from Table 2 that the strong cold bias during spring in the north for the ALARO-0 model has a larger negative impact on the spatially averaged bias than the warm bias during winter": I would avoid talking about "negative impact" of the bias over some region on the calculation of the spatial mean bias. Instead, you could say that the spatial bias is largely influenced by the pronounced negative/positive bias over specific regions.

*Thank you for the suggestion, we reformulated this sentence.*

- lines 264-267: "However, the biases during summer are ... due to the smaller spatial variability in temperature during summer". I think that this period is not very clear and needs to be reformulated, eventually considering additional analyses supporting your conclusions. First of all in summer, in the observations, you have less spatial variability (more accurate than smaller spatial range) than in the other seasons. This is evident from the figure, even though it would be nice if you could support such conclusion with a more quantitative analysis of the CRU spatial variability. Additionally (and most importantly), in your analyses you do not effectively demonstrate that a lower correlation is due to a lower spatial variability in summer. Why can it not be simply due to a worse agreement in the spatial variations between the models and the observational dataset?

*We agree that the sentence at line 264 is confusing and does not add any value, therefore we decided to remove this sentence. We have changed "smaller spatial range" into "less spatial variability". We will not include a more quantitative analysis of the CRU spatial variability to keep the document as concise as possible and since it is already visually clear from Fig. 2 that the spatial variability is smaller in summer. From Fig. 2 it is visually clear that the biases are lower in summer compared to winter and autumn, thus we assume that the lower spatial variability in summer is the reason for the lower correlation and not the worse agreement between the models and observational dataset.*

- lines 276-277: that is exactly one of the reasons why it would be better to consider the analyses per sub-regions.

*We agree and will take into account a subregional analysis.*

- lines 290-291: I think that your explanation on the reasons of a more negative bias for TMIN than for T2 is not exhaustive. Additionally, this needs to be moved to the discussion part.

*We will move this to the discussion section, where we can explain it exhaustively.*

- lines 297-298: "Following the main trend..": confusing, reformulate.

*We reformulated this sentence.*

- lines 299-301: "The warm minimum temperatures of the RCMs indicate that they underestimate the coldest diurnal temperatures or that the observational CRU dataset overestimates them." There are several issues in this period. First of all you need to reformulate your sentence because it is not the minimum temperature of the model that underestimates observation values but rather the model itself. Also, if the minimum temperatures are warmer than observations, it means that the model overestimates (and not underestimates) the coldest diurnal temperatures. Finally, from the comparison of model results against CRU you can only affirm that the models underestimate minimum diurnal temperatures. You can not prove that the observations overestimate them. The fact that CRU might overestimate them is a possibility, but still is not inherent to the behaviour of the model (nor it is evident from the figure you are commenting).

*We reformulated this sentence according to the suggestions.*

- lines 312-313: "except for the summer": why except if your are talking about annual values?

*We agree and reformulated this part of the text.*

- line 315: less good than what?

*We reformulated this part of the text and moved it to the discussion section.*

- line 323: you do not need to specify that temperature is a variable here

*We agree and, according to the remarks that were made for minimum temperature, we moved this sentence to the discussion section.*

- lines 323-324: You need to reformulate this sentence. In this case you have to specify that the negative TMAX bias is particularly remarkable in spring for the northern part of the domain, and also, to a less degree, in summer. In winter some other less extended parts of the domain, such as the north-eastern part, show a colder bias than REMO. In Autumn results are more similar between the two models.

*We agree and we reformulated this text part.*

- end of line 326: the cold bias

*We corrected the typo.*

- lines 326-328: Fig. 4 shows minimum temperatures. Then, how can we deduce from this figure that the bias in TMIN is due to maximum temperatures? please better explain and eventually reformulate this period.

*We referred to the wrong figure, it should be Fig. 6. This sentence is describing what was earlier mentioned: "specify that the negative TMAX bias is particularly remarkable in spring for the northern part of the domain". We moved the sentence up and rewrote it a bit so it is clear what we are trying to say.*

- line 342: "This means that ALARO-0 fails to reproduce the low nocturnal temperatures": This belongs to the discussion on minimum temperatures. Additionally, the model still fails in simulating warmer temperatures, despite the smaller bias when compared to TMIN.

*As suggested, we moved the last paragraph of this section to the discussion section and we explain more clearly that ALARO-0 fails to reproduce temperature in general (including mean, minimum and maximum temperature) in the northern part of the domain.*

-lines 344-346: You should discuss about minimum temperature in the appropriate section.

*As suggested, we moved the last paragraph of this section to the discussion section.*

- When you comment the maps of the bias, try to discuss the different seasons from up to down, consistently with the figures.

*We agree.*

- lines 364-366: This part should be moved to the discussion section.

*As suggested, we moved this text to the discussion section.*

- lines 365-366: why however? also, you did not discuss until this point the uncertainty of CRU: how can you claim that the reason for the wet bias is due to the observations?.

*We agree that "However" at the beginning of this sentence is not suited here. It was not our intention that this sentence was interpreted as a shortcoming of the CRU dataset since this is the results section. We wanted to express that it is known from the observations that the amount of precipitation is low in certain regions as seen in Fig. 8 (< 5 mm/month), not that CRU contains precipitation amounts that are too low (this follows in the discussion). We reformulated this sentence, to overcome the confusion.*

- lines 370-372: By whom is the bias turned into something else in summer? and how?

*We reformulated this sentence to make it clear that we talk about summer, when the East Asian Monsoon takes place.*

- lines 372-376: It would be nice if you could perform the analyses of the seasonal cycle to support your conclusions. This would make your evaluation more complete and exhaustive, while at the same time allowing to effectively confirm or deny your conclusions.

*Thanks for this suggestion. We agree and did an analysis of the annual cycle over multiple subdomains.*

- lines 390-391: "The dry biases for ALARO-0 in Table 5 are thus caused by the simulation of systematically less precipitation than the precipitation amounts in the CRU data.": Reformulate. It is obvious that if the model underestimates precipitation, it simulates less precipitation than observations.

*We reformulated this sentence. We intended to say that there is no region that has a strong dry bias which is compensated with a wet bias in another subregion. This differs from the finding of temperature where the strong warm bias in the north is partly compensated by a cold bias in the southern part of the domain.*

- line 393: systematically

*We corrected the typo.*

- lines 392-394: "The lower accuracy of simulated precipitation is due to the fact that precipitation is less systematic affected by land cover and topography compared to temperature": First of all that is quite a strong assumption given the extent and heterogeneity of the domain you are considering. Additionally, you did not perform (at least it is not reported in the paper) any analysis that supports your conclusion.

*We agree, we did not perform an analysis on this topic but it is known that it is harder to simulate the spatial pattern of precipitation compared to temperature (Kotlarski et al., 2014) due to the reason we mentioned.*

- lines 400-404: This is incorrect. In fact Russo et al. 2019 showed that uncertainty in observations is high over the north-eastern part of the domain, not that CRU overestimates the diurnal temperature range over the region.

*We agree and will reformulate this text part.*

- line 404: Why hence?

*We agree that this is an incorrect cause-consequence structure and we will reformulate it.*

- lines 404-406: Again, how can you surely state that the model underestimates values of the diurnal temperature range due to higher observation values?

*We will rewrite this part.*

- lines 407-408: why Czech Republic? what happens in other regions?

*We agree that it would be better to refer to literature over Central Asia instead of referring to literature over EURO-CORDEX where ALARO-0 and REMO were already evaluated. We will refer to Russo et al. (2019) who obtained similar findings.*

- lines 420-422: This is just an assumption that needs to be proven. Models develop their answer that is, to a certain degree, independent from the boundaries. To test your hypothesis, one easy experiment that could be conducted is to use different boundaries and compare the results.

*We agree and we will remove this.*

- line 425: "They related this warm bias already to shortcomings in the simulation of snow": this means that they explained the bias differently than with the boundary effect as you explained in the lines from 423 to 425.

*Ozturk et al. (2012) explained the bias indeed with a shortcoming in the simulation of snow cover. We will remove the part about the boundary effect.*

- lines 430-431: "Hence, we conclude that the warm forcing is the main reason for the warm bias over Eastern Russia during winter.": I further have to highlight that you can not make such conclusion, until you do not test different boundaries.

*We agree and we will remove the part about the boundary effect.*

- lines 435-436: As before, it would be nice if you could do the analyses of the seasonal cycle since you mention it for the interpretation of your results.

*We agree and we will add annual cycle graphs as mentioned before.*

- lines 440-442: How the fact that for Belgium there is some correlation between warm bias and cloud cover representation could explain the same for northeastern CAS. You could do some analyses on cloud cover to support your conclusion.

*We mentioned this study over Belgium since it is the only study that investigated the relationship between temperature and cloud cover for ALARO. We agree that we cannot draw strong conclusions from this and that this previous paper only gives a clue that cloud cover might be one of the reasons why the temperature is not well estimated. Cloud cover is thus only one out of the many possible reasons, which should be further investigated. In the meantime we did some analysis on cloud cover and we will include our findings to the new version of the manuscript.*

- lines 443-445: "Both could be due to too much cloud cover": according to whom? In theory it could be due to any reason.

*We agree and we investigated this further to say something about it in the discussion.*

- lines 448-451: These considerations are important: it would be nice to put them in a more objective context. Additionally, you say that New et al. show that CRU underestimates temperatures for Russia. Then you talk about Western Russia. If you state that temperatures from CRU are not good for Russia, then they can not be good for a part of it and bad for the rest. Reformulate.

*We reformultated these sentences based on the additional analysis over the subregions.*

- Fig. 10,11: To make the discussion easier I would suggest to plot the maps of the differences between different observational data-sets together, using the spread of the observations among the different data-sets. In this way you can easily know which areas are more reliable and which are not.

*We agree and produced new figures.*

- lines 465-467: the less reliable observations do not explain the bias, rather they do not allow to draw any conclusion.

*We reformultated these sentences.*

- line 471: "...winter and overestimate it during." During what?

*The word "summer" is missing, we added it to the text.*

- lines 482-484: what happens when you compare ALARO-0 with the other data-sets over the entire domain?

*The precipitation of ALARO-0 is for most grid points within the range of the different gridded datasets during the different seasons. When averaging over the complete domain, then the output of both RCMs is within the range of the spread between the reference datasets for the different seasons. However,*

*there are some subregions where the precipitation of ALARO-0 and/or REMO is lower or higher than the observational spread for a specific season. For example both RCMs slightly underestimate precipitation in summer over West Central Asia. We will add this information in the updated manuscript.*

- lines 484-486: you mention two gridded data-sets: to which data-sets are you referring here? please better specify.

*We are referring to GPCC and MW, these are observational gridded datasets. ERA-Interim is a reanalysis product, so we do not refer to it as an observational gridded dataset. We reformultated this sentence and we will make sure that this is clear throughout the complete manuscript.*

- lines 486-489: again, you can claim that the bias is relative to the employed boundaries only performing a new simulation with different boundaries. Also, how can you be sure that ERA-Interim overestimates specific humidity?

*We will reformulate these sentences since it was not intended to say that the boundary conditions of ERA-Interim affected our results. We just wanted to point to the similarities between the ERA-Interim data and the output of the RCMs. We agree to remove the suggestion of the overestimation of the specific humidity as we did not investigate it.*

- lines 489-490: You are claiming this from the field means I guess. I think that plotting the maps of the bias of the models against all the different observational data-sets might help the discussion of your results.

*We claim this based on Fig. 8, 11, S1 and S2 where the spatial patterns between ERA-Interim and REMO are visually very similar, while the patterns of ALARO-0 are similar to GPCC and MW. We agree that it can help to plot the maps of the bias of the models against the different observational datasets, however this will make the manuscript long.*

- lines 489-490: How do ERA and REMO parameterize precipitation since you mention that they do it differently? Specify.

*For REMO these specifications are included in Table S1. We will refer to this table at the end of this sentence and we will add that ERA-Interim uses a convection scheme modified from Tiedtke (1989) by Bechtold (2008; 2014) and the cloud scheme is based on Tiedtke (1993) with modifications made made by Forbes and Tompkins (2011), Forbes et al. (2011) and Tompkins et al. (2007) (https://www.ecmwf.int/en/research/modelling-and-prediction/atmospheric-physics). The similarities between ERA-Interim and REMO for precipitation are thus probably due to the fact that both use a modified scheme that is based on Tiedtke (1989). We did not further investigate this.*

- lines 492-493: "This difference between ALARO-0 and REMO is related to the 3MT cloud microphysics scheme of ALARO-0": where did you demonstrate this?

*We did not demonstrate or investigate this but it is an assumption since this is known to cause differences (Giot et al., 2016). We reformulated this statement so it is clear that it is an assumption that should be further investigated in the future.*

- lines 496-497: Again, this is hard to affirm simply using three observational data-sets. The authors have to acknowledge the low number of observational data-sets. As I mentioned in many previous comments I personally think that it would be better to approach the differences between the observational datasets in terms of reliability rather than determining who is more correct.

*We agree, we will mention the low number of observational datasets. We will reformulate the text so we focus on the reliability of the observations and the complications for our evaluation. It was not our intention that it looks like it is a research on which reference dataset is the best one.*

- Fig. 11: In the colorbar of the bias, are units percentage? with respect to what? Please specify.

*Indeed, the unit was wrong and we corrected it to the unit percentage. In Fig. 11 the precipitation of CRU is compared with the other datasets ERA-Interim, MW and GPCC. The relative values were obtained by dividing the difference by the value of CRU as already mentioned in section 2.4 Analysis methods. In order to make this clear in Fig. 11, we will specify this in the figure captation.*

- lines 501-502: Not completely true. Specify that, as evinced from your maps, the wet bias in ERA (with respect to the other 3 data-sets) is only relative to the eastern part of the domain.

*We reformulated this sentence as suggested.*

- line 520: "that that". Correct.

*We corrected this.*

- lines 536-537: "REMO simulates the precipitation fairly well and ALARO-0 performs very well." How can you state that their performances are good?

*The simulated precipitation of the RCMs is for most regions most of the time within the observational spread. We have clarified this in the manuscript.*

- lines 539-540: "The warm temperatures obtained with REMO ... can be linked with the dry and wet bias in winter and spring respectively." Why and how can they be linked?

*We agree that they cannot be linked without doing an in depth study on how they are exactly linked. We reformulated this sentence.*

- lines 540-541: In which way the link between temperatures and precipitation should strengthen your hypothesis of a delay by REMO in the simulation of snow cover? Can you be more specific?

*This was an assumption, we reformulated this sentence.*

- lines 545-546: "The persistent warm bias over Pakistan and Northern India of both RCMs can be explained by the persistent underestimation in simulated precipitation over this region by both RCMs.": how can you state that given your analyses?

*We agree that the warm temperature bias cannot be explained by an underestimation in precipitation. We reformulated this sentence.*

- lines 547-550: You refer to the fact that your results are within the ranges of models for other domains, but then you only mention the results of Kotlarski et al. for Europe. You need more references.

*We agree, we will add some papers that were recently published over parts of the CAS-CORDEX region.*

- lines 562-563: That is arguable, given your analyses. How do you define an acceptable range?

*An acceptable range is within the range of the observational spread. We will reformulate this so it is clearer.*

- lines 565-567: You cannot state this, until you force the model with different boundaries and you conduct an analysis of snow cover (what you can eventually do for snow cover is to reference to the evidences from other studies).

*We agree and we removed this sentence.*

- Table 1: This table is not easily readable. Could you find a way to make the distinction between the different data-sets a bit clearer?

*We will add a light gray background to the odd rows, so that the distinction between the information of the different rows is more clear.*

---

## Author Comment (AC3) · 30 May 2020

**Author response to the review of Anonymous Referee #3**

*Referee 3, thank you for reviewing our manuscript. We are delighted to take all of your comments into account to improve the manuscript. Our answers on all questions, suggestions and remarks can be found on the next pages. Firstly, we summarize the major changes we will make to the revised version of the manuscript based on the comments of the different reviewers:*

- *We will include an analysis of the annual cycle over the subdomains as defined by the IPCC6 report (Iturbide et al., 2020) which are situated within the CAS-CORDEX domain. The results, both for the RCMs and the gridded datasets, for the mean temperature and precipitation are given in Fig. A1 and A2.*
- *We will approach the differences between the gridded datasets in a different way. The spread between the gridded datasets (Fig. A3) will be used as an estimate of the uncertainty.*
- *We will improve the discussion section by describing which model features can explain the significant biases that were obtained over certain regions.*
- *We will include some additional recently published scientific papers in our revised manuscript e.g. Harris et al. 2020; Wang et al. 2020; Zhu et al. 2020.*

[Figure]

*Fig. A1: Annual cycle of the mean temperature (°C) over different subdomains.*

[Figure]

*Fig. A2: Annual cycle of the precipitation (mm month$^{-1}$) over different subdomains.*

[Figure]

*Fig. A3: Spread in mean temperature between the gridded datasets CRU, MW and ERA-Interim.*

This paper describes the results of two models (REMO and ALARO-0) simulations over CORDEX Central Asia domain. Authors compared simulated temperature and precipitation climatology and concluded that both the models are capable to reproduce CAS climate. Reading the paper I had an impression it is a kind of technical report but not a scientific manuscript suitable for GMD. I do not see any science by describing how large biases in models are without any reasonable explanation where they come from. Authors took models which were tuned for Europe, implemented them for CAS, obtained huge biases and concluded: "That's it." Therefore I would recommend the manuscript for publication only in case it will be substantially revised.

Major points

1. Analysis (but not referring to other models results) of model biases is required. Where they come from? Is it large scale atmospheric circulation or local processes, e.g. atmosphere – land heat/moisture exchange? In this sense it would be interesting to look in mean sea level pressure (MSLP) biases. For example, the warm temperature DJF bias as well as huge overestimation of DJF precipitation in REMO could be because of underestimation of Siberian High.

*We will improve our discussion section taking this comment into account. We are currently investigating possible causes that could explain the obtained biases (e.g. cloud cover, snow cover) and we will include our findings in the revised version of the manuscript.*

2. The models show quite a substantial differences in biases. Considering the eastern part of CAS it is clearly seen that in cold seasons REMO simulates 2m temperature much better then ALRO. Furthermore ALRO results with almost 10K bias over quarter of the domain are inacceptable. The opposite is seen for precipitation which is simulated by ALRO better. Based on these results authors can take heat (moisture) fluxes as well as heat (moisture) transports from both the models (assuming that "better" model reproduces better fluxes (transports)) and try to analyze which of them leads to produce mentioned above biases.

*We will improve our discussion section by trying to explain the obtained biases.*

3. For better understanding I would also recommend to analyze the climatological annual cycle of some quantities, like temperature, precipitation and heat fluxes at least for the eastern part of the domain (from Mongolia to the east), where the biases are really large. For such a big domain with a plenty of climatological zones Taylor diagrams are more a kind of speculation. E.g. in case the climatological temperature varies from +30C in the South to -30C in the North spatial correlation will be high with any kind of model.

*We agree with this remark. To gain insight into the model's performance and limitations we will include in the revised version an analysis of the annual cycles based on monthly means for five subdomains. However, we still find it valuable to do the evaluation (and make the Taylor diagrams) over the complete CAS-CORDEX domain since this region is set as a standard domain. Many papers use currently different subdomains over Central Asia and due to the small differences in the definition of these domains they applied the results cannot be equally compared. Standard regions such as the CORDEX and IPCC regions avoid this problem, that is why we will keep the scores over the complete domain in our manuscript.*

4. Authors should have a more deeper look into previous studies done with the same models. In particular ones were done with REMO. Since REMO existence (more then 20 years) there are many papers with REMO simulation results over regions partially included in CAS, e.g. whole the northern part: Niederdrenk, 2013 (PhD), Niederdrenk et al., 2016 (Clim. Dyn.), Sein et al., 2014 (Tellus); south-eastern part: Xu et al, 2018 (Clim. Dyn.).

*We took these papers into account and will refer to some of them in our updated text.*

5.Authors claim that some of the biases come from the ERA-Interim forcing. That is quite an ambitious conclusion, in particular for Siberian continental climate. This conclusion has to be proven with some additional simulations. It is not a big deal to take a lateral boundary conditions from some of the global climate model, to simulate ca. 10 years and to look if the large scale biases are similar or not. I think with available computer recourses it should be just 3-4 working days.

*Indeed we cannot claim that the biases are due to the ERA-Interim forcing without investigating this feature. We removed the text parts where we are claiming this.*

Minor points

L. 23: I do not think that with large scale 8-10K 2m temperature biases and more then 100% precipitation biases over quarter of the model area both models reproduce climate "reasonably well".

*For the precipitation we get sometimes more than 100% due to the very low amounts as discussed in the text. For example, if there is 1 mm of precipitation and the models estimate 2 mm monthly precipitation, the relative precipitation bias is huge. Therefore, we added the absolute differences as well in the supplementary material. Additionally, there is the spread between the gridded datasets. From the newly created annual cycles it can be seen that the RCMs are mostly within the spread of the gridded datasets.*

L.24-25: It has to be done in this work, but not postponed to the unclear future

*This would make the paper too long.*

L.35: Even being a not an expert in CORDEX and even for CORDEX domains mentioned by authors, I know much more works based on multi-model regional simulations. E.g. Africa: Paxian et al. (JGR-Atmos, 2016); Mediterranean: Damaraki et al. (Clim.Dyn, 2019), Gaertner et al. (Clim. Dyn, 2017), Soto-Navarra et al. (2020, Clim.Dyn).

*Since there are quite some publications about multi-model regional simulations we made a selection, discussing all of them is not in the aim of this paper that handles about CAS-CORDEX where there are no multi-model regional simulations available. Including all of the other domains would make the paper too long but we will add some of these references.*

L.61: "Absence of reliable observational data sets". Over China and Russia? Maybe 20 years ago "yes" (describing CRU data authors site work from 1999), but at the present time it sounds at least strange.

*We agree, Harris et al. (2014) is indeed better to refer to for the current information about CRU and we will add as well the Harris et al. (2020) reference which was published after we submitted our manuscript. We included the 1999 reference since this one describes the strategy and methodology of CRU.*

2. Methods. See above (L.35) Central America: Cabos et al. (2019, Clim. Dyn.), Southeast Asia: Zhu et al. (2020, TAC), Arctic: Akperov et al. (2019, Global and Planetary Change; 2018, JGR)

*We will at least refer to Zhu et al. (2020) in our updated paper.*

L.94: I would remove word "sea". In a middle school I have learned that Black, Caspian Red and Baltic seas are seas, but it is hard to say that they are barely covered with CAS domain.

*We agree, the Black Sea, Caspian Red Sea and Baltic Sea are seas in the CAS-CORDEX domain. We removed "sea" and replaced it with "open ocean" since we wanted to stress that the domain mainly exists out of landmass.*

L.96: Before claiming it, authors should "google" a word "HighResMIP". In the framework of this project there are many global climate model simulating climate on 25 km resolution, i.e. the same resolution as authors use for their regional simulations.

*We added the reference of Haarsma et al. (2016) with information about HighResMIP to the text.*

L.106 and in other places: I would suggest to use not "coupled zone", but "sponge zone". Forcing a regional model with reanalysis has nothing to do with coupling.

*To overcome confusion we will use "relaxation zone".*

L.129: But what about dynamical core itself? Please explain at least in the way it is done for ALRO above, i.e. special discretization, advection (e.g. in ALRO it is based on semi-Lagrangian algorithm and what about REMO?)

*See table S1 in the supplementary materials where these specifications are mentioned. We opted not to mention all of them in the text because of the readability and to keep the text as concise as possible.*

L.137-138: What about upper boundary? Which height does it have? 10hPa? 50hPa?

*The upper boundary of ERA-Interim configures for 60 levels in the vertical, with the top level at 0.1 hPa (https://www.ecmwf.int/en/elibrary/8174-era-interim-archive-version-20).*

L.202: As far as I know almost all the atmospheric models (including REMO and ALADIN) provide direct output of Tmax and Tmin which are obtained every model time step. Why not to use them directly?

*This is correct, Tmax and Tmin were used directly from the model output of REMO and ALARO-0. We reformulated our text to avoid confusion.*

3. Results: As I mentioned in "major points", not only seasonal means but also climatological annual cycle for the quantities averaged over different areas has to be included.

*We agree, we have added the annual cycles.*

L.229: Exceeded. How much does it exceeded? On the plot I can only see that it is larger then 10K.

*It depends on the subregion or the location. In winter the maximum bias obtained for REMO and ALARO-0 at one particular point is respectively 16.8 °C and 19.2 °C when compared to CRU.*

L.234: What has Scandinavia to do with Mongolia? They have completely different climate. In the same way REMO group can write: Paxian et al. (2016) showed a strong precipitation bias over Guinea in Africa. Maybe that is also a reason of REMO prcip. bias over East Siberia?

*We agree and we will add additional information.*

L238: Actually the strongest cold bias over Europe in REMO is at Spring. It is not visible in most of the papers, because mainly they show DJF and JJA only.

*Yes, that is true. We included all seasons to report our results as honestly as possible.*

L.360 (Fig.8) Relative difference in mm/month? I think it should be in (%)

*Indeed, we corrected this.*

To all the figures with biases: For the biases I would avoid linear color bar and extend it for larger values. E.g. for the temperature something like: 0,1,2,3,5,7.5,10,12.5,15 and for precip. (%) 0,10,20,30,50,75,100,125,150,200

*We will reduce the classes of the color scales in order to improve the readability of the figures and we will use a non-linear color bar as suggested.*

L.405: What the Czech Republic has to do with Central Asia? Do they have similar climate? I have here the same claim as at L.234. Authors should provide arguments which has something to do with CAS and not speculations like: we have warm bias in Mongolia, because in French Polynesia is to rainy.

*We agree.*

L.414: I would not say that up to 10K large scale temperature bias is something which is VERY well

*Biases over 10 °C are mainly found over the regions where the reference datasets are less reliable (see spread reference datasets in the newly created maps). We agree that we should formulate this differently e.g. the results are within the range of uncertainty of the used gridded datasets. Additionally, for some parameters significant biases are present over parts of the domain for some seasons and cannot be explained by the uncertainty in the gridded data. For example, the ALARO-0 RCM has a large positive temperature bias in winter over the northern part of the domain. The REMO model has difficulties in reproducing the observed precipitation patterns over the orography of Central-Asia. We agree that the biases observed in this study should be kept in mind when presenting future projections. We find it therefore important to publish an exhaustive evaluation study. In this evaluation study we saw that the main patterns are modelled correctly and therefore we concluded that we can move on towards climate projections. We will add to our conclusion that these large biases should be kept in mind when looking to the future projections. Additionally, to deal with the biases in impact studies, several bias adjustment methods have been tested within the AFTER project and the most suitable method will be applied before simulations for impact studies are done with these climate data. It is not in the scope of this evaluation study to explain the details about bias adjustments and impact modelling but to avoid misunderstandings we will add that bias adjustment is one of the possibilities when mentioning that the RCMs can be used for future projections.*

L.423: "..assigned to this forcing". As it was mentioned above (Major points), before speculating about it, please do some simulations with different forcing.

*We agree, we cannot claim that the biases are due to the ERA-Interim forcing without investigating this feature. We removed the text parts where we are claiming this.*

L.433: "Ozturk et al. . . ., but they did not explain it." And? If Ozturk did not explain it, it is over? Why don't you try to explain it in your manuscript.

*We will improve our discussion section by trying to explain the obtained biases.*

L.428, 448, etc. New et al. (1999). You discuss present climate and present observational data set citing a work from 1999? There is a quite a big difference between the number of observations before 1999 and now.

*Indeed there is a difference between the number of observations in the beginning of our evaluation period (1980) and the end (2017). New et al. (1999) is rather describing general features about gridded datasets, that is why we mentioned this reference. We agree that it is better to refer to more recent and concrete papers for the CRU dataset. Recently a new paper for the CRU data was published (Harris et al., 2020) and we updated our text, taking this paper into account.*

Fig. 11: I think should be MW, but not WM. As well as (%), but not mm/month

*Indeed, we corrected this.*

Conclusion: In the scientific sense conclusion is very poor simply describing how large model biases are only. The only one "explanation" of their origin is "models are good, but observations are bed", based on results obtained more then 20 years ago, in 1999. I would suggest to authors to bring more "scientific analysis" into the manuscript considering comments written above. Maybe it will bring the paper from "technical report" to "scientific manuscript".

---

## Referee Report (RR1)

**Review of "Evaluation of regional climate models ALARO-0 and REMO2015 at 0.22 resolution over the CORDEX Central Asia domain" by Top et al. 2020**

**General Comments**

The authors surely put some efforts in trying to answer my comments on their previous version of the manuscript. Nevertheless I do not think that all these comments were exhaustively considered. In my opinion the paper still suffers from a series of major issues that need to be carefully addressed before it may be considered for publication for Geoscientific Model Development.

- The quality of the text and the structure of the manuscript are surely the points that have received more attention by the authors, but still some parts need revision. In particular you should check for consistency among the different subsections. For example in the subsections of the methods you specified what you used ERAInterim for, but you did not do the same for other data-sets such as GPCC. Another similar example can be found in the results subsection about DTR. Here you discuss the table with spatial means for maximum temperatures but not for minimum temperatures. Check for such inconsistencies throughout the text and correct them.

- You are not very accurate in the specification of the model behaviour and when you discuss the maps of the bias. I found a lot of inaccuracies in the text and I invite you to review it accordingly. Here a couple of examples:

  - **At the annual scale, the bias of the minimum temperature ranges mostly between -3C and 3C for REMO and between 0C and 5C for ALARO-0 (Fig. 5).**
    For REMo annual biases exceed the absolute value of 3 C over several areas such as Mongolia and the Himalayas. For Alaro a large part of the domain has a negative bias exceeding -5C in some case.

  - l. 258-261 **Based on Fig. 3, both RCMs perform best during autumn and the spatial correlation is lowest during summer for ALARO-0 while, the biases during summer are smaller than during winter and spring for both RCMs (Table 2 and Fig. 2)**
    From figure 2 you cannot say that the biases in summer are smaller than in winter and spring for all the points of the domain.

  - l. 452-454 **Fig. 2 and 4 show that for most parts of the domain the mean temperatures of ALARO-0 and REMO are lying within the range of spread between the reference datasets during autumn. From this we conclude that both**

**RCMs simulate temperatures in autumn within the range of observational uncertainty.**

This is not true for the entire points of the domain. When you propose such conclusions I invite you to first compare directly the map of the spread with the one of the bias (for example plotting their differences).

- My main concern is that despite my previous comments, even though you added an analyses of the seasonal cycle for sub-region in the new version of the manuscript, you did not conduct the same analysis for the mean bias, spatial correlation, standard deviation and RMSE. The information of the mean bias calculated over the entire domain intuitively makes no sense, as already highlighted in my previous report. You should conduct the analyses of the bias per sub-domain too. One interesting thing that I would suggest you to do is to consider mean absolute bias instead of the bias (for both the entire domain and sub-regions, since spatial biases might compensate each other). Additionally, also Taylor diagrams should be calculated for every sub-region. For this I also think it would be important for you to independently check the values of the spatial correlations you obtained, since they seem to be too high given the spatial patterns of the bias. Since you use pre-defined functions (in R) for calculating the Taylor diagrams, I think it would make sense to double check the correctness of the results, independently. Finally, the plots of the seasonal cycle should be improved. In particular it was impossible to understand the ones drawn for temperature.

- The discussion part does not always result very clear and I would suggest you to carefully revise it while modifying it in consideration of my new comments.

- As a final remark, it seems that the authors are a bit too positive about the models performance for the region. I would suggest them to try to be more objective in their conclusions. Maybe new analyses might help in this sense. In my personal opinion the models results cannot be considered reliable over a large part of the domain. It is true that over some areas there is an issue with the poor reliability of observations, but over some regions the main issue is still the model. Having a bias above 10C does not make the model reliable. This is for example the case of temperatures simulated by ALARO over the north-western and north-central part of the domain, both in winter and in summer. Appropriate bias correction methods could be used to make the model more in-line with "reality", but this is not inherent to model evaluation and should be made clear in the text.

---

## Referee Report (RR2)

Review of "Evaluation of regional climate models ALARO-0 and REMO2015 at 0.22 resolution over the CORDEX Central Asia domain" by Top et al. 2020

The authors improved the quality of the paper compared to previous versions. In particular, the presented analyses are now more appropriate for an evaluation of models performance. Nonetheless, the quality of the manuscript is still not sufficient to be granted publication. In particular, the description of the results is very poor, full of inconsistencies, not exhaustive and detailed and not consistent throughout the different sections. I invite the authors first to describe each of their figures, extensively providing all the details, and only then summarizing the text. Also, please check that you proceed in the description of the results in the same way for each of the figures. Below you can find a detailed list of some of the main issues of the paper. Be aware that errors are not limited to the mentioned examples, but all the manuscript needs a thorough review. The paper is at a good point and a thorough and patient review should allow for its publication.

- l. 57-58: by whom was the simulation performed?
- l. 61: A new paper by Russo et al. 2020 with COSMO-CLM is available for Central Asia, published on GMD.
- 1. 102: comparable with what? you probably mean among each others in a coordinated framework?
- Fig. 1: Specify in the caption that all points with orography higher than 3000 m are set with the same height of 3000m.
- l. 120-122: A regionalization was applied and not the subdomains. Reformulate.
- l. 148-150: line repeated twice. Also, make example of which parameters you refer to.
- l. 153, "to produce an equilibrium for the soil temperature and soil moisture": does not sound good. Reformulate into something like: to let the model reach an equilibrium state for ...
- l. 156-157: it should be "were compared". Check for consistency of verbs tense throughout the text
- 1. 161 "since all gridded datasets are characterized by uncertainties": I would avoid such statement, since it is quite obvious. On the other hand I would say that you consider different observational data-sets for the calculation of given metrics, for assessing the reliability of the results.
- Section 2.3: make clear that you use CRU as reference and additional data-sets for assessing reliability of observations over different areas.

- l. 166: was used
- 1. 183-185: period a bit confusing, you need to reformulate. Basically you state that Hu et al. found that GPCC is better than CRU and MW for the inner part of Central Asia CORDEX domain. Than you say that precipitation is underestimated in montainous regions, but are you still referring to the inner part of the domain? Also, you state that GPCC underestimates precipitation in general: but for which region? globaly? for Central Asia? do you have a reference?
- 1. 199: as for GPCC, where ERAInterim generally overestimates precpitation? references?
- 1. 216: by computing the difference between maximum and minimum
- l. 220: specify that you calculate spatial MAE.
- l. 225: over the points of the domain
- l. 225-227: The Taylor Diagrams does not represent, but rather includes calculations of the spatial correlation between model and reference data. In your case you can eventually state that the Taylor diagrams are used to estimate spatial agreement between the climatological means of the different data-sets, by considering different metrics.
- 1. 239: maybe it would be nice to indicate the different rows corresponding to annual and different values.
- l. 242-244: I would discuss the results of the table together for annual and seasonal values, after introducing the results of Fig. 2. In any case, you have to provide more details on the results of Table 2 for yearly values, not simply mentioning that they are reported.
- 1. 252: Very pronounced biases are also present for REMO, in particular in winter, over the north-eastern part of the domain
- 1. 253: the reader should know which areas you are referring to: the map of the subdomains should be introduced directly in the main text and not in the supplementary part.
- 1. 255: as you did for ALARO, mention that the bias of REMO exceeds 10C in this case.
- l. 256-258: that's it for spring? please quantify the values of the bias in the two cases, consistently with other seasons. Also, what happens over other regions?
- 1. 258-259: as above, provide estimate of biases also in summer
- l. 260: On the contrary

- l. 260-261: Why do you stop providing estimates of the biases? This applies to all sections and figures.
- l. 260-261: what about summer biases of ALARO over Mongolia and North-eastern China?
- 1. 265-268: I would move this paragraph before the previous one, at the beginning of the description of Fig. 2.
- Fig. 3: How can you explain very high spatial correlation values for the entire domain when the values of correlation are lower in the case of all subdomains?
- 1. 275: how do you define best here? in terms of which metrics? as the points being closer to observations? be more specific. For example for REMO RSV, the model is better in summer than autumn in EEU; RSV is better in summer than Autumn over West Siberia with respect to all metrics; for East Siberia in summer, values of RSV are closer to 1 than in autumn; for WCA, summer correlation is slightly better than in autumn; for the TIB region, both summer and spring results have better correlation than in autumn. Only for ALARO results seem to be better in autumn than for other seasons, given the considered metrics. The description of the Taylor diagrams is not very accurate and precise throughout all the text, making the interpretation of the results very difficult for the reader. I suggest the authors to thoroughly review the comments of the Taylor diagrams, trying in a first essay to write down all possible information and only then summarizing the results. I can understand that this is complicated given the large amount of figures, but this is absolutely necessary given the current state of the manuscript.
- 277-278: not true. What do you consider as "WELL" for normalized standard deviation? For me the good performance of ALARO in EEU in Winter amd REMO in winter over WSB are quite arguable. Also, I would not be very convinced about the goodness of the winter results for the ESB domain for ALARO as well as winter results in WCA and autumn results in TIB for both models.
- 1. 279-280: how would the limited bias explain the higher SD values? biases could be higher but more homogeneous. Reformulate this period.
- 1. 282-285: This part is not exhaustive and accurate and needs to be reformulated: first of all a similar bipolar behaviour is found for different seasons for both models, not only for summer. One good example is SON spatial biases for the entire domain for REMO and the case of both REMO and ALARO in winter. Here it would be opportune to use the fact that you have small mean biases but large MAE for supporting the conclusions on the fact that the biases are the results of compensation effects. Also, poorer performance than what? Given your conclusions about the

performance of ALARO in summer, why your reasoning should work for summer and not for winter, where we also have a very well pronounced bipolar behavior of the bias?

- 1. 290: specify that you are discussing spatial variability
- 1. 289-293: I would reformulate this part: you basically say that REMO is better than ALARO in simulating spatial variability, except autumn and winter for some domains. Then you say that ALARO better captures spatial patterns except winter and summer over some subdomains. This sounds a bit contradictory.
- Fig. 3: specify in the text that the different Taylor diagrams have different scales.
- 1. 300: extremely well? it does not seem so for WCA and ESB; Also performance do not seem very good in November for ALARO over WSB.
- Fig. 4: I suggest you to split the figures in 3 figures, respectively for T2, TMIn and TMAX, since at the moment the current figure does not allow to appreciate differences in the different cases. Maybe you can introduce figures for TMIN and TMAX in the supplements.
- 1. 326: what happens for the orographically more complex regions? what are the biases?
- 1. 327: Specify that the bias of REMO over the Eastern part of the domain reaches 15C in winter.
- 1. 329-330: what about all the other parts of the domain in spring and summer?
- 1.345-354: Quantify values of bias and MAE.
- 1. 352-352: this does not seem the case for autumn and annual values of ALARO and for summer for REMO
- fig.6: I think you are not giving the same importance in the discussion of Fig 6 as for Fig. 3.
- Fig. 6: one general question concerning all Taylor diagrams: do you have any clue why you generally get high values of correlation for the entire domain, when in all subdomains you obtain smaller correlations?
- 1. 361: not exact: REMO better also for EUU in summer and winter and in WCA for spring.
- l. 361: specify spatial variability
- 1. 366: what happens in the Tibetan plateau? what is the magnitude of these biases?

- 1. 365-370: not only here, but also for other figures, fix an order for the discussion of the figures (for example from top to bottom) and follow it throughout the text
- 1.367-368: cold bias over northern part of the domain is present in all seasons, except DJF.
- 1. 364-371: reformulate and extend all the period. Some parts of the domain are never mentioned. What happens for example in the Himalayas and the Arabian Peninsula? exhaustive description of summer is missing.
- 1. 378-379:Specify that in the case of ALARO the bias exceeds 7C.
- 1. 376-385: you should extend the part describing the tables, discussing for example biases and MAE, giving indications on the fact that in some case biases are the result of compensation.
- 1. 380: not very accurate. In summer REMO is better for the Tibetan plateau in terms of MAE, while ALARO is better in winter for the EEU domain
- 1. 381: how can you conclude, from the sentence before where you state that the 2 models are better in autumn, that ALARO simulates TMAX poorly in any season? actually there are 3 cases where ALARO is better than REMO: DJF in EEU and MMA and SON in TIB. Anyhow, in many cases the results of the 2 models are very similar and the MAE is very close to 1. Importantly, you again omit to specify what is good and what is bad, in terms of the given metrics.
- 1. 403: it would be better to have a map of the bias of DTR. This could help your discussion that now is too generic based only on the maps of TMIN and TMAX.
- 1. 406: the model does not restore its balance, since it is also in equilibrium in winter. Reformulate.
- l. 414-415: Where? In the other sections you started commenting the figures and then the tables. Why you change this now?
- 1. 415-420: what about annual values in table 5? what about the spread of observations?
- l. 417-418: please quantify all the biases you mention.
- l. 419: are you sure the Tibetan plateau can be classified as a monsoon region?
- l. 418-420: which figure are you commenting now? If you are discussing Fig. 11, actually it does not seem that the bias is smaller but greater in summer, at least for ALARO.

- 1. 423-424: It is not totally true that the largest biases are present over extremely dry areas. One example is Northern India in Summer, presenting a remarkable dry bias despite observations are characterized by highest precipitation values.
- 1. 422-423: specify that when you talk about low precipitation you are referring to the observations.
- 1. 425: actually over Northern China in REMO the bias exceeds 5mm/month.
- l. 433: "is also present for REMO": why also?
- l. 435-436: which model are you referring to?
- l. 438 and l. 447: greater than -2mm/month
- l. 450-451: how can you claim that the spatial patterns are well represented by ALARO if for each subdomain, despite WCA, the considered metrics present relatively poor values (especially in terms of correlation)?
- 1. 449-452: why do you not discuss the observations?how do the different data-sets compare to each others? this is something that you should do when commenting all Taylor diagrams.
- You can extend the discussion of Fig. 10, consistently with the other subsections.
- 1. 476-479: Please specify the cases when the bias exceeds the spread of observations. At the moment it seems like the spread of observations is smaller than the bias in almost all the domain and seasons. Is that correct? So your figures would confirm that evinced biases are more inherent to the model than to observations, over almost all the domain in all seasons. One more elegant way (and probably more useful for your goals) to determine those points where the spread of observation exceeds the bias, is by plotting a map of the bias (in the supplements) with a point in correspondence of those points where the spread is larger than the biases.
- 1. 482-484: be more precise. It is not clear what you want to express. The observational spread is significantly high over complex-orography regions and not over the entire north-eastern part of the domain.
- 1. 486-488: also REMO bias exceeds the spread of the observations over large parts of Mongolia and Northern China.
- $\bullet$ l. 489-490: reformulate
- l. 497-498: This is also true for EEU in autumn
- l. 531: Also true over mountainous regions of north-eastern part of the domain.

- l. 536-538: but also in summer, at least for ALARO
- l. 542-543: Reference needed.
- l. 542-542: what are these processes?
- l. 543-544: why should it be? reformulate. Better specify what do you mean by shift in the annual cycle?
- 1. 546: how can you state that the bias increases when the snow-covered region expands? Have you directly analyzed snow cover in the two models? If what you affirm is true, this should be a feature of both models. However, winter biases are different in the two cases.
- 1. 545-546: are you sure about the warm bias appearing in the North during autumn when the snow appears over this region? Again, did you base these statements on some analysis of simulated snow cover? warm bias is very limited to a very small eastern part of the northern domain in ALARO.
- 1. 548-549: I would argue against your conclusions, based on the fact that also REMO is characterized by a very warm bias in winter over the North-eastern part of the domain. I do not think that based on your analyses you can raise strong conclusions on the driver of the bias over the northern part of the domain in both models. For sure I would not state that REMO does not encounter the same problem.
- 1. 570: wouldn't Eastern Europe be more consistent with your discussion than Western Russia?
- l. 571: Can you better specify what means acceptable? I would emphasize that the MAE in this case is smaller than the MAE between ERAInterim and CRU.
- 1. 575-577: How can you state this without a map of the bias in DTR? can you be sure that for all the points of the domain RCMs produce a smaller DTR in all seasons?

---

## Author Response (AR2)

**List of relevant changes**

*Dear Editor,*

*Thank you for giving us the opportunity to improve our paper. Based on the comments of the reviewer we have made the following major changes hoping to satisfy the reviewer with the more detailed text:*

- *We have added mean absolute error (MAE) as an extra score.*
- *We have conducted the same analysis for the five subregions as was done for the complete CAS-CORDEX region, including: mean bias, MAE, spatial correlation, standard deviation and RMSE. In this way more detailed information is provided to the reader.*
- *Based on the additional analysis over the five subregions, we have substantially rewritten the result and discussion sections (Sect. 3 and Sect. 4), taking into account consistency over the different subsections.*
- *Moreover, we have updated the figures and text of the precipitation section.*
- *Finally, we have revised the conclusion and abstract based on the more detailed information given by the analysis over the subregions.*

*The implementation of the changes requested by the reviewer has led to a significant extension of the manuscript. The level of detail is quite high, in accordance with the request for a more detailed analysis, even though this was not our intention when submitting the first version of our paper. We feel that this level of detail is required to address all of the concerns that were raised.*

**Author response to the review of the Anonymous Referee**

*Dear reviewer,*

*Thank you for your helpful comments. As you will read in the point-by-point answers, we have performed an analysis over the five subregions, similar to the complete CAS-CORDEX domain. As a consequence, our revised manuscript has a significantly higher level of detail.*

General Comments

The authors surely put some efforts in trying to answer my comments on their previous version of the manuscript. Nevertheless I do not think that all these comments were exhaustively considered. In my opinion the paper still suffers from a series of major issues that need to be carefully addressed before it may be considered for publication for Geoscientific Model Development.

- The quality of the text and the structure of the manuscript are surely the points that have received more attention by the authors, but still some parts need revision. In particular you should check for consistency among the different subsections. For example in the subsections of the methods you specified what you used ERAInterim for, but you did not do the same for other data-sets such as GPCC. Another similar example can be found in the results subsection about DTR. Here you discuss the table with spatial means for maximum temperatures but not for minimum temperatures. Check for such inconsistencies throughout the text and correct them.

  *We had another careful look at the different subsections and made them consistent. We added the used variables of the MW dataset that were not mentioned in an explicit way. Regarding which GPCC variables were used for this dataset, we stated: "In addition, the GPCC has no similar dataset for other variables and thus, only precipitation can be validated with this dataset." For ERA-Interim a more detailed explanation is provided since there are more options e.g. hourly, daily or monthly data, while the other datasets contain monthly data as highest time resolution for the studied variables. We mentioned in the text that monthly data is used for each gridded observational dataset, so it should be clear for the reader which data we used.*

- You are not very accurate in the specification of the model behaviour and when you discuss the maps of the bias. I found a lot of inaccuracies in the text and I invite you to review it accordingly. Here a couple of examples:

  - **At the annual scale, the bias of the minimum temperature ranges mostly between -3C and 3C for REMO and between 0C and 5C for ALARO-0 (Fig. 5).**
    For REMo annual biases exceed the absolute value of 3 C over several areas such as Mongolia and the Himalayas. For Alaro a large part of the domain has a negative bias exceeding -5C in some case.

    *This sentence should be placed in its context. The sentences following this sentence are indicating where the -3 °C- 3 °C range for REMO and the 0°C - 5 °C range for ALARO is exceeded, including Mongolia and the Himalayas. We understand this might not be clear when reading it for the first time and revised this paragraph to make sure it cannot be interpreted incorrectly.*

  - l. 258-261 **Based on Fig. 3, both RCMs perform best during autumn and the spatial correlation is lowest during summer for ALARO-0 while, the biases during summer are smaller than during winter and spring for both RCMs (Table 2 and Fig. 2)**
    From figure 2 you cannot say that the biases in summer are smaller than in winter and spring for all the points of the domain.

    *We agree, we meant that the biases are lower on average. We will revise this sentence.*

- l. 452-454 **Fig. 2 and 4 show that for most parts of the domain the mean temperatures of ALARO-0 and REMO are lying within the range of spread between the reference datasets during autumn. From this we conclude that both RCMs simulate temperatures in autumn within the range of observational uncertainty.**
  This is not true for the entire points of the domain. When you propose such conclusions I invite you to first compare directly the map of the spread with the one of the bias (for example plotting their differences).

  *The mean temperatures of the RCMs lying within the range of spread is indeed not true for all points of the domain, that is why we explicitly mention "for most parts of the domain". We added new maps with the difference between the absolute bias and the spread in the supplementary material. Based on these maps we can describe in more detail what we intended to say and in which areas the models do not perform that well.*

- My main concern is that despite my previous comments, even though you added an analyses of the seasonal cycle for sub-region in the new version of the manuscript, you did not conduct the same analysis for the mean bias, spatial correlation, standard deviation and RMSE. The information of the mean bias calculated over the entire domain intuitively makes no sense, as already highlighted in my previous report. You should conduct the analyses of the bias per sub-domain too. One interesting thing that I would suggest you to do is to consider mean absolute bias instead of the bias (for both the entire domain and sub-regions, since spatial biases might compensate each other). Additionally, also Taylor diagrams should be calculated for every sub-region. For this I also think it would be important for you to independently check the values of the spatial correlations you obtained, since they seem to be too high given the spatial patterns of the bias. Since you use pre-defined functions (in R) for calculating the Taylor diagrams, I think it would make sense to double check the correctness of the results, independently. Finally, the plots of the seasonal cycle should be improved. In particular it was impossible to understand the ones drawn for temperature.

  *We added the Taylor diagrams for the subregions including spatial correlation, standard deviation and RMSE. We agree with your comment that the mean bias over the entire domain does not provide much insight. We calculated instead this metric over the subregions, as it can give an insight into the regional dependence of the bias. We agree that the mean absolute error (MAE) is a better metric and we added MAE for the entire CAS-CORDEX domain and the subdomains.*

  *We calculated the spatial correlation, standard deviation and RMSE in two independent ways: once based on the formulas described by Kotlarski et al. (2014) and once with the pre-defined Taylor function in R. We obtained the same values for both approaches. The spatial correlations are based on the Pearson correlation method in the pre-defined Taylor function and can be described by the PACO formula (Kotlarski, et al.2014) as was mentioned in the manuscript. Calculations of the correlations based on the built in Pearson correlation in R and calculations with the PACO formula, both showed the same values as the values plotted with the built in Taylor function, so we are sure that the correlations are that high over the full domain.*

  *Figure 4 with the annual cycles of mean, minimum and maximum temperature was an incorrect figure, the lines of ERA-Interim for minimum and maximum temperature were incorrect. This might be the reason for not being able to understand the figure. We tried to put as much information on one figure to reduce the length of the manuscript and to be as concise as possible.*

- The discussion part does not always result very clear and I would suggest you to carefully revise it while modifying it in consideration of my new comments.

  *We revised this section.*

- As a final remark, it seems that the authors are a bit too positive about the models performance for the region. I would suggest them to try to be more objective in their conclusions. Maybe new analyses might help in this sense. In my personal opinion the models results cannot be considered

reliable over a large part of the domain. It is true that over some areas there is an issue with the poor reliability of observations, but over some regions the main issue is still the model. Having a bias above 10C does not make the model reliable. This is for example the case of temperatures simulated by ALARO over the north-western and north-central part of the domain, both in winter and in summer. Appropriate bias correction methods could be used to make the model more in-line with "reality", but this is not inherent to model evaluation and should be made clear in the text.

*We reformulated the conclusions based on the information of the subdomains.*

*Additionally, we used an incorrect transformation of units for the CRU precipitation data in the results of the previous version of the manuscript. We have therefore updated the evaluation of the precipitation.*

[revised manuscript text omitted]

| | ALARO-0 | REMO |
|---|---|---|
| **projection resolution** | Lambert conical projection 0.22° | rotated pole 0.22° |
| **horizontal spatial discretisation** | spectral on collocated grid | $2^{nd}$ order finite differences on staggered C-grid |
| **vertical coordinate levels** | 46 hybrid levels | 27 hybrid levels |
| **temporal discretisation** | semi-implicit semi-Lagrangian | leap-frog with semi-implicit correction and Asselin filter, semi-Lagrangian advection |
| **time step** | 450 s | 120 s |
| **convective scheme** | 3MT scheme | Tiedtke with modifications after Nordeng and Pfeifer (Pfeifer, 2006) |
| **radiation scheme** | The Action de Recherche Petite Echelle Grande Echell (ARPEGE) Calcul Radiatif avec Nebulosité (ACRANEB) scheme for radiation | Morcrette et al. (1986) and Giorgetta and Wild (1995) |
| **turbulence vertical diffusion** | A pseudoprognostic turbulent kinetic energy (pTKE) scheme (i.e., a Louis-type scheme for stability dependencies, but with memory, advection, and autodiffusion of the overall intensity of turbulence) | Louis-type with a higher order closure scheme for the transfer coefficients of momentum, heat, moisture and cloud water within and above the planetary boundary layer. Eddy diffusion coefficients are calculated as functions of the turbulent kinetic energy. |
| **cloud microphysics scheme** | A statistical sedimentation scheme for precipitation within a prognostic-type scheme for microphysics. | The cloud microphysical scheme by Lohmann and Roeckner (1996). |
| **land surface scheme** | The Interaction Sol-Biosphère-Atmosphère (ISBA) scheme | Based on the surface runoff scheme (Hagemann, 2002), inland glaciers (Kotlarski, 2007), and vegetation phenology (Rechid, 2009) |
| **institute** | RMIB-UGent | HZG-GERICS (https://remo-rcm.de/) |

[Figure]

5    **Figure S1: IPCC6 subdomains projected on the CAS-CORDEX region.**

[Figure]

**Figure S2: Difference between absolute value of bias and observational spread for the variable mean temperature (°C) of RCMs REMO and ALARO-0.**

[Figure]

10 **Figure S3: Difference between absolute value of bias and observational spread for the variable minimum temperature (°C) of RCMs REMO and ALARO-0.**

[Figure]

**Figure S4: Difference between absolute value of bias and observational spread for the variable maximum temperature (°C) of RCMs REMO and ALARO-0.**

[Figure]

[Figure]

**Figure S52: Absolute difference between the average seasonal and annual CRU precipitation (mm month⁻¹) and the precipitation simulated by REMO and ALARO-0 over the 1980-2017 period.**

**Table S2: Climatological mean CRU precipitation (mm month⁻¹) for the 1980-2017 period over the CAS-CORDEX domain and subdomains, and absolute biases (mm month⁻¹) and MAE (mm month⁻¹) against those CRU means for the RCMs (REMO and ALARO-0), and the other reference datasets (ERA-Interim, MW and GPCC).**

|  | DJF | MAM | JJA | SON | Annual |
|---|---|---|---|---|---|
| CRU | 30.38 | 43.46 | 87.03 | 47.72 | 52.26 |
| REMO - CRU | -1.23 | 1.33 | -19.81 | -5.24 | -6.26 |
| ALARO - CRU | -2.74 | -4.98 | -21.54 | -4.40 | -8.45 |
| ERA-Interim - CRU | -2.90 | 1.25 | -9.78 | -4.61 | -4.01 |
| MW - CRU | -9.06 | -12.37 | -24.03 | -13.06 | -14.66 |
| GPCC - CRU | -9.43 | -13.77 | -23.20 | -14.11 | -15.15 |

|  | EEU | | | | | WSB | | | | | ESB | | | | |
|---|---|---|---|---|---|---|---|---|---|---|---|---|---|---|---|
|  | DJF | MAM | JJA | SON | Annual | DJF | MAM | JJA | SON | Annual | DJF | MAM | JJA | SON | Annual |
| CRU | 34.91 | 34.16 | 55.26 | 45.62 | 42.51 | 22.74 | 27.99 | 51.53 | 35.94 | 34.60 | 11.13 | 22.10 | 72.28 | 29.62 | 33.90 |
| REMO - CRU | 4.18 | 6.83 | 4.02 | 4.31 | 4.84 | 3.73 | 7.10 | 6.72 | 4.96 | 5.64 | 3.33 | 14.01 | 5.73 | 6.35 | 7.38 |
| MAE REMO CRU | 5.62 | 7.86 | 8.64 | 5.61 | 5.62 | 6.23 | 9.83 | 10.73 | 7.69 | 7.31 | 5.18 | 15.00 | 11.37 | 9.09 | 8.88 |
| ALARO - CRU | 7.45 | 3.95 | 5.50 | 8.33 | 6.29 | 4.55 | 0.90 | -2.21 | 6.05 | 2.30 | 3.91 | -0.26 | -13.52 | 6.08 | -0.99 |
| MAE ALARO CRU | 8.04 | 5.72 | 10.58 | 8.73 | 7.18 | 5.93 | 4.66 | 8.11 | 7.59 | 4.85 | 5.24 | 5.14 | 18.38 | 8.65 | 5.96 |
| ERA-Interim - CRU | 4.49 | 6.53 | 5.75 | 3.98 | 5.19 | 3.99 | 7.46 | 8.35 | 5.50 | 6.34 | 3.26 | 12.61 | 8.19 | 9.07 | 8.30 |
| MAE ERA-Interim CRU | 5.18 | 6.59 | 6.62 | 4.46 | 5.33 | 4.80 | 8.47 | 9.41 | 6.10 | 6.97 | 4.08 | 12.88 | 11.21 | 9.71 | 8.84 |
| MW - CRU | -3.69 | -2.33 | -3.69 | -2.89 | -3.14 | -1.75 | -1.47 | -4.13 | -2.00 | -2.34 | -0.42 | -3.42 | -9.59 | -2.72 | -4.05 |
| MAE MW CRU | 4.49 | 3.07 | 4.44 | 4.49 | 4.49 | 3.48 | 3.69 | 6.10 | 3.48 | 3.48 | 2.09 | 4.42 | 11.04 | 2.09 | 2.09 |
| GPCC - CRU | -8.21 | -5.23 | -4.05 | -5.19 | -5.65 | -2.70 | -3.19 | -1.81 | -2.81 | -2.63 | -0.81 | -4.59 | -6.57 | -3.72 | -3.94 |
| MAE GPCC CRU | 8.82 | 5.68 | 5.85 | 8.82 | 8.82 | 4.88 | 5.02 | 4.70 | 4.88 | 4.88 | 2.38 | 5.15 | 8.24 | 2.38 | 2.38 |

|  | WCA | | | | | TIB | | | | | CAS-CORDEX | | | | |
|---|---|---|---|---|---|---|---|---|---|---|---|---|---|---|---|
|  | DJF | MAM | JJA | SON | Annual | DJF | MAM | JJA | SON | Annual | DJF | MAM | JJA | SON | Annual |
| CRU | 33.18 | 37.52 | 16.74 | 18.45 | 26.46 | 8.12 | 17.73 | 48.56 | 15.02 | 22.45 | 22.60 | 32.34 | 64.75 | 35.50 | 38.88 |
| REMO - CRU | 5.77 | -3.59 | -3.20 | 3.24 | 0.53 | 21.07 | 34.40 | 15.23 | 28.07 | 24.70 | 6.55 | 12.45 | 2.47 | 6.98 | 7.12 |
| MAE REMO CRU | 17.57 | 16.96 | 7.87 | 8.51 | 11.32 | 24.04 | 39.38 | 32.92 | 30.72 | 30.47 | 10.85 | 18.88 | 18.13 | 12.80 | 13.56 |
| ALARO - CRU | -0.71 | -1.75 | -3.00 | 1.61 | -0.96 | 2.15 | 6.37 | 6.85 | 5.64 | 5.26 | 5.04 | 6.14 | 0.74 | 7.82 | 4.93 |

| | | | | | | | | | | | | | | | |
|---|---|---|---|---|---|---|---|---|---|---|---|---|---|---|---|
| MAE ALARO CRU | 11.24 | 11.63 | 11.06 | 8.09 | 9.13 | 7.83 | 16.36 | 32.96 | 12.83 | 16.29 | 8.31 | 12.75 | 19.82 | 11.69 | 11.41 |
| ERA-Interim - CRU | 6.90 | 10.79 | 12.85 | 7.02 | 9.41 | 4.76 | 20.81 | 30.60 | 10.98 | 16.86 | 4.88 | 12.37 | 12.50 | 7.61 | 9.37 |
| MAE ERA-Interim CRU | 10.46 | 14.20 | 15.02 | 8.85 | 11.13 | 7.88 | 23.20 | 39.94 | 13.36 | 19.98 | 6.85 | 14.00 | 17.27 | 9.50 | 11.08 |
| MW - CRU | -1.22 | -2.98 | -0.33 | 1.20 | -0.83 | 1.11 | 0.53 | 4.14 | 3.07 | 2.21 | -1.28 | -1.25 | -1.75 | -0.84 | -1.28 |
| MAE MW CRU | 9.75 | 9.57 | 5.00 | 9.75 | 9.75 | 6.13 | 9.50 | 20.30 | 6.13 | 6.13 | 4.56 | 6.08 | 11.10 | 4.56 | 4.56 |
| GPCC - CRU | 0.08 | -2.50 | -1.21 | -0.38 | -1.01 | -0.74 | -3.08 | -2.11 | -0.30 | -1.57 | -1.65 | -2.65 | -0.92 | -1.89 | -1.77 |
| MAE GPCC CRU | 10.44 | 8.73 | 5.03 | 10.44 | 10.44 | 5.88 | 9.81 | 20.44 | 5.88 | 5.88 | 5.82 | 6.52 | 10.64 | 5.82 | 5.82 |

[Figure]

**Figure S6: Difference between absolute bias and observational spread for the variable precipitation of RCMs REMO and ALARO-0.**

[Figure]

35 **Figure S7: Absolute difference between the average seasonal and annual ERA-Interim precipitation (mm month$^{-1}$) and the precipitation simulated by REMO and ALARO-0 over the 1980-2017 period.**

**Table S32: Overview of the identifiers on the ESGF data platform (data node: esgf1.dkrz.de) of the used ALARO-0 and REMO RCM climate data.**

| Data | Identifier | PID |
|---|---|---|
| **ALARO-0** | | |
| precipitation | cordex.output.CAS-22.RMIB-UGent.CNRM-CERFACS-CNRM-CM5.historical.r1i1p1.ALARO-0.v1.mon.pr | / |
| temperature | cordex.output.CAS-22.RMIB-UGent.CNRM-CERFACS-CNRM-CM5.historical.r1i1p1.ALARO-0.v1.mon.tas | / |
| minimum temperature | Not available on the ESGF platform. Data can be downloaded with the key "userGMDpaper1" from: https://cloud.meteo.be/s/gRP2NFSfAWJas4g | / |
| maximum temperature | Not available on the ESGF platform. Data can be downloaded with the key "userGMDpaper1" from: https://cloud.meteo.be/s/8YEg4LY9DmX4EGF | / |
| **REMO** | | |
| precipitation | cordex.output.CAS-22.GERICS.ECMWF-ERAINT.evaluation.r1i1p1.REMO2015.v1.day.pr | hdl:21.14103/2ecffe86-b5e4-359c-8c34-e7152de17a43 |
| temperature | cordex.output.CAS-22.GERICS.ECMWF-ERAINT.evaluation.r1i1p1.REMO2015.v1.day.tas | hdl:21.14103/bf8468cf-b15c-3a20-ae42-4c42b14e749c |
| minimum temperature | cordex.output.CAS-22.GERICS.ECMWF-ERAINT.evaluation.r1i1p1.REMO2015.v1.day.tasmin | hdl:21.14103/74aa90a5-c99b-35f9-888e-acc0115dfc4d |
| maximum temperature | cordex.output.CAS-22.GERICS.ECMWF-ERAINT.evaluation.r1i1p1.REMO2015.v1.sem.tasmax | hdl:21.14103/a72e5ea1-533d-3685-b04d-5e4ab162e065 |

---

## Author Response (AR3)

**Reply to the review of "Evaluation of regional climate models ALARO-0 and REMO2015 at 0.22 resolution over the CORDEX Central Asia domain" by Top et al. 2020**

**List of relevant changes**

*Dear Editor,*

*Thank you for giving us the opportunity to improve our paper. Based on the comments of the reviewer we have made the following major changes.*

*We describe the figures in detail for the description of the mean temperature in section 3.1. Since the subsequent figures in sections 3.2 and 3.2 have exactly the same format as the ones in section 3.1, we do not repeat the detailed description of the figures since the reader should be familiar with them by then.*

*We have made sure that sections 3.1, 3.2 and 3.3 all follow the same structure, i.e.*
1. *description of the figure and the table (the table complements the figure) then discuss in this order:*
    1. *the performance of the annual mean,*
    2. *the performance of the seasonal mean,*
    3. *eventually the performance over steep orography*
2. *presentation and description of the Taylor diagram*
3. *presentation of the annual cycle.*

*Additionally, we added the abbreviations of the subregions to the Taylor diagrams and we made the font larger of the values on the axis.*

*There is one citation Nikulin et al. (2012) that contained errors in the names, probably due to a wrong copy. We have corrected it.*

*In the supplementary material we corrected and added the identifiers of the used ALARO-0 data that is now available on ESGF.*

**Author response to the review of the Anonymous Referee**

*Dear reviewer,*

*Thank you for your helpful comments.*

The authors improved the quality of the paper compared to previous versions. In particular, the presented analyses are now more appropriate for an evaluation of models performance. Nonetheless, the quality of the manuscript is still not sufficient to be granted publication. In particular, the description of the results is very poor, full of inconsistencies, not exhaustive and detailed and not consistent throughout the different sections. I invite the authors first to describe each of their figures, extensively providing all the details, and only then summarizing the text. Also, please check that you proceed in the description of the results in the same way for each of the figures. Below you can find a detailed list of some of the main issues of the paper. Be aware that errors are not limited to the mentioned examples, but all the manuscript needs a thorough review. The paper is at a good point and a thorough and patient review should allow for its publication.

• l. 57-58: by whom was the simulation performed?

*The simulations were performed by the Met Office Hadley Centre (MOHC). The sentence is reformulated and a more suitable reference has been added.*

• l. 61: A new paper by Russo et al. 2020 with COSMO-CLM is available for Central Asia, published on GMD.

*We refer to it in the revised manuscript.*

• l. 102: comparable with what? you probably mean among each others in a coordinated framework?

*Indeed, this sentence can be improved. We now write: "In order to obtain simulations that allow for coordinated intercomparisons"*

• Fig. 1: Specify in the caption that all points with orography higher than 3000 m are set with the same height of 3000m.

*We specify this in the revised manuscript.*

• l. 120-122: A regionalization was applied and not the subdomains. Reformulate.

*We agree that it is confusing to use domain and subdomains, while we did only runs over one domain. It is indeed better to consistently use the word subregion instead of subdomain to refer to the smaller areas that we discuss in more detail. We now write: "In the present paper, the CAS-CORDEX domain was further subdivided into five subregions ...". Subdomain has been replaced by subregion throughout the text.*

• l. 148-150: line repeated twice. Also, make example of which parameters you refer to.

*Thank you for pointing it out. The repeated line has been removed. The parameters were already summed up in the sentence that follows: "These include sea surface temperatures (SSTs), surface roughness length, surface albedo, surface emissivity and vegetation parameters."*

• l. 153, "to produce an equilibrium for the soil temperature and soil moisture": does not sound good. Reformulate into something like: to let the model reach an equilibrium state for …

*We agree and now write: "... REMO was spun-up for 10 years to allow the model to reach an equilibrium state for the soil temperature and soil moisture ..."*

• l. 156-157: it should be "were compared". Check for consistency of verbs tense throughout the text

*Good point. The rule is here that the work we performed is written in the past tense, while the description of the data sets is in the present tense. We have changed one sentence: "... annually averaged values for temperature and precipitation were compared with different reference datasets..." We checked for consistency of verbs tense throughout the text.*

• l. 161 "since all gridded datasets are characterized by uncertainties": I would avoid such statement, since it is quite obvious. On the other hand I would say that you consider different observational datasets for the calculation of given metrics, for assessing the reliability of the results.

*This is indeed an obvious statement. We now write: "A multitude of datasets were considered to assess the reliability of the gridded observational temperature and precipitation (Gómez-Navarro et al., 2012)."*

• Section 2.3: make clear that you use CRU as reference and additional data-sets for assessing reliability of observations over different areas.

*We made clear in section 2.4 that CRU is used as reference. However, we understand that it is worth to mention it in section 2.3 as well. Therefore, we added a sentence in 2.3.1: "In present paper, this data set is used as the reference while the spread of the data in all of the data sets is used to assess the reliability over the different areas."*

• l. 166: was used

*Thank you, we follow the rule mentioned under your comment on l. 156-157. We now write: "Monthly values of minimum, maximum and mean near surface air temperature and precipitation were used in the current study."*

• l. 183-185: period a bit confusing, you need to reformulate. Basically you state that Hu et al. found that GPCC is better than CRU and MW for the inner part of Central Asia CORDEX domain. Than you say that precipitation is underestimated in montainous regions, but are you still referring to the inner part of the domain? Also, you state that GPCC underestimates precipitation in general: but for which region? Globaly? for Central Asia? do you have a reference?

*We do indeed still mean that the overall underestimation in precipitation and underestimation in precipitation in mountainous regions by GPCC was found for the subregion defined by Hu et al. (2018). This period has been reformulated in the revised manuscript based on the suggestions.*

• l. 199: as for GPCC, where ERAInterim generally overestimates precpitation? References?

*Here, we meant that ERA-interim overestimates precipitation globally, especially over mountainous regions based on the reference of Sun et al. (2018).*

• l. 216: by computing the difference between maximum and minimum

*Indeed, this can be stated more clearly. We now write: " ... is calculated for each grid point by computing the difference between the maximum value and the minimum value of the different reference datasets, and this for every 3-month period (season) averaged over the 1980-2017 period." We moved this sentence as well to the end of the section.*

• l. 220: specify that you calculate spatial MAE.

*Indeed, we have to specify that it is the spatial MAE. We now write "The climatological means, biases, and mean absolute errors (MAE) were spatially averaged to obtain one mean value over the complete domain and each of the subdomains, respectively."*
*We removed the sentence "Additionally, the mean absolute error (MAE) was calculated to account for compensating errors." since it is obvious.*

• l. 225: over the points of the domain

*Good point, we added the sentence: "These metrics are computed over all grid points of the CAS-CORDEX domain." and now write: "averaged over the domain".*

• l. 225-227: The Taylor Diagrams does not represent, but rather includes calculations of the spatial correlation between model and reference data. In your case you can eventually state that the Taylor diagrams are used to estimate spatial agreement between the climatological means of the different data-sets, by considering different metrics.

*We have reformulated the sentence: "These diagrams supplement the bias analysis by visualizing in a concise way information about the spatial correlation, the centered root mean square error (RMSE) and the ratio of spatial variability (RSV) between the model and the observational dataset (Taylor, 2001)."*
*We have removed the sentence "In this study the Taylor diagrams represent the spatial pattern correlation between model and reference data, which is obtained by calculating correlations across the grid points of the CAS-CORDEX domain." since it doesn't add any non-trivial information to the text.*

• l. 239: maybe it would be nice to indicate the different rows corresponding to annual and different values.

*Here we think it is sufficiently clear when looking at the figure.*

• l. 242-244: I would discuss the results of the table together for annual and seasonal values, after introducing the results of Fig. 2. In any case, you have to provide more details on the results of Table 2 for yearly values, not simply mentioning that they are reported.

*As mentioned in reply to your general comments, we have put a bit more order in the structure of section 3. First the figure and table are introduced, then the annual averaged performance is discussed, followed by summarizing the seasonal averaged performance.*
*We added a summary of the annual results from Table 2: "The biases and MAE of the annual mean temperature are very comparable between ALARO-0 and REMO (Table 2), with small biases and MAEs that are only slightly larger than the spread of the observational data sets."*

• l. 252: Very pronounced biases are also present for REMO, in particular in winter, over the north-eastern part of the domain

*Indeed, we now write: "On the seasonal timescale, biases over larger areas are mainly pronounced in winter (DJF) and spring (MAM). In particular both models locally show strong biases in the north-eastern part of the domain for winter with values ranging up to 15 °C. Additionally, ALARO-0 shows strong negative biases up to -15 °C during spring in this area."*
*We removed the sentence: "In winter the most pronounced bias is found for REMO over the north-western part of Mongolia in the Altai mountains, resulting in a large MAE of 3.40 °C over the ESB domain." since it does not add any substantial information anymore.*

• l. 253: the reader should know which areas you are referring to: the map of the subdomains should be introduced directly in the main text and not in the supplementary part.

*The figure has been added to the main text of the revised manuscript.*

• l. 255: as you did for ALARO, mention that the bias of REMO exceeds 10C in this case.

*As mentioned in the answer to comment l. 252 we now include in the text that REMO has biases up to 15 °C in the northeast.*
*Additionally, we now write: "These large biases are reflected by the values in Table 2 for the northern subregions EEU, WSB and ESB for ALARO-0 and the ESB subregion for REMO."*

• l. 256-258: that's it for spring? please quantify the values of the bias in the two cases, consistently with other seasons. Also, what happens over other regions?

*Spring is partly treated together with winter, since the main bias in ALARO extends from winter to spring. We prefer to keep it like that since splitting the two would make the paper longer.*

• l. 258-259: as above, provide estimate of biases also in summer

*In fact, all the estimates are in Table 2. This is precisely why Table 2 is added; to avoid that the paper becomes (even) longer.*

• l. 260: On the contrary

*The correction is done.*

• l. 260-261: Why do you stop providing estimates of the biases? This applies to all sections and figures.

*The purpose of the text is not to provide the biases by numbers. Otherwise, the text would become even more lengthy. We only provide them to describe the main deficiencies here. For instance for summer, both models behave within the range of the observations, so we do not provide the numbers.*

• l. 260-261: what about summer biases of ALARO over Mongolia and North-eastern China?

*We find that they are rather small compared to the others and prefer to not extend the text too much. See our reply to your previous point.*

• l. 265-268: I would move this paragraph before the previous one, at the beginning of the description of Fig. 2.

*Given the structure as we proposed it in the reply to your general comments, we prefer to keep it here. Otherwise it would feature before the description of the annual performance while it describes the performance of the seasons.*

• Fig. 3: How can you explain very high spatial correlation values for the entire domain when the values of correlation are lower in the case of all subdomains?

*Positive and negative terms can cancel each other out. When computing the spatial correlation over the CAS domain, biases are more likely to have an opposite sign (canceling each other out) compared to the subregions. Not surprisingly, the models simulate the global climate better than the regional ones. This is what is meant with: "On the other hand, the Taylor diagrams for the subregions illustrate how scores calculated over the complete CAS-CORDEX domain can hide underlying regional trends."*

• l. 275: how do you define best here? in terms of which metrics? as the points being closer to observations? be more specific. For example for REMO RSV, the model is better in summer than autumn in EEU; RSV is better in summer than Autumn over West Siberia with respect to all metrics;

for East Siberia in summer, values of RSV are closer to 1 than in autumn; for WCA, summer correlation is slightly better than in autumn; for the TIB region, both summer and spring results have better correlation than in autumn. Only for ALARO results seem to be better in autumn than for other seasons, given the considered metrics. The description of the Taylor diagrams is not very accurate and precise throughout all the text, making the interpretation of the results very difficult for the reader. I suggest the authors to thoroughly review the comments of the Taylor diagrams, trying in a first essay to write down all possible information and only then summarizing the results. I can understand that this is complicated given the large amount of figures, but this is absolutely necessary given the current state of the manuscript.

*This was referring to the different metrics taken into account, namely spatial correlation and RMSE. The RSV is described in the sentences that follow. We reformulated the period to explain more clearly what we want to say. We consider centered RMSE <0.5 and spatial correlation > 90% here and it is made clear by the previous sentences. We would prefer not to extend the period or add the proposed detailed observations, again to make the paper not more lengthy than necessary. Most of them are already included in the second part of the sentence or in the sentences that follow. Our aim is to write down the general outcomes, not a long summation of detailed information that directly is seen from the graph.*
*Indeed, the RMSE for ALARO-0 is always smaller during autumn in the different subregions (and the spatial correlation is often higher), while, as mentioned in the text, this is not the case for REMO which only shows smaller RMSE values during autumn over the EEU, WCA and ESB subregions. However, REMO is closest to CRU during autumn over the full CAS-CORDEX domain.*

• 277-278: not true. What do you consider as "WELL" for normalized standard deviation? For me the good performance of ALARO in EEU in Winter amd REMO in winter over WSB are quite arguable. Also, I would not be very convinced about the goodness of the winter results for the ESB domain for ALARO as well as winter results in WCA and autumn results in TIB for both models.

*Indeed, the RMSE is large in winter for some subregions, but here we are talking about the normalized standard deviation (RSV). We write "in general". By "in general" we mean that the RSV deviates less than 0.25 from 1 in most cases, so we think the statement can be defended. We have added this to the text. If we would expand on all the subtleties it would make the paper again more lengthy.*

• l. 279-280: how would the limited bias explain the higher SD values? biases could be higher but more homogeneous. Reformulate this period.

*Indeed, we now write:*
*"During spring the cold bias in the north is limited to -5 °C for the REMO model but not for ALARO-0, which is reflected in a higher RSV for the northern regions."*

• l. 282-285: This part is not exhaustive and accurate and needs to be reformulated: first of all a similar bipolar behaviour is found for different seasons for both models, not only for summer. One good example is SON spatial biases for the entire domain for REMO and the case of both REMO and ALARO in winter. Here it would be opportune to use the fact that you have small mean biases but large MAE for supporting the conclusions on the fact that the biases are the results of compensation effects. Also, poorer performance than what? Given your conclusions about the performance of ALARO in summer, why your reasoning should work for summer and not for winter, where we also have a very well pronounced bipolar behavior of the bias?

*The bipolar behaviour over the complete domain in winter was pointed out earlier in the text, when we describe Fig. 3 and Table 2. It does not make sense to repeat this here. We moved the explanation for the summer to the same paragraph.*
*Indeed, what we wanted to point out here is poorly formulated. We now write:*
*"The small mean bias during summer (JJA) for ALARO-0 over the complete domain (Table 2) is the result of averaging the warm biases in the south and the cold biases in the north (Fig. 3)." and:*

*"High RSVs are also observed for ALARO-0 in summer over the complete domain (Fig. 4) and this is due to the underestimation of the cold temperatures in cold regions, while warm temperatures are overestimated in regions that are characterized by warmer temperatures (Fig. 3)."*
*The reasoning works as well for winter. In winter a smaller RSV (RSV < 1) over the complete domain is obtained for ALARO-0 and REMO due to the warm bias in the north. This is not the case for the ESB subregion since there are also areas with cold biases that cancel the effect of the warm bias on the RSV out. In other words when colder regions are simulated colder, then the RSV is larger and when colder regions are simulated warmer, then a larger RSV is obtained.*

• l. 290: specify that you are discussing spatial variability

*Good point. In fact, it should be seasonal and not spatial. We now write:*
*"Comparing the metrics of the RCMs (Fig. 3, Fig. 4 and Table 2) shows that REMO is better in simulating the seasonal variability in temperature compared to ALARO-0, except for the autumn in all subdomains and winter in the WSB and TIB subdomain. On the other hand ALARO-0 ..."*

• l. 289-293: I would reformulate this part: you basically say that REMO is better than ALARO in simulating spatial variability, except autumn and winter for some domains. Then you say that ALARO better captures spatial patterns except winter and summer over some subdomains. This sounds a bit contradictory.

*Indeed, but the confusion comes from the notion of the variability. It is seasonal, see our previous reply.*

• Fig. 3: specify in the text that the different Taylor diagrams have different scales.

*We prefer not to overload the text. It is clear when viewing the figure.*

• l. 300: extremely well? it does not seem so for WCA and ESB; Also performance do not seem very good in November for ALARO over WSB.

*For this sentence we based us only on Fig. 4 (now Fig. 5) and not on Table 2, Fig. 2 (now Fig. 3) or Fig. 3 (now Fig.4). The spatially averaged temperatures of the models are within the observational spread or only deviate slightly (<1 °C). With a 2 °C difference, the spatially averaged temperatures for ALARO over WSB and WCA do indeed deviate more from the observational datasets in November. We have now written: "From figure 5, it can be seen that the RCMs simulate the spatially averaged temperatures extremely well during the autumn months (months 9, 10 and 11), since they are within the observational spread or deviate slightly from the observational spread (<1 °C). The exceptions are the spatially averaged temperatures for ALARO-0 over WSB and WCA in November where the spatially averaged temperature deviates 2 °C from CRU." To make the order more consistent with the previous paragraphs we moved these sentences to the end of the paragraph.*

• Fig. 4: I suggest you to split the figures in 3 figures, respectively for T2, TMIn and TMAX, since at the moment the current figure does not allow to appreciate differences in the different cases. Maybe you can introduce figures for TMIN and TMAX in the supplements.

*We prefer to keep this figure in the text since it limits the length of the manuscript and all information is included. As it can be seen in the following figure for mean temperature, splitting the figure into three figures for mean, minimum and maximum temperature does not resolve in showing the differences between the datasets better. If it is not possible to see the differences visually, then the difference between the datasets is < 1 °C, which is small and not worthwhile to mention it.*

[Figure]

• l. 326: what happens for the orographically more complex regions? What are the biases?

*Indeed, that sentence is not clear. We now write:*
*"Annual biases of the minimum temperature over Russia in general vary mostly between -3 °C and 3 °C for REMO and between -1 °C and 5 °C for ALARO-0, with a few exceptions in the orographically complex regions, e.g. in the Stanovoy Range and Central Siberian Plateau where higher biases are found."*

• l. 327: Specify that the bias of REMO over the Eastern part of the domain reaches 15C in winter.

*Indeed, we now write:*
*"The warm biases for REMO in the Eastern part of the domain are most pronounced during winter reaching up to 15 °C. ALARO-0 also shows equally large biases, but ..."*

• l. 329-330: what about all the other parts of the domain in spring and summer?

*In the other parts, we think there is nothing particularly to report.*

• l.345-354: Quantify values of bias and MAE.

*We did not specify the values for the Table 2 within the text, since the purpose of the table is precisely to list them in a way that allows to easily overview them. We discuss the table only qualitatively. To stay consistent and to avoid that the paper becomes even longer we do not do so for Table 3.*

• l. 352-352: this does not seem the case for autumn and annual values of ALARO and for summer for REMO

*Indeed, but we prefer to make a general statement here. We write it now more precisely:*

*"The normalized Taylor diagrams in Fig. 7 confirm that, in general, the RCMs struggle to simulate the spatial pattern of minimum temperature well over the north-eastern part of the domain (ESB), while on annual level ALARO-0 is able to simulate the spatial pattern well."*

• fig.6: I think you are not giving the same importance in the discussion of Fig 6 as for Fig. 3.

*The reader should, at this point, be familiar with reading the Taylor diagrams. Indeed, as mentioned in reply to your previous comment, we prefer to limit us here to general statements. We prefer not to extend the paper.*

• Fig. 6: one general question concerning all Taylor diagrams: do you have any clue why you generally get high values of correlation for the entire domain, when in all subdomains you obtain smaller correlations?

*In fact, there is no reason why this should not be the case. Positive and negative terms can cancel each other out. When computing the spatial correlation over the CAS domain, biases are more likely to have an opposite sign (canceling each other out) compared to the subregions. Not surprisingly, the models simulate the global climate better than the regional ones.*

• l. 361: not exact: REMO better also for EUU in summer and winter and in WCA for spring.

*We now write: "REMO has a better centered RMSE and spatial variability during summer, except for the WCA region". It would require extra sentences to describe both RMSE and spatial variability in detail for each season since the outcomes differ quite a bit for each subregion. The manuscript is already long, so we prefer to not add this.*

• l. 361: specify spatial variability

*Indeed, we now write: "On the other hand, REMO has a better centered RMSE and spatial variability during summer, except for the WCA region ... "*

• l. 366: what happens in the Tibetan plateau? what is the magnitude of these biases?

*Indeed, this paragraph can be formulated in a better way. We rewrote the paragraph including more numbers of the biases that can be seen on Fig. 7 (now Fig. 8).*

• l. 365-370: not only here, but also for other figures, fix an order for the discussion of the figures (for example from top to bottom) and follow it throughout the text

*As said in the reply to your general comment, we have paid attention to the structure of the text. However, this is not always possible when certain features appear for different seasons since we prefer to mention this at the same time to limit the length of the text.*

• l.367-368: cold bias over northern part of the domain is present in all seasons, except DJF.

*Indeed, this can be improved. We now write: "Biases in Fig. 8 and Table 4 show that for both RCMs a pronounced cold bias is present for maximum temperatures over the northern part of the domain at the annual scale and for all seasons, except for ALARO-0 in winter."*

• l. 364-371: reformulate and extend all the period. Some parts of the domain are never mentioned. What happens for example in the Himalayas and the Arabian Peninsula? exhaustive description of summer is missing.

*We rewrote the paragraph. Also here, we prefer to not extent the manuscript. For the Himalayas it is obvious. For the Arabian Peninsula there is nothing substantial to report. It would only make the manuscript longer.*

• l. 378-379:Specify that in the case of ALARO the bias exceeds 7C.

*This is substantial. We added the sentence: "Both RCMs have a cold bias over a large area in the north during spring, which is very pronounced for the ALARO-0 model in the north-east (< -15 °C), while the biases remain limited to -7°C for REMO (Fig. 8)."*

• l. 376-385: you should extend the part describing the tables, discussing for example biases and MAE, giving indications on the fact that in some case biases are the result of compensation.

*All the information is in the tables and the figures. We do not think that every value in the table and figure has to be repeated in the text. So we prefer to not extend the manuscript since it is already quite long.*

• l. 380: not very accurate. In summer REMO is better for the Tibetan plateau in terms of MAE, while ALARO is better in winter for the EEU domain

*Indeed, we adjusted this sentence.*

• l. 381: how can you conclude, from the sentence before where you state that the 2 models are better in autumn, that ALARO simulates TMAX poorly in any season? actually there are 3 cases where ALARO is better than REMO: DJF in EEU and MMA and SON in TIB. Anyhow, in many cases the results of the 2 models are very similar and the MAE is very close to 1. Importantly, you again omit to specify what is good and what is bad, in terms of the given metrics.

*Indeed, this sentence is incorrect, we removed the sentence: "From this we can conclude that ALARO-0 simulates the maximum temperature poorly in any season."*

• l. 403: it would be better to have a map of the bias of DTR. This could help your discussion that now is too generic based only on the maps of TMIN and TMAX.

*We agree that this would be useful, but the discussion is already very exhaustive and we have to put a limit somewhere. So we prefer to not extend the text.*

• l. 406: the model does not restore its balance, since it is also in equilibrium in winter. Reformulate.

*Indeed. We now write: "In summer the model is able to evolve to a more correct balanced state and to simulate spatial averaged minimum temperatures ..."*

• l. 414-415: Where? In the other sections you started commenting the figures and then the tables. Why you change this now?

*Indeed, as said in reply to your general comment, we have restructured the text.*

• l. 415-420: what about annual values in table 5? what about the spread of observations?

*Also here, we have to draw a limit and we cannot include all numbers in the text.*

• l. 417-418: please quantify all the biases you mention.

*Also here, the purpose of the table is to list the biases. If they can be found in the table there is no need to repeat them all in the text. The purpose of the text is to describe the main features.*

• l. 419: are you sure the Tibetan plateau can be classified as a monsoon region?

*We now write: "... over the annual cycle for both RCMs over the East Asian monsoon region, with a less notable wet bias during summer ..."*

• l. 418-420: which figure are you commenting now? If you are discussing Fig. 11, actually it does not seem that the bias is smaller but greater in summer, at least for ALARO.

*This is now clear with the new structure of the text.*

• l. 423-424: It is not totally true that the largest biases are present over extremely dry areas. One example is Northern India in Summer, presenting a remarkable dry bias despite observations are characterized by highest precipitation values.

*We now write: "Some of the largest relative biases can be found in relatively dry regions ..."*

• l. 422-423: specify that when you talk about low precipitation you are referring to the observations.

*We now write: "This is partly due to the low observed precipitation quantities in several regions ..."*

• l. 425: actually over Northern China in REMO the bias exceeds 5mm/month.

*This sentence is wrong in this context, it was written for the Gobi desert region only. We have removed it since it does not add any value here.*

• l. 433: "is also present for REMO": why also?

*Indeed, it is not clear what "also" refers to. We removed it.*

• l. 435-436: which model are you referring to?

*Indeed. We now write: "The wet bias for REMO over ESB during spring is low ..."*

• l. 438 and l. 447: greater than -2mm/month

*Indeed, we wanted to point out the smaller absolute value of the bias. We changed smaller than into greater than -25 mm/month here.*

• l. 450-451: how can you claim that the spatial patterns are well represented by ALARO if for each subdomain, despite WCA, the considered metrics present relatively poor values (especially in terms of correlation)?

*It is better in comparison to REMO.*

• l. 449-452: why do you not discuss the observations?how do the different data-sets compare to each others? this is something that you should do when commenting all Taylor diagrams.

*The spread between the observational datasets and the implications on the results are according to the structure of the text described in the discussion section. Again, we have to put a limit to the content of the paper, it is already long, so we do not discuss all the details of the different observational datasets.*

• You can extend the discussion of Fig. 10, consistently with the other subsections.

*Also here, we have to put a limit to the content of the paper, it is already long.*

• l. 476-479: Please specify the cases when the bias exceeds the spread of observations. At the moment it seems like the spread of observations is smaller than the bias in almost all the domain and seasons. Is that correct? So your figures would confirm that evinced biases are more inherent to the model than to observations, over almost all the domain in all seasons. One more elegant way (and probably more useful for your goals) to determine those points where the spread of observation exceeds the bias, is by plotting a map of the bias (in the supplements) with a point in correspondence of those points where the spread is larger than the biases.

*Yes, the spread of the observations is smaller than the bias for a large part of the domain. We did write this down in the next paragraph: "Figure S1 shows that for the majority of grid points the mean temperatures of ALARO-0 and REMO lie within the range of spread between the reference datasets during autumn." "During winter and spring none of the RCMs are able to reproduce temperatures that can be completely explained by the observational uncertainty over a large part of the CAS-CORDEX domain, while this is also the case for ALARO-0 during summer (Fig. 3 and Table 2)." The maps in the supplementary material give the information of those points where the spread is larger than the biases, namely the areas covered by the white color that corresponds with values < 0. Everything in red is above 0 and indicates thus that the bias between the RCM and CRU is larger than the observational spread. We chose to add a scale with multiple levels, to give additional information about where the bias of the RCMs is much larger than the spread between the observational datasets. All the information is in the current figures or can be derived from the figures.*

• l. 482-484: be more precise. It is not clear what you want to express. The observational spread is significantly high over complex-orography regions and not over the entire north-eastern part of the domain.

*We now write: " ...which makes the evaluation of the models less reliable over these mountainous regions."*

• l. 486-488: also REMO bias exceeds the spread of the observations over large parts of Mongolia and Northern China.

*Yes, but we have to put a limit to the scope of the paper.*

• l. 489-490: reformulate

*We now write: "For instance, the strong biases in the north-eastern part of the domain for ALARO-0 during winter and spring exceed the spread in temperatures between the different reference datasets, indicating that ..."*

• l. 497-498: This is also true for EEU in autumn

*No, we do not agree. Over EEU, the MAE for both RCMs is larger in autumn compared to the MAE of MW and ERA-Interim.*

• l. 531: Also true over mountainous regions of north-eastern part of the domain.

*Yes, but we limit the discussion here.*

• l. 536-538: but also in summer, at least for ALARO

*Idem as for the previous reply.*

• l. 542-543: Reference needed.

*We added the references Jacob et al. (2012) and Remedio et al. (2019).*

• l. 542-542: what are these processes?

*We do not know. We only say that we see the same features here, so it must be the same processes that are the source of it.*

• l. 543-544: why should it be? reformulate. Better specify what do you mean by shift in the annual cycle?

*We mean a temporal shift in the annual cycle. We now write: "The warm bias during winter and cold bias during spring in the north-eastern part of the domain for ALARO-0 are not due to a temporal shift in the annual cycle in the northern part of the domain, ..." When you briefly look at the maps it looks like there is a temporal shift because of the warm temperatures in winter and the cold temperatures in spring, but this is not the case.*

• l. 546: how can you state that the bias increases when the snow-covered region expands? Have you directly analyzed snow cover in the two models? If what you affirm is true, this should be a feature of both models. However, winter biases are different in the two cases.

*The zone with a warm bias becomes larger during the months when the snow cover expands. We investigated for both models and ERA5 the monthly evolution of the snow cover but we did not implement the figures here, since this is not the aim of the paper and to overcome an even longer text. ALARO-0 produces too much snow over a too large extent compared to ERA5. The snow covered regions are not the same and this makes it more complex to understand the temperature biases that are warm for the regions where both ALARO-0 and ERA5 show snow cover. Therefore we wrote: "This shows that a more complex multi-layer snow scheme might not be enough to solve the warm bias for ALARO-0 during winter. Therefore, further investigation should be done to see whether the warm bias in winter over the northern part of the domain is due to the inability of the current snow scheme to reproduce the heat conductivity of snow." Here we discuss only the biases in ALARO-0 as written in the text, since for REMO the snow covered regions agree well with ERA5. The sentence at line 546 is only an observation. We cite the analysis of Mašek (2017) that reported the same observation. We identified that this error can be traced to the interactions with the surface which manifest themselves when the snowpack is growing. More detailed research is needed to find out what exactly is going on.*

• l. 545-546: are you sure about the warm bias appearing in the North during autumn when the snow appears over this region? Again, did you base these statements on some analysis of simulated snow cover? Warm bias is very limited to a very small eastern part of the northern domain in ALARO.

*Yes, we did check the snow cover and it was seen that the warm bias appears over regions where both ALARO-0 and ERA5 simulate a snow cover. In October there is only snow in the most northern part of the domain. In November most of the northern subregions are having a significant amount of snow cover. Averaging over the three months reduces the warm bias present over the northern region with snow during November to only a small area with warm bias in the northeast. The cold bias in the northeast can be explained by the fact that this is a region that is covered by snow for ALARO-0 but not for ERA5 during autumn.*

• l. 548-549: I would argue against your conclusions, based on the fact that also REMO is characterized by a very warm bias in winter over the North-eastern part of the domain. I do not think that based on your analyses you can raise strong conclusions on the driver of the bias over the northern part of the domain in both models. For sure I would not state that REMO does not encounter the same problem.

*Perhaps REMO also has underlying problems but more research should be done to investigate this and to know the cause. And if it has, it does not manifest itself as strongly as ALARO-0.*
*The warm bias for REMO does rather occur for regions where there is no snow (or not enough snow) simulated by REMO, while there is a (larger) snowpack present for ERA5. The opposite is true for ALARO-0, there is more snow simulated than given by ERA5, so as we mentioned in the text, the biases cannot be explained by the same feature.*

• l. 570: wouldn't Eastern Europe be more consistent with your discussion than Western Russia?

*No, since we aim here to exclude some East-European countries such as Poland, Lithuania, Latvia, Estonia… where the bias in minimum or maximum temperature is larger than the spread (now Fig. S2 and S3). When investigating Fig. 4 (now Fig. 5) it is seen that the averaged values of the EEU subregion are not completely within the observational range for minimum temperature, so we cannot change it to EEU.*

• l. 571: Can you better specify what means acceptable? I would emphasize that the MAE in this case is smaller than the MAE between ERAInterim and CRU.

*This is literally what we wrote in the second part of this sentence: "… since the MAE between ERA-Interim and CRU is larger (Table 3 and 4)." To make it even more clear we now write: "…since the MAE between ERA-Interim and CRU is larger than the MAE between REMO and CRU (Table 3 and 4)."*

• l. 575-577: How can you state this without a map of the bias in DTR? can you be sure that for all the points of the domain RCMs produce a smaller DTR in all seasons?

*Indeed, but we do not intend to make a mathematical statement here. We now write: "Both RCMs generally produce a smaller daily temperature range …"*

---

## Author Response (AR4)

**Reply to the review of "Evaluation of regional climate models ALARO-0 and REMO2015 at 0.22 resolution over the CORDEX Central Asia domain" by Top et al. 2020**

**List of relevant changes**

*Dear Editor,*

*Thank you for mentioning the minor detailed corrections. We took all of them into account. According to your suggestion, we changed the scales of the maps to a sequential color range (Figures 3, 6, 8, 10 and S4). We are grateful for the effort you have put in the review process of our paper. The contribution of the reviewers and yourself have been added to the acknowledgement section of the text.*